# Safe-EF: Error Feedback for Non-smooth Constrained Optimization

**Rustem Islamov** [1]  **Yarden As** [2 3]  **Ilyas Fatkhullin** [2 3]

## Abstract

Federated learning faces severe communication bottlenecks due to the high dimensionality of model updates. Communication compression with contractive compressors (e.g., Top-$K$) is often preferable in practice but can degrade performance without proper handling. Error feedback (EF) mitigates such issues but has been largely restricted for smooth, unconstrained problems, limiting its real-world applicability where non-smooth objectives and safety constraints are critical. We advance our understanding of EF in the canonical non-smooth convex setting by establishing new lower complexity bounds for first-order algorithms with contractive compression. Next, we propose Safe-EF, a novel algorithm that matches our lower bound (up to a constant) while enforcing safety constraints essential for practical applications. Extending our approach to the stochastic setting, we bridge the gap between theory and practical implementation. Extensive experiments in a reinforcement learning setup, simulating distributed humanoid robot training, validate the effectiveness of Safe-EF in ensuring safety and reducing communication complexity.

## 1. Introduction

Federated learning is a crucial framework for training machine learning models across distributed environments (Konečný et al., 2016; Kairouz, 2019), where data is naturally stored in a distributed fashion. Formally, such problems can be expressed as

$$\min_{x \in \mathcal{X}} f(x) := \frac{1}{n} \sum_{i=1}^{n} f_i(x), \tag{1}$$

where $n$ represents the number of workers or machines participating in the training, and $x \in \mathbb{R}^d$ denotes the model

parameters to be optimized. The function $f_i \colon \mathbb{R}^d \to \mathbb{R}$ is the local (possibly non-smooth) loss associated with data on worker $i \in [n] := \{1, \dots, n\}$, and $\mathcal{X}$ is a subset of $\mathbb{R}^d$.

This paradigm is particularly valuable in privacy-sensitive and resource-constrained settings, where data remains decentralized, and collaboration is achieved without requiring direct data sharing. For instance, consider a fleet of robots that operate in homes (Kalashnikov et al., 2018; Brohan et al., 2022). In such settings, traditional centralized learning approaches are impractical, as transmitting raw sensory data from each robot to a central server would pose severe privacy risks and require enormous bandwidth. Furthermore, these robots must adapt to diverse household environments, necessitating personalized learning while still benefiting from collective experience across the fleet. Despite its advantages, distributed training faces significant communication bottlenecks due to the high dimensionality of model updates. This challenge necessitates the development of communication-efficient algorithms.

**Communication compression with Top-$K$.** One prominent strategy to reduce communication costs is the communication compression technique, which applies possibly randomized compression to updates prior to transmission. One of the most practical and versatile classes of compression operators are those that satisfy the contractive property:

$$\mathbb{E}\left[\|\mathcal{C}(x) - x\|^2\right] \le (1 - \delta)\|x\|^2 \qquad \text{for all } x \in \mathbb{R}^d,$$

where $\delta \in (0, 1]$ represents the accuracy of the compression. Prominent examples are Top-$K$ sparsifier that preserves $K$ largest components of vector $x$ in magnitude, and random sampling methods such as Rand-$K$ that preserves a subset of $K$ components of $x$ chosen uniformly at random. Although both Top-$K$ and Rand-$K$ are contractive with $\delta \ge K/d$, methods utilizing Top-$K$ operator are often empirically superior due to their greedy nature (You et al., 2016).

**Non-smooth challenges.** The majority of works focusing on communication compression assume that the objective function is smooth, i.e., differentiable with Lipschitz continuous gradient, simplifying theoretical analysis (Stich et al., 2018; Richtárik et al., 2021). However, this assumption limits the applicability of developed methods to many real-world problems, where non-smooth functions frequently arise. For instance, consider problems involving ReLU activations (Glorot et al., 2011) or clipped objectives such

---
[1]University of Basel, Switzerland [2]ETH Zürich, Switzerland [3]ETH AI Center, Switzerland. Correspondence to: Rustem Islamov <rustem.islamov@unibas.ch>.

*Proceedings of the 42nd International Conference on Machine Learning*, Vancouver, Canada. PMLR 267, 2025. Copyright 2025 by the author(s).

as those in proximal policy optimization (PPO, Schulman et al., 2017). This motivates the first key question of our study:

> **Question 1:** What are the limits of compressed gradient methods in the non-smooth regime?

To illustrate the challenges of designing meaningful methods with contractive compressors like Top-$K$, we present a non-convergence example for vanilla compressed gradient descent (CGD) in the non-smooth setting. Consider

$$\text{CGD} \qquad x^{t+1} = x^t - \frac{\gamma}{n} \sum_{i=1}^{n} \mathcal{C}(f_i'(x^t)), \qquad (2)$$

where $f_i'(x^t) \in \partial f_i(x^t)$ is a subgradient of $f_i$ and $\gamma \geq 0$ is a stepsize.

**Example 1.1.** *For any $n \geq 1$, there exists a specific instance of problem (1) where $\mathcal{X} = \mathbb{R}^2$, and $f(x) = \|x\|_1$ is non-smooth, convex, and 1-Lipschitz continuous. For this instance, with some initial vector $x^0 \in \mathbb{R}^2$, the iterates of* CGD *(2) applied with the Top-1 compressor and any stepsize $\gamma \geq 0$, satisfy*

$$f(x^t) - \min_x f(x) = 1 + \frac{\gamma}{2} \qquad \text{for any } t \geq 0.$$

This example implies that running vanilla CGD with the Top-1 compressor even on a simple non-smooth problem may yield no improvement. It is remarkable that this failure occurs even in the identical data regime $f_i = f$ for all $i \in [n]$, the setting where CGD is known to converge in smooth case (Nesterov, 2012; Nutini et al., 2015; Beznosikov et al., 2023). The idea of the construction in Example 1.1 is that due to a rapid change of the gradients $f'$ in consecutive iterations, CGD consistently ignores the direction of the second component of $x^t$, which results in a pathological cyclic behavior. See Figure 1 for an illustration.

**Error feedback can make things worse!** A common remedy for non-convergence issues of compressed gradient methods is error feedback (EF), a mechanism that has inspired several variants (Seide et al., 2014; Richtárik et al., 2021; Fatkhullin et al., 2024; Gao et al., 2024). Among these, EF21 is a recent approach with state-of-the-art performance guarantees in smooth optimization due to Richtárik et al. (2021):

$$\text{EF21} \qquad \begin{aligned} x^{t+1} &= x^t - \gamma\, v^t, \qquad v^t = \frac{1}{n} \sum_{i=1}^{n} v_i^t, \\ v_i^{t+1} &= v_i^t + \mathcal{C}(f_i'(x^{t+1}) - v_i^t). \end{aligned} \qquad (3)$$

where $f_i'(x^{t+1}) \in \partial f_i(x^{t+1})$ is a subgradient of $f_i$ and $v_i^t$ is a local gradient estimator at each worker. While Richtárik et al. (2021) only analyze this algorithm in the smooth non-convex case, we extend its analysis to smooth convex setup in Appendix C. However, we show that, surprisingly, EF21 fails to converge on the same problem as CGD.

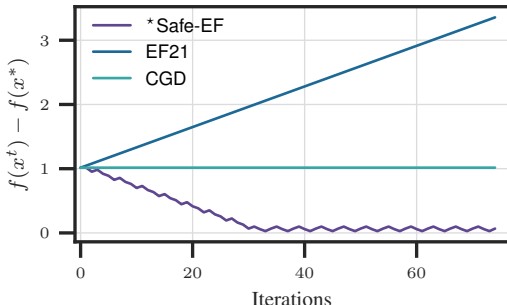

Figure 1: Non-convergence of CGD, divergence of EF21 and convergence of Safe-EF for the problem $f(x) = \|x\|_1$, $i = 1, d = 2$ used in the proofs of Examples 1.1 and 1.2 with Top-1 compressor. We run all algorithms for $T = 10^3$ iterations with $x^0 = (\gamma/2, -1)^\top$, $\gamma = 1/\sqrt{T}$, and $v^0 = (1, 1)^\top$ (for EF21). *Safe-EF coincides with EF14 (Seide et al., 2014) in this example.

**Example 1.2.** *Consider the problem instance from Example 1.1. For this instance, with some initial vectors $x^0, v^0 \in \mathbb{R}^2$, the iterates of* EF21 *(3) applied with the Top-1 compressor and any stepsize $\gamma \geq 0$ satisfy*

$$f(x^t) - \min_x f(x) = 1 + \frac{\gamma}{2} + t\gamma \qquad \text{for any } t \geq 0.$$

This example shows that EF21 does not converge for non-smooth problems despite achieving an excellent performance in smooth case, see Theorem C.1, and reaching the optimal iteration complexity in smooth non-convex optimization (Huang et al., 2022). Moreover, if we pick the classical stepsize $\gamma = 1/\sqrt{T}$, EF21 diverges from the optimum with a rate $\Omega(t/\sqrt{T}) \approx \sqrt{T}$ for $t \approx T$, which is even worse than CGD. We show the divergence in Figure 1, where we also observe that another EF variant, EF14[1], (Seide et al., 2014) converges without problems. We find such stark difference surprising in light of the equivalence of EF21 and EF14, established under additivity assumption of $\mathcal{C}$ (Richtárik et al., 2021). The catch is that Top-1 is not additive, and thus the equivalence does not hold here.

*Motivated by this fairly toy example, we find it important to understand EF in non-smooth setup, and aim to study* EF14.

**Safety considerations.** In addition to these challenges, safety constraints play a critical role in real-world applications (Altman, 1999). Ensuring solutions satisfy feasibility requirements is essential, particularly in scenarios like federated reinforcement learning (FedRL) (Nadiger et al., 2019; Qi et al., 2021; Jin et al., 2022). Despite their importance, constrained optimization with communication compression remains under-explored. Although some work develop methods assuming $\mathcal{X}$ is simple, i.e.,

---

[1]Our Safe-EF method presented in Algorithm 1 reduces to EF14 in unconstrained setting with $\mathcal{C}_0 = \mathbf{Id}$.

using projection (Fatkhullin et al., 2021) or linear minimization (Nazykov et al., 2024) oracles, they crucially rely on smoothness. Moreover, the applications in Safe FedRL motivate us to pay attention to problems with more complex constraints of the form

$$\mathcal{X} := \left\{ x \in \mathbb{R}^d \mid g(x) := \frac{1}{n} \sum_{i=1}^{n} g_i(x) \leq 0 \right\}, \quad (4)$$

where $g_i \colon \mathbb{R}^d \to \mathbb{R}$ defines a constraint for worker $i$.

> **Question 2:** Can we design a provably convergent compressed gradient method with a Top-$K$ compressor for non-smooth constrained problems?

Perhaps, the most common approach to solve (1) with (4) in non-distributed optimization ($n = 1$) is to reformulate it as a saddle point problem, which is then solved by primal-dual methods (Nemirovski, 2004; Hamedani & Aybat, 2021). This approach is popular in practice (Ding et al., 2020; Moskovitz et al., 2023; Ding et al., 2024; Müller et al., 2024) and has rich theory, e.g., (Boob et al., 2023; Boob & Khalafi, 2024; Zhang & Lan, 2022). However, such methods have several limitations. First, they are known to be sensitive to the tuning of the initial dual variable (e.g., the experiments and discussion in Appendix G) and often require an estimate of the upper bound of the optimal dual variable. Second, their theoretical justification often requires projecting both primal and dual variables onto an unknown bounded set, which is not aligned with practical implementations. In the context of EF-type methods, this projection requirement implies several algorithmic and technical challenges because only certain smooth variants of EF seem to be compatible with projection, e.g., (Fatkhullin et al., 2021). An alternative is to adopt a primal only approach, e.g., switching subgradient (Polyak, 1967; Lan & Zhou, 2020; Ma et al., 2020; Huang & Lin, 2023; Jia & Grimmer, 2022), methods based on the velocity field (Yu et al., 2017; Muehlebach & Jordan, 2022; Schechtman et al., 2022; Kolev et al., 2024), or level-set methods (Lin et al., 2018; Boob et al., 2024). Primal methods have also been used in (non-distributed) RL applications, e.g., (Xu et al., 2021; Chen et al., 2021; Jordan et al., 2024; Li et al., 2024). The key advantage of such primal schemes is their simplicity and convergence under mild assumptions without the need for the estimation of dual variables.

## 2. Contributions

- First, we establish a $\Omega\left(\frac{MR}{\sqrt{\delta T}}\right)$ convergence lower bound for non-smooth convex distributed optimization with contractive compressors for function suboptimality gap and a constraint violation. Here $T$ is the iteration count, $R$ is the initial distance to the optimum, $M$ bounds the norm of subgradients of $f_i$, and $\delta \in (0, 1]$ is the compression accuracy.

- Next, we propose Safe-EF (Algorithm 1), an extension of EF14 (Seide et al., 2014) incorporating safety constraints (4) via primal switching subgradient approach and bidirectional compression including the workers to server compressor $\mathcal{C}_0$. Safe-EF provably works in non-smooth distributed settings and efficiently minimizes the objective function, while controlling the constraint violation. We prove the convergence rate of Safe-EF matching the above-mentioned lower bound up to a numerical constant under a constant accuracy of the server compression $\mathcal{C}_0$. It seems our upper bound is new even when $g(x) \equiv 0$ and $\mathcal{C}_0 = \mathbf{Id}$.

- We further study Safe-EF in practically relevant stochastic scenarios, where exact subgradients and function evaluations are unavailable and need to be estimated. We establish high probability bounds with a mild logarithmic dependence on failure probability, which is significant even without compression, since our bounds feature the distance to the optimum $R$ instead of the diameter of the set, which is not bounded in our set-up.

- Finally, we conduct extensive experiments and ablation studies of Safe-EF, putting the method to the test on a challenging task of distributed humanoid robot training and providing important practical insights into the performance of non-smooth EF methods.

## 3. Assumptions and Communication Protocol

We consider distributed constrained optimization problem (1) with a constraint (4), and denote the optimal solution to this problem by $x^*$. Unless specified otherwise, we denote by $\|\cdot\|$ the Euclidean norm in $\mathbb{R}^d$.

**Assumption 3.1.** We assume that $f_i$ and $g_i$ are convex for all $i \in [n]$, namely, for all $x, y \in \mathbb{R}^d$ we have

$$\begin{aligned} f_i(y) &\geq f_i(x) + \langle f_i', y - x \rangle \quad \forall f_i' \in \partial f_i(x), \\ g_i(y) &\geq g_i(x) + \langle g_i', y - x \rangle \quad \forall g_i' \in \partial g_i(x). \end{aligned} \quad (5)$$

Each worker $i$ has access to the oracles $O_{f_i,i}(x)$ and $O_{g_i,i}(x)$, which return the subgradients $f_i' \in \partial f_i(x)$, $g_i' \in \partial g_i(x)$, and the function values $f_i(x)$, $g_i(x)$ respectively for any $x \in \mathbb{R}^d$. We assume bounded subgradient, which is a common assumption in non-smooth optimization (Nesterov et al., 2018)

**Assumption 3.2.** We assume that $f_i$ and $g_i$ have $M$-bounded subgradients, i.e. for any $x \in \mathbb{R}^d$ and $i \in [n]$ we have

$$\max \{\|f_i'(x)\|, \|g_i'(x)\|\} \leq M. \quad (6)$$

We let the function classes $\mathcal{F}_{R,M}$ and $\mathcal{G}_{RM}$ denote the set of all functions satisfying Assumptions 3.1-3.2 for any underlying dimension $d$ and a given initialization $x^0 \in \mathbb{R}^d$

such that $\|x^0 - x^*\| \leq R$. We denote by $\mathcal{H}_{R,M}$ the class of problems of form (1), (4), where functions $\{f_i\}_{i=1}^n$ and $\{g_i\}_{i=1}^n$ are taken from $\mathcal{F}_{R,M}$ and $\mathcal{G}_{R,M}$ respectively.

**Compression operators.** We focus on the class of algorithms using contractive compressors.

**Definition 3.3.** We say that a (possibly randomized) mapping $\mathcal{C}: \mathbb{R}^d \to \mathbb{R}^d$ is a contractive compression operator if for some constant $\delta \in (0, 1]$ it holds

$$\mathbb{E}\left[\|\mathcal{C}(x) - x\|^2\right] \leq (1 - \delta)\|x\|^2. \tag{7}$$

Beyond Top-$K$ and Rand-$K$ mentioned in Section 1, examples satisfying (7) include sparsification (Alistarh et al., 2018; Stich et al., 2018; Islamov et al., 2021) and quantization (Wen et al., 2017; Bernstein et al., 2018; Horváth et al., 2022; Compagnoni et al., 2025) techniques, and low-rank approximations (Vogels et al., 2019; Qian et al., 2021a; Islamov et al., 2023). We refer to (Beznosikov et al., 2023; Safaryan et al., 2021) for further examples. We denote by $\mathbb{C}(\delta)$ the set of all $\delta$-contractive compressors.

**Algorithm class.** We follow Huang et al. (2022) to introduce the class of algorithms of interest. We consider a centralized and synchronous algorithm $A$, where: $i$) workers are restricted to communicating directly with a central server and cannot exchange information with one another directly; $ii$) all iterations are synchronized, meaning all workers begin each iteration simultaneously. In this setup, each worker $i$ maintains a local copy of the model, denoted as $x_i^t$, at iteration $t$. The output $\hat{x}^t$ of the algorithm $A$ after $t$ iterations can be expressed as any linear combination of all previous local models, namely,

$$\hat{x}^t \in \text{span}\left(\{x_i^s : 0 \leq s \leq t, 1 \leq i \leq n\}\right). \tag{8}$$

We additionally require that the algorithm $A$ satisfies the "zero-respecting" property (Carmon et al., 2020; Lu & De Sa, 2021). This ensures that the number of non-zero entries in a worker's local model can only increase through local subgradient queries, or synchronization with the central server. This property is upheld by a broad range of existing distributed optimization algorithms (Tang et al., 2019; Xie et al., 2020; Richtárik et al., 2021; Gao et al., 2024). In addition to these properties, the algorithm $A$ must support communication compression. To achieve this, each worker $i \in [n]$ is equipped with a compressor $\mathcal{C}_i$. The formal definition of this algorithm class with worker to server compression is provided below, see Appendix E for details.

**Definition 3.4.** Given compressors $\{\mathcal{C}_1, \ldots, \mathcal{C}_n\}$, we denote $\mathcal{A}_{\{\mathcal{C}_i\}_{i=1}^n}^U$ as the class of all centralized, synchronous, zero-respecting algorithms that support unidirectional compression, where compressor $\mathcal{C}_i, i \in [n]$, is applied to messages from worker $i$ to the server.

## 4. Main Results

We start by presenting our first main contribution, which is the lower iteration/communication complexity bound for a class of first-order compressed gradient methods.

### 4.1. Lower Bound

Given a problem $h := (\{f_i\}_{i=1}^n, \{g_i\}_{i=1}^n) \subseteq \mathcal{H}_{R,M}$, subgradient/function value oracles $\{\mathcal{O}_{f_i,i}\}_{i=1}^n, \{\mathcal{O}_{g_i,i}\}_{i=1}^n$, compressors $\{\mathcal{C}_i\}_{i=1}^n \subseteq \mathbb{C}(\delta)$, and an algorithm $A \in \mathcal{A}_{\{\mathcal{C}_i\}_{i=1}^n}^U$, let $\hat{x}_{A,T} := \hat{x}_{A,\{f_i\}_{i=1}^n,\{g_i\}_{i=1}^n,\{\mathcal{C}_i\}_{i=1}^n,T}$ represent the output of algorithm $A$ after at most $T$ oracle queries and communication rounds per worker. We define the minimax convergence measure

$$\inf_A \sup_{\{\mathcal{C}_i\}_{i=1}^n} \sup_{h \in \mathcal{H}_{R,M}} \left\{\mathbb{E}\left[f(\hat{x}_{A,T}) - f(x^*)\right], \mathbb{E}\left[g(\hat{x}_{A,T})\right]\right\}.$$

We do not require operators $\{\mathcal{C}_i\}_{i=1}^n$ to be neither distinct nor independent, and parameter $\delta$ can be utilized by the algorithm $A$. Our first contribution is the lower bound for algorithms that support unidirectional compression.

**Theorem 4.1.** *For any $R, M > 0, n \geq 2, \delta \leq 0.3, T \geq \delta^{-2}$ there exists a problem $h \subseteq \mathcal{H}_{R,M}$, oracles $\{\mathcal{O}_{f_i,i}\}_{i=1}^n$, $\{\mathcal{O}_{g_i,i}\}_{i=1}^n$, compressors $\{\mathcal{C}_i\}_{i=1}^n \subseteq \mathbb{C}(\delta)$, and the starting point $x^0 = 0$ such that for any first-order algorithm $A \in \mathcal{A}_{\{\mathcal{C}_i\}_{i=1}^n}^U$ run for $T \leq d$ iterations from $x^0$, satisfies*

$$\mathbb{E}\left[f(\hat{x}_{A,T}) - f(x^*)\right] \geq \Omega\left(\frac{RM}{\sqrt{\delta T}}\right), \quad \text{and}$$

$$\mathbb{E}\left[g(\hat{x}_{A,T})\right] \geq \Omega\left(\frac{RM}{\sqrt{\delta T}}\right). \tag{9}$$

When $\delta = 1$ and $g \equiv 0$, indicating no compression and no constraints, (9) recovers the classical lower bounds for non-smooth convex optimization (Nemirovskij & Yudin, 1983; Nemirovski, 1994; Nesterov, 2014; Braun et al., 2017; Scaman et al., 2018). However, when worker to server compression is large, the convergence rate degrades by a factor of $1/\sqrt{\delta}$. Similar degradation appears in the constraint violation. An interesting implication of Theorem 4.1 is that the convergence rate does not improve when increasing the number of workers $n$, which is different from prior work in smooth stochastic optimization (Huang et al., 2022; He et al., 2023). The key idea of the proof is to extend and modify the "worst-case" function from (Nesterov, 2014) and account for compression in the distributed setting, specifically, we use for all $i \in [n]$

$$f_i(x) := C \cdot \max_{1 \leq j \leq T} x_j + \frac{\mu}{2}\|x\|_2 \cdot \max\left\{\|x\|_2; \frac{R}{2}\right\},$$

$$g_i(x) := f_i(x) - \min_{x \in \mathbb{R}^d} f_i(x),$$

where $C, \mu > 0$ are some constants depending on the bound of subgradients $M$ and the compression level $\delta$. We refer to Appendix E for the formal proof.

---

**Algorithm 1** Safe-EF with bidirectional compression

---

1: **Input:** $w^0 = x^0, \{\mathcal{C}_i\}_{i=1}^n, \gamma, c > 0, e_i^0 = 0$
2: **for** $t = 0, \ldots, T - 1$ **do**
3:     **for** $i = 1, \ldots, n$ **in parallel do**
4:        Send $g_i(x^t)$ to server     ➤ cheap one float comm.
5:     **end for**
6:     Send $g(x^t) = \frac{1}{n}\sum_{i=1}^n g_i(x^t)$ to workers
7:     **for** $i = 1, \ldots, n$ **in parallel do**
8:        Compute $h_i^t = f_i'(x^t)$ **if** $g(x^t) \le c$ **else** $g_i'(x^t)$
9:        Send $v_i^t = \mathcal{C}_i(e_i^t + h_i^t)$ to server
10:       Compute $e_i^{t+1} = e_i^t + h_i^t - v_i^t$
11:     **end for**
12:     Compute $v^t = \frac{1}{n}\sum_{i=1}^n v_i^t$ and $w^{t+1} = w^t - \gamma v^t$
13:     Compute $x^{t+1} = x^t + \mathcal{C}_0(w^{t+1} - x^t)$
14:     Send $\mathcal{C}_0(w^{t+1} - x^t)$ to workers
15: **end for**

---

## 4.2. Safe-EF Method

In this section, we describe Safe-EF, our main algorithm detailed in Algorithm 1, which addresses two main challenges simultaneously: handles *non-smoothness* and *constraints*. The distinct feature of our method is a dynamical switch between the subgradients of the objective $f_i$ and those of the constraints $g_i$ depending on if the constraint violation exceeds a predefined threshold $c$. To implement this, workers compute the constraint violations $g_i(x^t)$ and communicate them to the server. This process does not increase communication overhead, as it requires transmitting only a single float per iteration. Equipped with this switching rule, we use EF14 (Seide et al., 2014) type updates to limit the communication overhead of sub-gradients from workers to server. Furthermore, we additionally enhance Safe-EF with server to workers compression using a "primal" EF21 variant, EF21-P, due to Gruntkowska et al. (2023), which compresses the difference between two estimates of the model parameters $w^{t+1}$ and $x^t$.[2]

## 4.3. Convergence Upper Bound

In our next theorem, we provide the convergence guarantees for Safe-EF summarized in Algorithm 1. The set $\mathcal{B}$ denotes all iteration counters when the constraint violation is below the threshold $c$, i.e., $\mathcal{B} := \{t \in [T-1] \mid g(x^t) \le c\}$.

**Theorem 4.2.** *Assume Assumptions 3.1-3.2 hold, the server and workers use compressors $\mathcal{C}_0 \in \mathbb{C}(\delta_s), \{\mathcal{C}_i\}_{i=1}^n \subseteq \mathbb{C}(\delta)$.*

---

[2] In fact, it was noted by Gruntkowska et al. (2023) that a pure EF21-P used at the server level can be reformulated as EF14 on the worker level. However, we only use EF21-P formulation for algorithmic presentation and design the convergence proof using EF14 formulation.

*Then there exist a choice of stepsize $\gamma$ and threshold $c$ such that the iterates of Safe-EF with bidirectional compression satisfy*

$$\mathbb{E}\left[f(\overline{x}^T) - f(x^*)\right] \le \mathcal{O}\left(\frac{RM}{\sqrt{\delta_s \delta T}}\right), \quad and$$

$$\mathbb{E}\left[g(\overline{x}^T)\right] \le \mathcal{O}\left(\frac{RM}{\sqrt{\delta_s \delta T}}\right), \tag{10}$$

*where $\overline{x}^T := \frac{1}{|\mathcal{B}|}\sum_{t \in \mathcal{B}} x^t$.*

The proof of the theorem is detailed in Appendix D, where we also give explicit choice of $\gamma$ and $c$. Next, we discuss the obtained result in several special cases as well as the main difficulties in the convergence proof.

**Single-node training with no compression.** In the special case where $n = 1$ and $\delta_s = \delta = 1$, corresponding to the non-distributed setting without compression, (10) recovers the rates in (Nesterov et al., 2018; Lan & Zhou, 2020).

**No constraints, i.e., $g \equiv 0$, and $\mathcal{C}_0 \equiv \mathbf{Id}$.** In this case, our algorithm, Safe-EF, simplifies to the well-known EF14 method (Seide et al., 2014). EF14 was previously analyzed in the non-smooth setting for single-node training ($n = 1$) by Karimireddy et al. (2019). Theorem 4.2 extends the analysis to the distributed setup. Notably, the convergence rate is consistent with that presented in their work in this special case.

**Unidirectional compression.** Next, we consider the setting with unidirectional compression, i.e., $\delta_s = 1$ and $\mathcal{C}_0 \equiv \mathbf{Id}$. We observe that both the functional suboptimality gap and constraint violation diminish at a rate of $\mathcal{O}(1/\sqrt{\delta T})$, consistent with the lower bound established in Theorem 4.1, thereby confirming the optimality of Safe-EF assuming $\delta_s$ is a numerical constant independent of $d$ and $K$.

**Bidirectional compression.** Now we discuss the setting when the compression is applied in both directions. It is worth noting that most prior studies focus on a more restricted class of compressors, such as absolute compressors (Tang et al., 2019) or unbiased compressors (Philippenko & Dieuleveut, 2021; Gruntkowska et al., 2023; 2024; Tyurin & Richtarik, 2023), in the bidirectional setting. In contrast, our work does not impose any additional constraints on the compressors. Other related work considers only server to worker compression (Sokolov & Richtárik, 2024), while often compression in both directions is important. The convergence rate in (10) highlights a slowdown by a factor of $\sqrt{\delta_s \delta}$, which aligns with similar dependencies observed in prior works on smooth distributed optimization (Fatkhullin et al., 2021). It remains an open question whether the dependence on the compression levels $\delta$ and $\delta_s$ can be improved in the non-smooth setting. Perhaps, this dependency could potentially be reduced from $\sqrt{\delta_s \delta}$ to $\sqrt{\delta} + \sqrt{\delta_s}$ by incorporating multiple communication rounds per iteration, similar

to the approach in (Huang et al., 2022). However, this procedure can be impractical since $\lceil K/\delta_s \rceil$ coordinates are communicated at every iteration as observed in (Fatkhullin et al., 2024), and we leave the study of this strategy for future work.

**Key theoretical challenges.** We emphasize that controlling constraints significantly complicates the analysis compared to prior work (Karimireddy et al., 2019), which is limited to the unconstrained, unidirectional, non-distributed setting. A key novelty of our analysis lies in demonstrating that an appropriate choice of the stepsize $\gamma$ and threshold $c$ ensures that the number of iteration counters in $\mathcal{B}$ with constraint violations below $c$ is sufficiently large to guarantee progress in reducing functional suboptimality. In particular, it is not empty and thus $\overline{x}^T$ is well-defined.

**Communication complexity with Top-$K$.** In unidirectional case with $\mathcal{C}_i$ is Top-$K$ and $\mathcal{C}_0 \equiv \mathbf{Id}$, the total communication complexity is[3]

$$\underbrace{K}_{\text{floats per iteration}} \times \underbrace{\frac{R^2 M^2}{\delta \varepsilon^2}}_{\text{\# iterations}} \leq \frac{K R^2 M^2}{\frac{K}{d} \varepsilon^2} = \frac{d R^2 M^2}{\varepsilon^2}, \quad (11)$$

where we utilize the condition $\delta \geq \frac{K}{d}$ for Top-$K$. This finding indicates that the communication complexity of Safe-EF aligns with that of parallel switching subgradient method (Safe-EF without compression) in the worst-case scenario. However, an improvement is possible when $\delta > \frac{K}{d}$, which occurs if the entries differ substantially in magnitude (Beznosikov et al., 2023).

**Key Steps of the Proof.** Our convergence proof builds on the "virtual iterates" construction of Stich & Karimireddy (2019) (see Equation (22)). In Lemma D.1, we then derive a unified bound controlling both the function sub-optimality and the constraint violation. Crucially, by enforcing appropriate choices of the step size $\gamma$ and threshold $c$, we show that this bound can be made small enough. The same lemma also guarantees that after $T$ iterations, either the number of approximately feasible points are at least $|\mathcal{B}| \geq \frac{T}{2}$ or the sub-optimality is already below the desired tolerance. Together with the preliminary lemma on the virtual iterates, this yields our full convergence theorem for Safe-EF. Finally, in Corollary D.4 we verify that the stipulated conditions on $\gamma$ and $c$ are indeed feasible.

# 5. Extension to Stochastic Setting

In this section, we consider a stochastic formulation of our the problem (1), (4), namely,

$$f_i(x) \coloneqq \mathbb{E}_{\xi^i \sim \mathcal{D}_i} \left[ f_i(x, \xi^i) \right], \quad (12)$$

---

[3]We omit the numerical constants and logarithmic factors in comparison.

and

$$g_i(x) \coloneqq \mathbb{E}_{\xi^i \sim \mathcal{D}_i} \left[ g_i(x, \xi^i) \right], \quad (13)$$

where $\mathcal{D}_i$ is a distribution of local environment (dataset) at worker $i \in [n]$. We assume that the noise follows a sub-Gaussian distribution.

**Assumption 5.1.** Workers have access to $M$-bounded stochastic subgradients and $\sigma_{\text{fv}}^2/N_{\text{fv}}$-sub-Gaussian function evaluations of $g_i$, namely, for some $M, \sigma_{\text{fv}}^2/N_{\text{fv}} > 0$, any $x \in \mathbb{R}^d$, and any $i \in [n]$ we have

$$\|f_i'(x, \xi^i)\|^2, \|g_i'(x, \xi^i)\|^2 \leq M^2, \quad (14)$$

$$\mathbb{E}\left[ \exp\left( \frac{(g_i(x, \xi^i) - g_i(x))^2}{\sigma_{\text{fv}}^2/N_{\text{fv}}} \right) \right] \leq \exp(1), \quad (15)$$

where $\xi^i$ is a sample from the local dataset $\mathcal{D}_i$. The latter assumption on sub-Gaussian function evaluation can be satisfied by implemented a mini-batch estimation of the constraints with batch-size $N_{\text{fv}}$. Moreover, we assume that the workers compute subgradients and function evaluations independently for any given $x$.

**Assumption 5.2.** We assume that for all $i \in [n]$ and for all $\xi^i \in \mathcal{D}_i$ the functions $f_i(x, \xi^i)$ and $g_i(x, \xi^i)$ are convex, i.e. for all $x, y \in \mathbb{R}^d$ we have

$$f_i(y, \xi^i) \geq f_i(x, \xi^i) + \langle f_i'(x, \xi^i), y - x \rangle, \quad (16)$$

$$g_i(y, \xi^i) \geq g_i(x, \xi^i) + \langle g_i'(x, \xi^i), y - x \rangle, \quad (17)$$

for all $f_i'(x, \xi^i) \in \partial f_i(x, \xi^i)$ and $g_i'(x, \xi^i) \in \partial g_i(x, \xi^i)$.

*Remark* 5.3. We highlight that in the special (semi-stochastic) case when subgradient evaluations $f_i'(x, \xi^i)$, $g_i'(x, \xi^i)$ are stochastic, but the constraint evaluation of $g_i$ is deterministic, the proof significantly simplifies, and convergence analysis can be repeated as in Appendix D. However, the stochastic estimation of constraint violation $g(x)$ poses a significant challenge and we need to use advanced techniques to conduct high probability analysis.

**Theorem 5.4.** *Let $\beta \in (0, 1/2)$ be a failure probability and $R \geq \|x^0 - x^*\|$. Assume workers use deterministic compressors $\{\mathcal{C}_i\}_{i=1}^n \subseteq \mathbb{C}(\delta)$. Then there exists a choice of stepsize $\gamma$, threshold $c$, and large enough batch-size $N_{\text{fv}} \geq \widetilde{\mathcal{O}}(\frac{\sigma_{\text{fv}}^2}{nc^2})$ such that the iterates of Safe-EF with unidirectional compression satisfy with probability at least $1 - 2\beta$*

$$f(\overline{x}^T) - f(x^*) \leq \mathcal{O}\left( \frac{(MR + \frac{\sigma_{\text{fv}}}{\sqrt{N_{\text{fv}}}})(1 + \log \frac{1}{\beta})}{\sqrt{\delta T}} \right),$$

$$g(\overline{x}^T) \leq \mathcal{O}\left( \frac{(MR + \frac{\sigma_{\text{fv}}}{\sqrt{N_{\text{fv}}}})(1 + \log \frac{1}{\beta})}{\sqrt{\delta T}} \right). \quad (18)$$

To achieve $\varepsilon$-accuracy, i.e., $f(\overline{x}^T) - f(x^*), g(\overline{x}^T) \leq \varepsilon$, Safe-EF requires a batch-size of order $\widetilde{\mathcal{O}}\left( \sigma_{\text{fv}}^2/n\varepsilon^2 \right)$. The convergence rate matches the lower bound (9) up to numerical and logarithmic factors. The proof is deferred to Appendix F. One of the key technical challenges of the above result is that the analysis in the prior (non-distributed) work (Lan &

Zhou, 2020) relies on bounded domain assumption, while the iterates of our algorithm can be potentially unbounded. To address this issue we use the ideas from (Liu et al., 2023) to establish a strong high probability convergence.

*Remark* 5.5. While the iteration (and communication) complexity of the method in the stochastic setting matches the lower bound up to numerical and logarithmic factors, its sample complexity is suboptimal. Taking into account the necessity of $\widetilde{\mathcal{O}}(\frac{1}{\varepsilon^2})$ batch-size, the sample complexity of the method becomes $\widetilde{\mathcal{O}}(\frac{1}{\varepsilon^4})$. Nevertheless, this complexity is no worse than the one given by non-distributed gradient switching method (Lan & Zhou, 2020). We use a different technique to conduct high probability analysis than Lan & Zhou (2020) because their analysis crucially relies on bounded diameter assumption, which we do not have in our formulation.

*Remark* 5.6. We emphasize that the proof in the stochastic unidirectional setting can be advanced to the bidirectional setting following the derivations of Theorem 4.2 and Theorem 5.4. The convergence guarantees in the stochastic bidirectional setting matches that in the deterministic up to numerical and logarithmic factors.

# 6. Experiments

Now we test Safe-EF in practice. Below we provide experiments on a simple problem with synthetic data which satisfies all our assumptions, and later test our approach in more challenging task of training the Humanoid Robot. We include additional experiments on the classical Cartpole problem and Neyman-Pearson classification in Appendix H.

## 6.1. Synthetic Data

We begin with a simple empirical setup designed to easily verify that all assumptions of Safe-EF are satisfied. Specifically, we consider the unconstrained problem of the form (1), where $f_i = \|\mathbf{A}_i x - b_i\|_1$. For this objective, the subgradient $f_i'(x) = \mathbf{A}_i^\top \mathrm{sign}(\mathbf{A}_i x - b_i)$ (Beck, 2017). This choice ensures that all assumptions required for Safe-EF hold. The data $\{\mathbf{A}_i, b_i\}_{i=1}^n \subseteq \mathbb{R}^{d \times d} \times \mathbb{R}^d$ is synthetically generated, where the parameter $s$ controls the variability across local datasets: smaller values of $s$ result in matrices $\mathbf{A}_i$ that are more similar to each other. We set $n = 10, d = 1000$, and use the Top-$K$ compressor with $K = \frac{d}{10}$ for all algorithms tested. Details of the data generation process can be found in Appendix I. We compare the proposed Safe-EF with CGD, EF21, EF21M (Fatkhullin et al., 2024), and EControl (Gao et al., 2024). For each method, hyper-parameters are tuned (see Appendix I for details) based on function value after $T = 1000$ iterations, and performance with the optimal parameters is shown in Figure 2. Our results indicate that for $s \in \{0.1, 1.0\}$, Safe-EF converges faster than all other algorithms. When heterogeneity is large, $s = 10.0$, EControl is

initially faster; however, Safe-EF catches up with EControl by the end of the training.

## 6.2. Policy Gradients for Humanoid Robot Fleet

In this suite of experiments, we demonstrate an application of Safe-EF for reinforcement learning. In this setup, each worker represents a humanoid robot that collects noisy measurements of some utility and constraint functions, to solve a *constrained Markov decision process* (Altman, 1999, CMDP).

**Constrained Markov decision processes.** We define a CMDP as the tuple $(\mathcal{S}, \mathcal{A}, r, c, p, \gamma, \rho)$, where $\mathcal{S}$ describes a state space (e.g. joint positions and velocities) and $\mathcal{A}$ describes a set of admissible actions (e.g. motor torques). The function $r : \mathcal{S} \times \mathcal{A} \to \mathbb{R}$ describes a reward function that is ought to be maximized, while $c : \mathcal{S} \times \mathcal{A} \to \mathbb{R}$ is a cost signal that must remain bounded. The system dynamics, $p$, describes a probability distribution over the next state, given a state $s \in \mathcal{S}$ and action $a \in \mathcal{A}$. States are initially drawn from the distribution $\rho$, and $\gamma$ denotes a discounting factor. In what follows, each robot-worker interacts with a separate CMDP, such that CMDPs differ only in their dynamics, i.e., each robot collects trajectories from a slightly perturbed $p_i$, relative to the nominal model $p$. Collecting trajectories entails carrying out actions determined by a *policy* $\pi(a \mid s)$, a stochastic mapping from states to actions. The objective and constraint for *each* CMDP are defined as $J_r^i(\pi) := \mathbb{E}_{\pi, p_i}[\sum_{t=0}^\infty \gamma^t r(s_t, a_t)]$ and $J_c^i(\pi) := \mathbb{E}_{\pi, p_i}[\sum_{t=0}^\infty \gamma^t c(s_t, a_t)]$ where the expectations are w.r.t. $p_i$, $\rho$ and $\pi_x$, a policy parameterized by $x \in \mathbb{R}^d$.

**Policy gradient.** A common approach for policy search is via the class of *policy gradient* algorithms (Sutton et al., 1999; Schulman et al., 2017). In essence, policy gradient algorithms use Monte Carlo sampling to obtain stochastic gradient estimates of $x$ w.r.t. the objective and constraints by "rolling out" the policy and measuring the returned rewards and costs along several trajectories. In our experiments, each worker collects data independently to obtain these estimates, which are then used to compute the PPO (Schulman et al., 2017) loss

$$f_i(x) = \mathbb{E}_{s,a \sim \bar{\pi}} \left[ \min \left\{ \frac{\pi_x(a \mid s)}{\bar{\pi}(a \mid s)} A_{p_i}^{\bar{\pi}}(s, a), \right. \right.$$
$$\left. \left. \mathrm{clip}\left( \frac{\pi_x(a \mid s)}{\bar{\pi}(a \mid s)}, 1 - \tilde{\epsilon}, 1 + \tilde{\epsilon} \right) A_{p_i}^{\bar{\pi}}(s, a) \right\} \right],$$

where, $A_{p_i}^{\bar{\pi}}$ denotes the advantage (Schulman et al., 2015) in terms of cumulative rewards, for picking an action compared to expected action of $\pi_x$, $\bar{\pi}$ is the policy with which the trajectory data was drawn and $\tilde{\epsilon}$ is a hyperparameter. Similarly, a surrogate for the constraint $g_i(x)$ is given by replacing rewards with costs when computing the advantage.

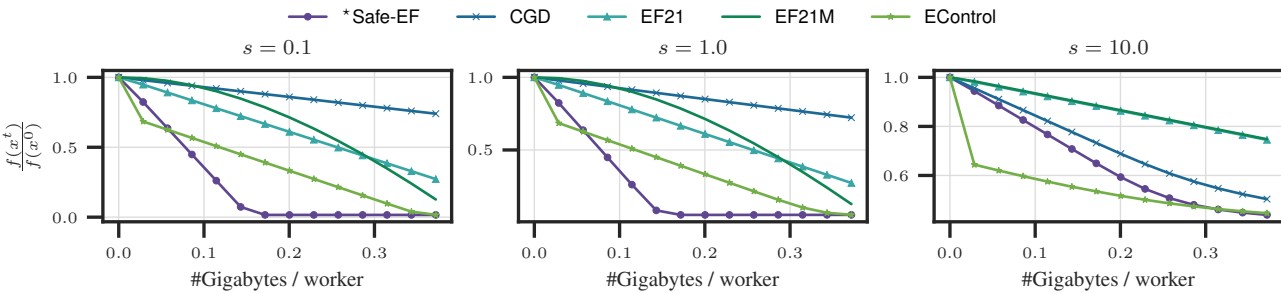

Figure 2: Comparison of Safe-EF against CGD, EF21, EF21M, and EControl on synthetic non-smooth problem. *Safe-EF coincides with EF14 (Seide et al., 2014) in this problem.

Crucially, both $f_i$ and $g_i$ *are non-smooth functions* due to $\text{clip}(x,l,u) := \max\{l, \min\{x, u\}\}$.

**Setup.** Unless specified otherwise, in all our experiments, the default number of workers is $n = 16$, compression ratio is $K/d = 0.1$ with Top-$K$ compression. We parameterize a neural network policy with $d = 0.2\text{M}$ parameters and use a batch size $N_{\text{fv}} = 1024$ to evaluate $f_i$ and $g_i$. Moreover, we treat the NN parameters as a single "flat" vector when compressing, rather than performing layer-wise compression. We run all our experiments for 5 random seed initializations and report the median and a $68\%$ confidence interval when applicable. Empirical estimates of the objective and constraint are denoted as $\hat{J}_r$ and $\hat{J}_c$ respectively. We use a batch of 128 trajectories to obtain these estimates. Further details, regarding the perturbations of models, the reward and cost functions and additional experiments are provided in Appendices H and I.

**Experiment 1: Robust compression.** We evaluate Safe-EF with Top-$K$ and Rand-$K$ sparsifiers and compare it with a constrained version of CGD with a Top-$K$ sparsifier. To adapt CGD to enforce the constraint, we follow the same approach as Safe-EF and use the switching subgradient method. Figure 3 shows the amount of communication (in gigabytes per worker) required to reach a fixed performance of $\hat{J}_r = 7500$ as the compression ratio $K/d$ increases. As illustrated, both Top-$K$ and Rand-$K$ significantly reduce communication costs compared to CGD, with Top-$K$ demonstrating the most robust performance across varying compression rates with about $2000\times$ improvement in communication reduction!

**Experiment 2: Safety.** We study the performance of Safe-EF in terms of constraint satisfaction and compare it against the unsafe error feedback algorithms EF14 (Seide et al., 2014) and EF21 (Richtárik et al., 2021). Additionally, we compare Safe-EF against a parallel variant of CRPO (Xu et al., 2021), a CMDP solver that enforces constraints via the subgradient switching method. Our parallel variant of it, indicated as Parallel-CRPO, operates independently on each worker and transmits parameters $x$ to the server without

compression. The results are presented in Figure 4, where Safe-EF satisfies the constraints with a slight performance reduction, while EF14 violates the constraint. EF21 diverges, possibly due to non-smoothness of the objective and constraint. Next, given the same communication budget in gigabytes per worker, Parallel-CRPO fails to converge. This outcome highlights the non-trivial nature of the task, emphasizing that optimal policies in the unconstrained case are insufficient to meet the constraints.

**Experiment 3: Number of workers.** We analyze the performance of Safe-EF under varying number of available workers and present our findings in Figure 5. Our results reveal two key observations. First, the convergence rate decreases significantly when the number of workers is very small. Second, beyond a certain threshold, increasing the number of workers yields diminishing performance gains. The latter aligns with our theoretical lower bounds in Theorem E.2, which establish that no improvement in $n$ is possible in the worst case.

**Experiment 4: Effect of batch-size.** Theorem 5.4 has a certain requirement of sufficiently large batch-size $N_{\text{fv}}$ due to constraint estimation process. If this requirement is met, the convergence rate is improved when increasing $N_{\text{fv}}$ until it reaches the lower bound in Theorem 4.1. To study this effect in practice, we vary the batch size $N_{\text{fv}} \in \{256, 512, 1024, 2048, 4096\}$. Our results in Figure 6, indicate that by increasing the batch size from $N_{\text{fv}} = 1024$ to $2048$, we can see the improvement, however, a further increase from $N_{\text{fv}} = 2048$ to $N_{\text{fv}} = 4096$ does not yield more improvement. For smaller batch sizes $N_{\text{fv}} \in \{256, 512\}$, Safe-EF did not converge, resulting in non-numeric values, and therefore are not presented in Figure 6. These findings are in line with our large-batch requirement in Theorem 5.4 and highlight the need to design algorithms that are robust to smaller batch sizes—suggesting an important direction for future work.

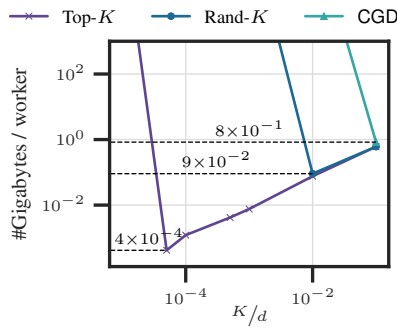
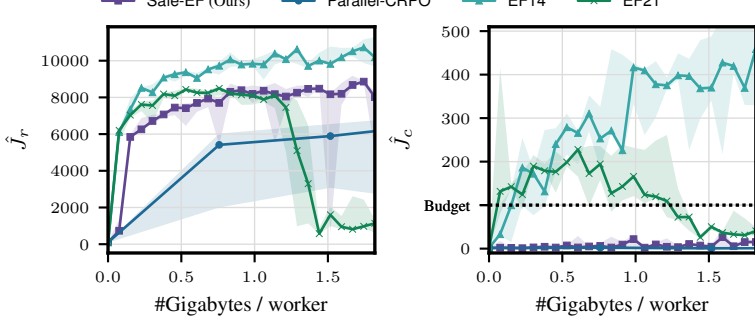

Figure 3: Gigabytes required to reach a fixed benchmark performance for different compression ratios. Top-$K$ can achieve the same performance as CGD, but with approximately two orders of magnitude less gigabytes.

Figure 4: Objective and constraint during learning. Budget denotes the level below which $J_c$ must remain to satisfy the constraint. EF14, an unsafe baseline, fails to satisfy the constraint. EF21, another unsafe baseline designed for smooth problems, diverges. Parallel-CRPO, a safe baseline without compression, suffers from communication overhead. In contrast, Safe-EF ensures constraint satisfaction with minimal performance loss.

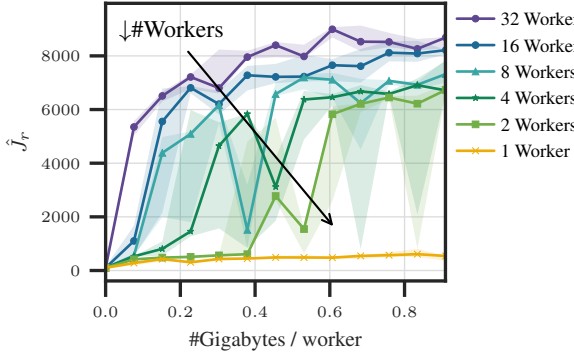
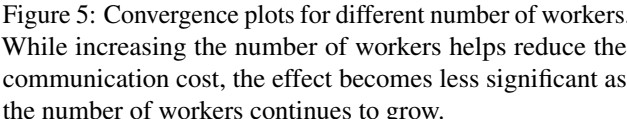

Figure 5: Convergence plots for different number of workers. While increasing the number of workers helps reduce the communication cost, the effect becomes less significant as the number of workers continues to grow.

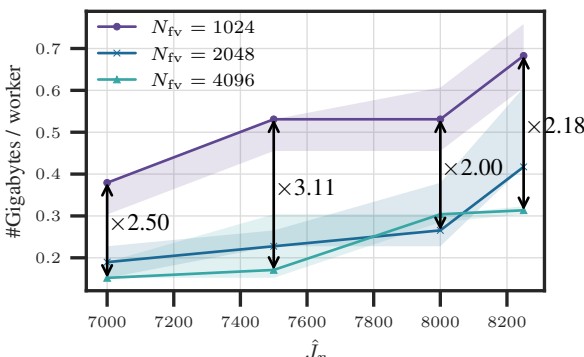

Figure 6: Communication required to reach a desired performance level for different batch samples $N_{\mathrm{fv}}$. Beyond a certain batch size, improvement diminishes.

## 7. Limitations and Future Work

While we make significant progress in understanding non-smooth EF, there are certain limitations in our work. First, we assume all functions are convex, while Safe-EF seems to excel even in challenging, highly non-convex RL tasks. Thus, it is crucial to understand non-convex problems: in a general setting (e.g. Boob et al., 2023; Jia & Grimmer, 2022; Grimmer & Jia, 2025) as well as in structured RL problems (e.g. Agarwal et al., 2021; Xu et al., 2021; Lan, 2023; Fatkhullin et al., 2023a; Barakat et al., 2023; Islamov et al., 2024a). Second, our noise assumptions are relatively stringent, and can be potentially relaxed using gradient clipping (Nazin et al., 2019; Gorbunov et al., 2024) or normalization (Hübler et al., 2024) techniques, although this is non-trivial due to constraint estimation. Finally, our algorithm requires large batch-sizes and is not sample efficient in the stochastic setting due to constraint estimation,

and our experiments indicate it is likely an issue with the algorithm. Primal-dual approaches (Juditsky et al., 2011; Boob et al., 2023) can help mitigate this limitation.

## Acknowledgements

Rustem Islamov acknowledges the financial support of the Swiss National Foundation, SNF grant No 207392. Yarden As is supported by the grant of the Hasler foundation (grant no. 21039). Yarden As and Ilyas Fatkhullin are funded by ETH AI Center.

## Impact Statement

This paper presents work whose goal is to advance the field of Machine Learning. There are many potential societal consequences of our work, none of which we feel must be specifically highlighted here.

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

# A. Additional Related Work

The Error Feedback (EF) mechanism was initially studied in the single-node setting ($n = 1$) by Stich et al. (2018); Alistarh et al. (2017). Subsequent research extended its analysis to the smooth convex setting, incorporating additional unbiased compressors (Gorbunov et al., 2020; Stich, 2020; Qian et al., 2021b). The EF21 algorithm, introduced by Richtárik et al. (2021), was the first to establish provable convergence in the large-batch smooth regime without data heterogeneity bounds. Later, Fatkhullin et al. (2024) removed this large-batch requirement by integrating a momentum mechanism into the EF21 framework, achieving an optimal asymptotic rate. An extension of EF14, called EControl, was proposed by Gao et al. (2024), demonstrating convergence in both smooth convex and non-convex settings while attaining optimal asymptotic complexity. Recent research has further advanced the analysis of EF, extending it to variational inequalities (Beznosikov et al., 2022), decentralized communication graphs (Koloskova et al., 2020; Singh et al., 2021; Zhao et al., 2022; Islamov et al., 2024b), local updates (Huang et al., 2023), bilevel optimization (He et al., 2024), differentially private training (Shulgin et al., 2025; Islamov et al., 2025), and reinforcement learning (Mitra et al., 2023; Adibi et al., 2024; Beikmohammadi et al., 2024). Additionally, Richtárik et al. (2022); Makarenko et al. (2022); Islamov et al. (2023) expanded EF analysis to a broader class of 3PC compression operators, encompassing contractive compressors as a special case. Recent works analyzed the EF mechanism as a special case of biased gradient descent in the single-node setting (Ajalloeian & Stich, 2020; Demidovich et al., 2023) while Richtárik et al. (2024) improved the constant dependencies in the rate of EF21.

EF21 variant of EF has been analyzed in the context of $(L_0, L_1)$-smooth optimization (Khirirat et al., 2024), which is different from our non-smoothness since $(L_0, L_1)$-smoothness implies smoothness on any compact set and failure examples as in Example 1.2 cannot happen under such assumption. On the other hand, if not limited to a compact set, the gradients under $(L_0, L_1)$-smoothness can grow when $\|x\| \to \infty$.

# B. Failure of CGD and EF21 in Non-smooth Convex Setting

**Proof of Example 1.1. Non-convergence of CGD.**

*Proof.* Consider a 2-dimensional problem $f_i(x) = \|x\|_1$, $f(x) = \frac{1}{n}\sum_{i=1}^n f_i(x)$ with $f(x_*) = 0$. Set the initial vectors $x^0 = (\gamma/2, -1)^\top$ and consider CGD (2) with Top-1 compressor.

The proof for the case when $\gamma = 0$ is trivial. We consider the case when $\gamma > 0$. In this case, the function is differentiable at every point of its trajectory, and for any $t \geq 0$ it holds that

$$x^t = \begin{pmatrix} \frac{\gamma(-1)^t}{2} \\ -1 \end{pmatrix}, \qquad \partial f_i(x^t) = \left\{ \begin{pmatrix} (-1)^t \\ -1 \end{pmatrix} \right\}.$$

The base of induction ($t = 0$) is trivial. For the induction step, we make the calculation

$$x^{t+1} = x^t - \gamma g^t = \begin{pmatrix} \frac{\gamma(-1)^t}{2} \\ -1 \end{pmatrix} - \gamma \operatorname{Top-1}\begin{pmatrix} (-1)^t \\ 1 \end{pmatrix} = \begin{pmatrix} \frac{\gamma(-1)^{t+1}}{2} \\ -1 \end{pmatrix},$$

where in the last step, Top-1 operator always selects the first coordinate since the entries are equal in absolute value. It remains to compute the function value at these iterates $f(x^t)$ to conclude the proof. $\qquad\square$

We remark that divergence issues of gradient methods using biased compressors were previously raised in (Karimireddy et al., 2019). However, their examples only apply to Sign operator, while we are mainly interested in the behavior of Top-$K$ compressor for distributed optimization. Thus, a different construction is required to capture the interplay of Top-$K$ compressor with non-smoothness of $f$. Another divergence example using Top-$K$ is shown by Beznosikov et al. (2023), however, their example is smooth, strongly convex and the key effect is different, since their divergence happens due to heterogeneity. Finally, Fatkhullin et al. (2024) show an example of divergence of EF21 in the stochastic setting, which is also different since their function is smooth, strongly convex and the divergence occurs due to noise.

**Proof of Example 1.2. Divergence of EF21.**

*Proof.* Similarly to the proof of Example 1.1, we consider a 2-dimensional problem $f_i(x) = \|x\|_1$, $f(x) = \frac{1}{n}\sum_{i=1}^n f_i(x)$ with $f(x_*) = 0$. Set the initial vectors $x^0 = (\gamma/2, -1)^\top$, $v_i^0 = (1,1)^\top$, and consider EF21 (3) with Top-1 compressor.

The proof for the case when $\gamma = 0$ is trivial. We consider the case when $\gamma > 0$. In this case the function is differentiable at every point of its trajectory and for any $t \geq 0$ it holds that

$$x^t = \begin{pmatrix} \frac{\gamma(-1)^t}{2} \\ -1 - t\gamma \end{pmatrix}, \qquad \partial f_i(x^t) = \left\{ \begin{pmatrix} (-1)^t \\ -1 \end{pmatrix} \right\}, \qquad v_i^t = \begin{pmatrix} (-1)^t \\ 1 \end{pmatrix}.$$

The base of induction ($t = 0$) is trivial. For the induction step, we make the calculation

$$x^{t+1} = x^t - \gamma v^t = \begin{pmatrix} \frac{\gamma(-1)^t}{2} \\ -1 - t\gamma \end{pmatrix} - \gamma \begin{pmatrix} (-1)^t \\ 1 \end{pmatrix} = \begin{pmatrix} \frac{\gamma(-1)^{t+1}}{2} \\ -1 - (t+1)\gamma \end{pmatrix},$$

$$v^{t+1} = v_i^{t+1} = \begin{pmatrix} (-1)^t \\ 1 \end{pmatrix} + \operatorname{Top-1}\left( \begin{pmatrix} (-1)^{t+1} \\ -1 \end{pmatrix} - \begin{pmatrix} (-1)^t \\ 1 \end{pmatrix} \right) = \begin{pmatrix} (-1)^{t+1} \\ 1 \end{pmatrix},$$

where in the last step, Top-1 operator selects the first coordinate since the entries are equal in absolute value. It remains to compute the function value at these iterates $f(x^t)$ to conclude the proof. $\qquad\square$

## C. Convergence Upper Bound for EF21 in Smooth Convex Setting

In this section, we consider EF21 method with projection

$$\text{Projected-EF21} \qquad x^{t+1} = \Pi_{\mathcal{X}}(x^t - \gamma\, v^t), \qquad v^t = \frac{1}{n}\sum_{i=1}^{n} v_i^t, \tag{19}$$

$$v_i^{t+1} = v_i^t + \mathcal{C}(\nabla f_i(x^{t+1}) - v_i^t).$$

where $\Pi_{\mathcal{X}}$ is a projection operator on a convex set $\mathcal{X}$. This method was proposed and analyzed earlier in (Fatkhullin et al., 2021) for non-convex smooth problems. In Example 1.2, we showed that this algorithm is not suitable for non-smooth optimization because it diverges even in a simple convex example like $\|x\|_1$. While this algorithm was extensively studied for smooth non-convex problems, we are not aware of any convergence results for this algorithm under convexity (with convergence in the function value). To close this gap and complement the failure example of this method in Example 1.2 in non-smooth convex case, we provide the convergence result for this method in smooth convex setting.

**Theorem C.1.** *Let each $f_i(\cdot)$ be differentiable and $L_i$-smooth on $\mathcal{X}$ for all $i = 1, \ldots, n$, i.e., $\|\nabla f_i(x) - \nabla f_i(y)\| \le L_i \|x - y\|$ for all $x, y \in \mathcal{X}$, and let $f(\cdot)$ be convex over a convex compact set $\mathcal{X} \subseteq \mathbb{R}^d$ with diameter $R_{\mathcal{X}}$. Then for any $T \ge 1$* Projected-EF21 *with stepsize $\gamma \le \frac{\delta}{2\sqrt{6}L}$ satisfies*

$$\mathbb{E}\left[f(x^T) - f(x^*)\right] \le \frac{R_{\mathcal{X}}^2}{\gamma T}\left(1 + \log\left(\frac{\gamma \Lambda_0 T}{R_{\mathcal{X}}^2}\right)\right),$$

*where $\Lambda_0 := f(x^0) - f(x^*) + \frac{1}{\sqrt{6}L}\|g^0 - \nabla f(x^0)\|^2$, and $L := \sqrt{\frac{1}{n}\sum_{i=1}^{n} L_i^2}$.*

*Remark* C.2. The current stepsize restriction is $\gamma \le \frac{\delta}{2\sqrt{6}L}$, where $L$ is the quadratic mean of the smoothness constants $L_i$. This restriction can be further improved by following the results in Richtárik et al. (2024), which requires weighting workers' contributions by non-uniform constants. This leads to the improved step-size (and eventually improved rate) of the form $\gamma \le \mathcal{O}(1/\overline{L})$, where $\overline{L} = \frac{1}{n}\sum_{i=1}^{n} L_i$, since $\overline{L} \le L$ always. However, setting the weights requires access to the smoothness constants $\{L_i\}_{i=1}^{n}$, which might be challenging in practice.

Before we move to the proof of this result, a few comments are in order. First, if we set $\gamma = \frac{\delta}{2\sqrt{6}L}$, this theorem implies $\widetilde{\mathcal{O}}\left(\frac{LR_{\mathcal{X}}^2}{\delta T}\right)$ convergence rate for Projected-EF21, where $\widetilde{\mathcal{O}}$ hides numerical constants and a logarithmic term. This convergence rate recovers (up to a logarithmic factor) the rate of subgradient descent when $\delta = 1$ (no compression), and is $1/\delta$ times worse in the presence of compression. This is consistent with rates in non-convex and strongly convex settings (Richtárik et al., 2021; Fatkhullin et al., 2021). We believe the logarithmic factor can be removed by a more careful choice of parameter $\lambda$ in the proof below. Second, the compactness of the set $\mathcal{X}$ is critical in the analysis of the method, it would be interesting to explore if this requirement can be removed. Finally, the extension of this method to stochastic setting is possible by replacing $\nabla f_i(x^{t+1})$ with a large mini-batch or momentum estimator, however, a batch-free version of this method may not converge due to a counter-example in (Fatkhullin et al., 2024).

*Proof.* Since each $f_i$ is $L_i$-smooth, it follows that $f(x) = \frac{1}{n}\sum_{i=1}^{n} f_i(x)$ is $L$-smooth with $L = \sqrt{\frac{1}{n}\sum_{i=1}^{n} L_i^2}$. Next, we follow the proof technique similar to Theorem 8 in (Fatkhullin et al., 2023b). By smoothness of $f$, we have for any $z \in \mathcal{X}$

$$
\begin{aligned}
f\left(x^{t+1}\right) &\le f\left(x^t\right) + \left\langle \nabla f\left(x^t\right), x^{t+1} - x^t \right\rangle + \frac{L}{2}\left\|x^{t+1} - x^t\right\|^2 \\
&= f\left(x^t\right) + \left\langle v^t, x^{t+1} - x^t \right\rangle + \frac{1}{2\gamma}\left\|x^{t+1} - x^t\right\|^2 + \left\langle \nabla f\left(x^t\right) - v^t, x^{t+1} - x^t \right\rangle - \left(\frac{L}{2} - \frac{1}{2\gamma}\right)\left\|x^{t+1} - x^t\right\|^2 \\
&\le f\left(x^t\right) + \frac{1}{2\gamma}\left\|x^t - z\right\|^2 - \frac{1}{2\gamma}\left\|x^{t+1} - z\right\|^2 + \left\langle v^t, z - x^t \right\rangle \\
&\qquad + \left\langle \nabla f\left(x^t\right) - v^t, x^{t+1} - x^t \right\rangle - \left(\frac{L}{2} - \frac{1}{2\gamma}\right)\left\|x^{t+1} - x^t\right\|^2 =: (*),
\end{aligned}
$$

where the last inequality follows by the update rule of the algorithm. Next, rearranging we get

$$
\begin{aligned}
(*) &= f\left(x^t\right) + \frac{1}{2\gamma}\left\|x^t - z\right\|^2 - \frac{1}{2\gamma}\left\|x^{t+1} - z\right\|^2 + \left\langle \nabla f\left(x^t\right), z - x^t \right\rangle \\
&\quad + \left\langle \nabla f\left(x^t\right) - v^t, x^{t+1} - z \right\rangle - \left(\frac{L}{2} - \frac{1}{2\gamma}\right)\left\|x^{t+1} - x^t\right\|^2 \\
&\leq f\left(x^t\right) + \frac{1}{2\gamma}\left\|x^t - z\right\|^2 - \frac{1}{2\gamma}\left\|x^{t+1} - z\right\|^2 + \left\langle \nabla f\left(x^t\right), z - x^t \right\rangle \\
&\quad + \frac{\gamma}{2}\left\|v^t - \nabla f\left(x^t\right)\right\|^2 + \frac{1}{2\gamma}\left\|x^{t+1} - z\right\|^2 - \left(\frac{L}{2} - \frac{1}{2\gamma}\right)\left\|x^{t+1} - x^t\right\|^2 \\
&= f\left(x^t\right) + \frac{1}{2\gamma}\left\|x^t - z\right\|^2 + \left\langle \nabla f\left(x^t\right), z - x^t \right\rangle + \frac{\gamma}{2}\left\|v^t - \nabla f\left(x^t\right)\right\|^2 - \left(\frac{L}{2} - \frac{1}{2\gamma}\right)\left\|x^{t+1} - x^t\right\|^2,
\end{aligned}
$$

where we used Young's inequality $\langle a, b \rangle \leq \frac{\gamma}{2}\|a\|^2 + \frac{2}{\gamma}\|b\|^2$ for any $a, b \in \mathbb{R}^d$. Using (lower curvature) smoothness of $f$, we derive

$$
\begin{aligned}
f\left(x^{t+1}\right) &\leq f(z) + \left(\frac{1}{2\gamma} + \frac{L}{2}\right)\left\|x^t - z\right\|^2 + \frac{\gamma}{2}\left\|v^t - \nabla f\left(x^t\right)\right\|^2 - \left(\frac{L}{2} - \frac{1}{2\gamma}\right)\left\|x^{t+1} - x^t\right\|^2 \\
&\leq f(z) + \frac{1}{\gamma}\left\|x^t - z\right\|^2 + \frac{\gamma}{2}\frac{1}{n}\left\|v_i^t - \nabla f_i\left(x^t\right)\right\|^2 - \left(\frac{L}{2} - \frac{1}{2\gamma}\right)\left\|x^{t+1} - x^t\right\|^2,
\end{aligned}
$$

where the last inequality holds since $\gamma \leq 1/L$. Now we fix some $\lambda \in [0, 1]$ and select $z = (1 - \lambda)x^t + \lambda x_* \in \mathcal{X}$, where $x_* \in \mathcal{X}_*$. By convexity of $f(\cdot)$, we have

$$
f(z) \leq (1 - \lambda)f(x^t) + \lambda f(x_*) - \frac{\lambda(1 - \lambda)}{2L}\|\nabla f(x^t) - \nabla f(x_*)\|^2 \leq (1 - \lambda)f(x^t) + \lambda f(x_*).
$$

Moreover, $\|x^t - z\| = \lambda\|x^t - x_x\| \leq \lambda R_{\mathcal{X}}$, where $R_{\mathcal{X}} = \max_{x, y \in \mathcal{X}}\|x - y\|$. Thus, we get for any $\lambda \in [0, 1]$

$$
f(x^{t+1}) - f(x_*) \leq (1 - \lambda)(f(x^t) - f(x_*)) + \frac{\lambda^2 R_{\mathcal{X}}^2}{\gamma} + \frac{\gamma}{2}V_t - \left(\frac{L}{2} - \frac{1}{2\gamma}\right)\|x^{t+1} - x^t\|^2. \tag{20}
$$

For a contractive compressor we have $\mathbb{E}\|\mathcal{C}(x) - x\|^2 \leq (1 - \delta)\|x\|^2$ for some $\delta \in (0, 1]$. Let $V_{t,i} := \mathbb{E}\|g_i^t - \nabla f_i(x^t)\|^2$, $V_t := \frac{1}{n}\sum_{i=1}^n V_{t,i}$. Then

$$
\begin{aligned}
V_{t+1,i} &= \mathbb{E}\|g_i^{t+1} - \nabla f_i(x^{t+1})\|^2 = \mathbb{E}\|\mathcal{C}(\nabla f_i(x^{t+1}) - g_i^t) + g_i^t - \nabla f_i(x^{t+1})\|^2 \\
&\leq (1 - \delta)\mathbb{E}\|g_i^t - \nabla f_i(x^{t+1})\|^2 \\
&\leq (1 - \delta)\left(1 + \frac{\delta}{2}\right)\mathbb{E}\|g_i^t - \nabla f_i(x^t)\|^2 + \left(1 + \frac{2}{\delta}\right)\mathbb{E}\|\nabla f_i(x^{t+1}) - \nabla f_i(x^t)\|^2 \\
&\leq \left(1 - \frac{\delta}{2}\right)V_{t,i} + \frac{3L_i^2}{\delta}\mathbb{E}\|x^{t+1} - x^t\|^2.
\end{aligned}
$$

By averaging for $i = 1, \ldots, n$, we get

$$
V_{t+1} \leq \left(1 - \frac{\delta}{2}\right)V_t + \frac{3L^2}{\delta}\mathbb{E}\|x^{t+1} - x^t\|^2. \tag{21}
$$

Define $\Delta_t := \mathbb{E}[f(x^t) - f(x_*)]$, then adding (20) $+\frac{2}{\delta}$ times (21) and taking $\gamma \leq \frac{\delta}{2\sqrt{6}L}$, we have

$$
\begin{aligned}
\Lambda_{t+1} &:= \Delta_{t+1} + \frac{2\gamma}{\delta}V_{t+1} \\
&\leq (1 - \lambda)\Delta_t + \frac{\gamma}{2}V_t + \frac{2\gamma}{\delta}\left(1 - \frac{\delta}{2}\right)V_t + \frac{\lambda^2}{\gamma}R_{\mathcal{X}}^2 - \left(\frac{L}{2} - \frac{1}{2\gamma} + \frac{3L^2 \cdot 2\gamma}{\delta}\right)\mathbb{E}\|x^{t+1} - x^t\|^2 \\
&= (1 - \lambda)\Delta_t + \frac{2\gamma}{\delta}\left(1 - \frac{\delta}{2} + \frac{\gamma}{2}\frac{\delta}{2\gamma}\right)V_t + \frac{\lambda^2}{\gamma}R_{\mathcal{X}}^2 - \left(\frac{L}{2} - \frac{1}{2\gamma} + \frac{3L^2}{\delta}\frac{2\gamma}{\delta}\right)\mathbb{E}\|x^{t+1} - x^t\|^2 \\
&\leq (1 - \lambda)\Delta_t + \frac{2\gamma}{\delta}\left(1 - \frac{\delta}{4}\right)V_t + \frac{\lambda^2}{\gamma}R_{\mathcal{X}}^2 \\
&\leq (1 - \lambda)\Lambda_t + \frac{\lambda^2}{\gamma}R_{\mathcal{X}}^2,
\end{aligned}
$$

where in the last step we assume the choice $\lambda \leq \delta/4$. Finally, we unroll the recursion for $t = 0, 1, \ldots, T-1$ and setting $\lambda = \min\left\{\frac{\delta}{4}; \frac{1}{N} \log\left(\frac{\gamma \Lambda_0 N}{R_\mathcal{X}^2}\right)\right\}$, we derive

$$
\begin{aligned}
\Lambda_T &\leq (1-\lambda)^T \Lambda_0 + \left(\sum_{t=0}^{T-1}(1-\lambda)^t\right) \frac{\lambda^2 R_\mathcal{X}^2}{\gamma} \leq (1-\lambda)^T \Lambda_0 + \frac{\lambda R_\mathcal{X}^2}{\gamma} \\
&= \exp(T\log(1-\lambda))\Lambda_0 + \frac{\lambda R_\mathcal{X}^2}{\gamma} \leq \exp\left(-\log\left(\frac{\gamma \Lambda_0 T}{R_\mathcal{X}^2}\right)\right)\Lambda_0 + \frac{\lambda R_\mathcal{X}^2}{\gamma} \\
&\leq \frac{R_\mathcal{X}^2}{\gamma T} + \frac{R_\mathcal{X}^2}{\gamma T}\log\left(\frac{\gamma \Lambda_0 T}{R_\mathcal{X}^2}\right).
\end{aligned}
$$

$\square$

## D. Convergence Upper Bound for Safe-EF with Bidirectional Compression

The analysis uses the "virtual iterates" framework, which is often used in the literature (Stich & Karimireddy, 2019; Koloskova et al., 2023; Mishchenko et al., 2023; Islamov et al., 2024c). Define the virtual iterates $\hat{x}^t := w^t - \gamma e^t$ with $\hat{x}^0 = x^0$. Note that then we have $\hat{x}^{t+1} = \hat{x}^t - \gamma h^t$. Indeed, assume that it is true at iteration $t$, then

$$\hat{x}^{t+1} = w^{t+1} - \gamma e^{t+1} = (w^t - \gamma v^t) - \gamma(e^t + h^t - v^t) = (w^t - \gamma e^t) - \gamma h^t = \hat{x}^t - \gamma h^t. \tag{22}$$

We additionally define $\hat{e}^t := w^t - x^t$, an error that appears due to down-link (server to worker) compression.

**Lemma D.1.** *For any $x \in \mathbb{R}^d$, the following inequality holds*

$$\sum_{t \in \mathcal{B}} \gamma(f(x^t) - f(x)) + \sum_{t \in \mathcal{N}} \gamma[c - g(x)] \leq \frac{1}{2}\|x^0 - x\|^2 + \frac{1}{2}\sum_{t=0}^{T-1}\gamma^2\|h^t\|^2 + \sum_{t=0}^{T-1}\gamma^2\|h^t\|\cdot\|e^t\|$$

$$+ \sum_{t=0}^{T-1}\gamma\|h^t\|\cdot\|\hat{e}^t\|$$

*Proof.* From the update rule (22), we have

$$\|\hat{x}^{t+1} - x\|^2 = \|\hat{x}^t - x\|^2 - 2\gamma\langle h^t, \hat{x}^t - x\rangle + \gamma^2\|h^t\|^2.$$

Rewriting the above, we get

$$2\gamma\langle h^t, x^t - x\rangle = \|\hat{x}^t - x\|^2 - \|\hat{x}^{t+1} - x\|^2 + \gamma^2\|h^t\|^2 + 2\gamma\langle h^t, x^t - w^t\rangle + 2\gamma\langle h^t, w^t - \hat{x}^t\rangle$$

$$\leq \|\hat{x}^t - x\|^2 - \|\hat{x}^{t+1} - x\|^2 + \gamma^2\|h^t\|^2 + 2\gamma^2\|h^t\|\|e^t\| + 2\gamma\|h^t\|\|\hat{e}^t\|.$$

Summing up both sides, we derive

$$2\sum_{t=0}^{T-1}\gamma\langle h^t, x^t - x\rangle \leq \|x^0 - x\|^2 - \|\hat{x}^T - x\|^2 + \sum_{t=0}^{T-1}\gamma^2\|h^t\|^2 + 2\sum_{t=0}^{T-1}\gamma^2\|h^t\|\cdot\|e^t\| + 2\sum_{t=0}^{T-1}\gamma\|h^t\|\cdot\|\hat{e}^t\|.$$

Dropping the non-negative term $\|\widetilde{x}^T - x\|^2$ and using $\hat{x}^0 = x^0$ we obtain

$$2\sum_{t=0}^{T-1}\gamma\langle h^t, x^t - x\rangle \leq \|x^0 - x\|^2 + \sum_{t=0}^{T-1}\gamma^2\|h^t\|^2 + 2\sum_{t=0}^{T-1}\gamma^2\|h^t\|\cdot\|e^t\| + 2\sum_{t=0}^{T-1}\gamma\|h^t\|\cdot\|\hat{e}^t\|.$$

Now we split the sum over $\mathcal{N}$ and $\mathcal{B}$. For $t \in \mathcal{B}$ we have, $h_i^t = f_i'(x^t)$, i.e. $h^t = f'(x^t)$, and for $t \in \mathcal{N}$ $h_i^t = g_i'(x^t)$, i.e. $h^t = g'(x^t)$. Therefore, armed with the convexity of $f$ and $g$ we have

$$\langle f'(x^t), x^t - x\rangle \geq f(x^t) - f(x), \quad \forall k \in \mathcal{B},$$

$$\langle g'(x^t), x^t - x\rangle \geq g(x^t) - g(x) \geq c - g(x), \quad \forall k \in \mathcal{N}.$$

Therefore, we have

$$\sum_{t \in \mathcal{B}}\gamma(f(x^t) - f(x)) + \sum_{t \in \mathcal{N}}\gamma[c - g(x)] \leq \sum_{t \in \mathcal{B}}\gamma\langle f'(x^t), x^t - x\rangle + \sum_{t \in \mathcal{N}}\gamma\langle g'(x^t), x^t - x\rangle$$

$$\leq \frac{1}{2}\|x^0 - x\|^2 + \frac{1}{2}\sum_{t=0}^{T-1}\gamma^2\|h^t\|^2 + \sum_{t=0}^{T-1}\gamma^2\|h^t\|\cdot\|e^t\| + \sum_{t=0}^{T-1}\gamma\|h^t\|\cdot\|\hat{e}^t\|.$$

$\square$

We now present the main convergence theorem, providing explicit bounds under appropriate conditions on $\gamma$ and $c$. To do so, we need to define $\overline{x}^T$ as follows

$$\overline{x}^T := \frac{1}{\sum_{t \in \mathcal{B}}\gamma}\sum_{t \in \mathcal{B}}\gamma x^t = \frac{1}{|\mathcal{B}|}\sum_{t \in \mathcal{B}}x^t. \tag{23}$$

**Lemma D.2.** *Suppose that the stepsize $\gamma$ and threshold $c$ satisfy*

$$\frac{T}{2}\gamma c > \frac{1}{2}R^2 + \frac{1}{2}M^2\gamma^2 T + M^2\gamma^2\frac{2\sqrt{1-\delta}}{\delta}T + M^2\gamma^2\frac{2\sqrt{10(1-\delta_s)}}{\delta_s\delta}T. \tag{24}$$

*Then we have*

$$\gamma\mathbb{E}\left[\sum_{t\in\mathcal{B}}f(x^t)-f(x^*)\right]+\gamma\mathbb{E}\left[\sum_{t\in\mathcal{N}}c-g(x^*)\right]\le\frac{1}{2}R^2+\frac{1}{2}M^2\gamma^2T+2M^2\gamma^2\frac{2\sqrt{1-\delta}}{\delta}T+2M^2\gamma^2\frac{\sqrt{10(1-\delta_{\mathrm{s}})}}{\delta_{\mathrm{s}}\delta}T. \quad (25)$$

*Moreover, suppose that* (25) *holds. Then $\mathcal{B}$ is non-empty, i.e. $\overline{x}^T$ is well-defined, and one of the two following conditions holds*

1. $|\mathcal{B}|\ge\frac{T}{2}$, *or*

2. $\gamma\mathbb{E}\left[\sum_{t\in\mathcal{B}}f(x^t)-f(x^*)\right]\le 0.$

*Proof.* Let us use $x=x^*$ in Lemma D.1. Taking the expectation and using the fact that $\|h^t\|\le M$, we get

$$\mathbb{E}\left[\gamma\sum_{t\in\mathcal{B}}f(x^t)-f(x^*)\right]+\mathbb{E}\left[\gamma\sum_{t\in\mathcal{N}}c-g(x^t)\right]\le\frac{1}{2}R^2+\frac{1}{2}M^2\sum_{t=0}^{T-1}\gamma^2+M\sum_{t=0}^{T-1}\gamma^2\mathbb{E}\left[\|e^t\|\right] \quad (26)$$

$$+\ M\sum_{t=0}^{T-1}\gamma\mathbb{E}\left[\|\hat{e}^t\|\right]. \quad (27)$$

Using the properties of the compressors $\{\mathcal{C}_i\}_{i=1}^n$, we get by induction[4] that (with the choice $\eta=\frac{\delta}{2(1-\delta)}$)

$$\mathbb{E}\left[\|e^{t+1}\|^2\right]=\mathbb{E}\left[\left\|\frac{1}{n}\sum_{i=1}^n e_i^{t+1}\right\|^2\right]\le\frac{1}{n}\sum_{i=1}^n\mathbb{E}\left[\|e_i^{t+1}\|^2\right]=\frac{1}{n}\sum_{i=1}^n\mathbb{E}\left[\|e_i^t+h_i^t-\mathcal{C}_i(e_i^t+h_i^t)\|^2\right]$$

$$\le\frac{1-\delta}{n}\sum_{i=1}^n\mathbb{E}\left[\|e_i^t+h_i^t\|^2\right]$$

$$\le(1-\delta)(1+\eta)\frac{1}{n}\sum_{i=1}^n\mathbb{E}\left[\|e_i^t\|^2\right]+(1-\delta)\left(1+\eta^{-1}\right)M^2$$

$$\le\sum_{l=0}^t[(1-\delta)(1+\eta)]^{t-l}(1-\delta)(1+\eta^{-1})M^2$$

$$\le\frac{(1-\delta)(1+\eta^{-1})}{1-(1-\delta)(1+\eta)}M^2=\frac{(1-\delta)(1+\eta^{-1})}{\delta-\eta(1-\delta)}M^2=\frac{2(1-\delta)(1+\eta^{-1})}{\delta}M^2\le\underbrace{\frac{4(1-\delta)}{\delta^2}}_{=:C^2}M^2.$$

Similarly, we bound $\mathbb{E}\left[\|\hat{e}^t\|^2\right]$

$$\mathbb{E}\left[\|\hat{e}^{t+1}\|^2\right]=\mathbb{E}\left[\|w^{t+1}-x^{t+1}\|^2\right]=\mathbb{E}\left[\|w^{t+1}-x^t-\mathcal{C}(w^{t+1}-x^t)\|^2\right]$$

$$\le(1-\delta_{\mathrm{s}})\mathbb{E}\left[\|w^{t+1}-x^t\|^2\right]$$

$$=(1-\delta_{\mathrm{s}})\mathbb{E}\left[\|w^t-\gamma v^t-x^t\|^2\right]=(1-\delta_{\mathrm{s}})\mathbb{E}\left[\|\hat{e}^t-\gamma v^t\|^2\right]$$

$$\le(1-\delta_{\mathrm{s}})(1+\hat{\eta})\mathbb{E}\left[\|\hat{e}^t\|^2\right]+(1-\delta_{\mathrm{s}})(1+\hat{\eta}^{-1})\gamma^2\mathbb{E}\left[\|v^t\|^2\right]. \quad (28)$$

Note that

$$\mathbb{E}\left[\left\|\frac{1}{n}\sum_{i=1}^n e_i^t+h_i^t\right\|^2\right]\le\frac{2}{n}\sum_{i=1}^n\mathbb{E}\left[\|e_i^t\|^2\right]+\mathbb{E}\left[\|h_i^t\|^2\right]$$

$$\le\frac{2}{n}\sum_{i=1}^n\left(\frac{4(1-\delta)}{\delta^2}M^2+M^2\right)=2M^2\frac{4(1-\delta)+\delta^2}{\delta^2}\le\frac{10M^2}{\delta^2}.$$

---

[4]The base of induction obviously holds since $\|e_i^0\|=0$.

Therefore,

$$
\begin{aligned}
\mathbb{E}\left[\|v^t\|^2\right] &\leq \frac{2}{n}\sum_{i=1}^{n}\mathbb{E}\left[\left\|v_i^t - (e_i^t + h_i^t)\right\|^2\right] + 2\mathbb{E}\left[\left\|\frac{1}{n}\sum_{i=1}^{n}(e_i^t + h_i^t)\right\|^2\right] \\
&\leq 2(1-\delta)\frac{1}{n}\sum_{i=1}^{n}\mathbb{E}\left[\left\|e_i^t + h_i^t\right\|^2\right] + \frac{2}{n}\sum_{i=1}^{n}\mathbb{E}\left[\left\|e_i^t + h_i^t\right\|^2\right] \\
&\leq \frac{8}{n}\sum_{i=1}^{n}\mathbb{E}\left[\|e_i^t\|^2\right] + \frac{8}{n}\sum_{i=1}^{n}\mathbb{E}\left[\|h_i^t\|^2\right] \\
&\leq \frac{40M^2}{\delta^2}.
\end{aligned}
$$

Then we continue (28) as follows

$$
\begin{aligned}
\mathbb{E}\left[\|\hat{e}^{t+1}\|^2\right] &\leq \sum_{l=0}^{t}[(1-\delta_{\mathrm{s}})(1+\hat{\eta})]^{t-l}(1-\delta_{\mathrm{s}})(1+\hat{\eta}^{-1})\gamma^2\cdot\frac{40M^2}{\delta^2} \\
&\leq \frac{(1-\delta_{\mathrm{s}})(1+\hat{\eta}^{-1})}{1-(1-\delta_{\mathrm{s}})(1+\hat{\eta})}\gamma^2\cdot\frac{40M^2}{\delta^2} \\
&\leq \gamma^2\underbrace{\frac{160(1-\delta_{\mathrm{s}})M^2}{\delta_{\mathrm{s}}^2\delta^2}}_{:=B^2},
\end{aligned}
$$

i.e. $\mathbb{E}\left[\|e^t\|\right] \leq C$ and $\mathbb{E}\left[\|\hat{e}^t\|\right] \leq \gamma B$. Therefore, we continue (26) as follows

$$
\mathbb{E}\left[\gamma\sum_{t\in\mathcal{B}}f(x^t) - f(x^*)\right] + \mathbb{E}\left[\gamma\sum_{t\in\mathcal{N}}c - g(x^t)\right] \leq \frac{1}{2}R^2 + \frac{1}{2}M^2\gamma^2T + M^2\gamma^2\frac{2\sqrt{1-\delta}}{\delta}T
$$
$$
+ M^2\gamma^2\frac{4\sqrt{10(1-\delta_{\mathrm{s}})}}{\delta_{\mathrm{s}}\delta}T. \tag{29}
$$

Assume that $\mathcal{B} = \emptyset$, then we have using the fact that $g(x^*) \leq 0$

$$
T\gamma c \leq \frac{1}{2}R^2 + \frac{1}{2}M^2\sum_{t=0}^{T-1}\gamma^2 + M\sum_{t=0}^{T-1}\gamma^2\|e^t\| + M^2\gamma^2\frac{2\sqrt{1-\delta}}{\delta}T + M^2\gamma^2\frac{4\sqrt{10(1-\delta_{\mathrm{s}})}}{\delta_{\mathrm{s}}\delta}T.
$$

This contradicts the assumption of the lemma (24). Therefore, we must have $\mathcal{B} \neq \emptyset$. If we have

$$
\gamma\mathbb{E}\left[\sum_{t\in\mathcal{B}}f(x^t) - f(x^*)\right] \leq 0,
$$

then part 2. holds automatically. If we have the opposite, i.e.

$$
\gamma\mathbb{E}\left[\sum_{t\in\mathcal{B}}f(x^t) - f(x^*)\right] > 0,
$$

then from (29) we have

$$
\gamma\mathbb{E}\left[\sum_{t\in\mathcal{N}}(c - g(x^*))\right] \leq \frac{1}{2}R^2 + \frac{1}{2}M^2\gamma^2T + M^2\gamma^2\frac{2\sqrt{1-\delta}}{\delta}T + M^2\gamma^2\frac{4\sqrt{10(1-\delta_{\mathrm{s}})}}{\delta_{\mathrm{s}}\delta}T.
$$

Since $g(x^*) \leq 0$, we have $c - g(x^*) \geq c$. Therefore, we have

$$
\mathbb{E}\left[\sum_{t\in\mathcal{N}}\gamma c\right] \leq \frac{1}{2}R^2 + \frac{1}{2}M^2\gamma^2T + M^2\gamma^2\frac{2\sqrt{1-\delta}}{\delta}T + M^2\gamma^2\frac{4\sqrt{10(1-\delta_{\mathrm{s}})}}{\delta_{\mathrm{s}}\delta}T. \tag{30}
$$

Assume $|\mathcal{B}| < \frac{T}{2}$, this means that $|\mathcal{N}| \geq \frac{T}{2}$. Therefore, from (30) we derive

$$
\frac{T}{2}\gamma c \leq \mathbb{E}\left[\sum_{t\in\mathcal{N}}\gamma c\right] \leq \frac{1}{2}R^2 + \frac{1}{2}M^2\gamma^2T + M^2\gamma^2\frac{2\sqrt{1-\delta}}{\delta}T + M^2\gamma^2\frac{4\sqrt{10(1-\delta_{\mathrm{s}})}}{\delta_{\mathrm{s}}\delta}T,
$$

which contradicts (24). Therefore, $|\mathcal{B}| \geq \frac{T}{2}$, i.e. part 1. holds.

$\square$

Now we are ready to prove our main theorem.

**Theorem D.3.** *Suppose that $\gamma$ and $c$ are chosen such that (24) holds. Then we have*

$$\mathbb{E}\left[f(\overline{x}^T) - f(x^*)\right] \leq \frac{R^2}{\gamma T} + M^2\gamma + 4M^2\gamma\frac{\sqrt{1-\delta}}{\delta} + 8M^2\gamma\frac{\sqrt{10(1-\delta_{\mathrm{s}})}}{\delta_{\mathrm{s}}\delta},$$

$$\mathbb{E}\left[g(\overline{x}^T)\right] \leq c.$$

*Proof.* We start by using the results of Lemma D.1. Using convexity of $f$ and Jensen inequality we get that if part 2. of Lemma D.1 holds, we have

$$\mathbb{E}\left[f(\overline{x}^T) - f(x^*)\right] \leq 0.$$

If part 2. does not hold, then we must have $|\mathcal{B}| \geq \frac{T}{2}$. Since $g(x^*) \leq 0$, from (25) we obtain

$$\gamma\mathbb{E}\left[\sum_{t\in\mathcal{B}} f(x^t) - f(x^*)\right] \leq \frac{1}{2}R^2 + \frac{1}{2}M^2\gamma^2 T + M^2\gamma^2\frac{2\sqrt{1-\delta}}{\delta}T + M^2\gamma^2\frac{4\sqrt{10(1-\delta_{\mathrm{s}})}}{\delta_{\mathrm{s}}\delta}T.$$

This implies that

$$\mathbb{E}\left[f(\overline{x}^T) - f(x^*)\right] \leq \frac{2}{\gamma T}\left(\frac{1}{2}R^2 + \frac{1}{2}M^2\gamma^2 T + M^2\gamma^2\frac{2\sqrt{1-\delta}}{\delta}T + M^2\gamma^2\frac{4\sqrt{10(1-\delta_{\mathrm{s}})}}{\delta_{\mathrm{s}}\delta}T.\right)$$

$$= \frac{R^2}{\gamma T} + M^2\gamma + 4M^2\gamma\frac{\sqrt{1-\delta}}{\delta} + 8M^2\gamma\frac{\sqrt{10(1-\delta_{\mathrm{s}})}}{\delta_{\mathrm{s}}\delta}.$$

Since $g(x^t) \leq c$ for $t \in \mathcal{B}$ we get from convexity of $g$ and Jensen inequality that

$$\mathbb{E}\left[g(\overline{x}^T)\right] \leq c.$$

$\square$

**Corollary D.4.** *If $\gamma = \frac{R\sqrt{\delta_{\mathrm{s}}\delta}}{M\sqrt{T}}$ and $c = \frac{32RM}{\sqrt{\delta_{\mathrm{s}}\delta T}}$, then we have*

$$\mathbb{E}\left[f(\overline{x}^T) - f(x^*)\right] \leq \frac{32MR}{\sqrt{\delta T}},$$

$$\mathbb{E}\left[g(\overline{x}^T)\right] \leq \frac{32MR}{\sqrt{\delta T}}.$$

*Proof.* Note that $\gamma c = \frac{R\sqrt{\delta_{\mathrm{s}}\delta}}{M\sqrt{T}}\frac{32RM}{\sqrt{\delta_{\mathrm{s}}\delta T}} = \frac{32R^2}{T}$, i.e. $\frac{T}{2}\gamma c = 16R^2$, and

$$\frac{1}{2}R^2 + \frac{1}{2}M^2\gamma^2 T + M^2\gamma^2\frac{2\sqrt{1-\delta}}{\delta}T + M^2\gamma^2\frac{\sqrt{10(1-\delta_{\mathrm{s}})}}{\delta_{\mathrm{s}}\delta}T$$

$$= \frac{1}{2}R^2 + \frac{1}{2}M^2 T\frac{R^2\delta\delta_{\mathrm{s}}}{M^2 T} + M^2 T\frac{2\sqrt{1-\delta}}{\delta}\frac{R^2\delta\delta_{\mathrm{s}}}{M^2 T} + M^2 T\frac{4\sqrt{10(1-\delta_{\mathrm{s}})}}{\delta_{\mathrm{s}}\delta}\frac{R^2\delta_{\mathrm{s}}\delta}{M^2 T}$$

$$= \frac{1}{2}R^2 + \frac{1}{2}R^2\delta\delta_{\mathrm{s}} + 2R^2\sqrt{1-\delta}\delta_{\mathrm{s}} + 4\sqrt{10(1-\delta_{\mathrm{s}})}R^2 \leq 16R^2.$$

Therefore, (24) is satisfied. Hence, we have from Theorem D.3

$$\mathbb{E}\left[f(\overline{x}^T) - f(x^*)\right] \leq \frac{R^2}{\frac{R\sqrt{\delta_{\mathrm{s}}\delta}}{M\sqrt{T}}T} + M^2\frac{R\sqrt{\delta_{\mathrm{s}}\delta}}{M\sqrt{T}} + 4M^2\frac{R\sqrt{\delta_{\mathrm{s}}\delta}}{M\sqrt{T}}\frac{\sqrt{1-\delta}}{\delta} + 8M^2\frac{R\sqrt{\delta_{\mathrm{s}}\delta}}{M\sqrt{T}}\frac{\sqrt{10(1-\delta_{\mathrm{s}})}}{\delta_{\mathrm{s}}\delta}$$

$$= \frac{MR}{\sqrt{\delta_{\mathrm{s}}\delta T}} + \frac{MR\sqrt{\delta_{\mathrm{s}}\delta}}{\sqrt{T}} + \frac{4MR\sqrt{(1-\delta)\delta_{\mathrm{s}}}}{\sqrt{\delta T}} + \frac{8MR\sqrt{10(1-\delta_{\mathrm{s}})}}{\sqrt{\delta_{\mathrm{s}}\delta}}$$

$$\leq \frac{32MR}{\sqrt{\delta_{\mathrm{s}}\delta T}},$$

and

$$g(\overline{x}^T) \leq c = \frac{32MR}{\sqrt{\delta_{\mathrm{s}}\delta T}}.$$

$\square$

# E. Lower bound under Communication Compression for Non-smooth Convex Setting

In this section, we establish a lower bound in non-smooth convex setting, assuming workers can compute exact subgradients $f'(x) \in \partial f(x)$ or $g'(x) \in \partial g(x)$, and the compression is the only source of stochasticity in the training. First, in the next subsection, we provide some preliminary background on the class of zero-respecting algorithms following the exposition in (Huang et al., 2022), and justify that our Safe-EF method satisfies this general property. In the subsequent Appendices E.2 and E.3, we provide the proof of Theorem 4.1.

## E.1. Zero-respecting algorithms

Let $[x]_j$ denote the $j$-th coordinate of a vector $x \in \mathbb{R}^d$ for $j \in [d]$, and define $\mathrm{prog}(x)$ as

$$\mathrm{prog}(x) := \begin{cases} 0 & \text{if } x = 0; \\ \max_{1 \leq j \leq d} \{j \colon [x]_j \neq 0\}, & \text{otherwise.} \end{cases}$$

Similarly, for a set of points $X = \{x_1, x_2 \dots\}$, we define $\mathrm{prog}(X) := \max_{x \in X} \mathrm{prog}(x)$. It holds that $\mathrm{prog}(X \cup Y) = \max\{\mathrm{prog}(X), \mathrm{prog}(Y)\}$ for any $X, Y \subseteq \mathbb{R}^d$, and $\mathrm{prog}(X) \leq \mathrm{prog}(\widetilde{X})$ for any $X \subseteq \widetilde{X} \subseteq \mathbb{R}^d$.

We examine a distributed learning framework incorporating communication compression. For each worker $i$ and time step $t \geq 0$, we denote by $y_i^t$ and $z_i^t$ the points at which worker $i$ queries its subgradient (of $f_i$ and/or $g_i$) and function (of $f_i$ and/or $g_i$) oracles, respectively[5]. In more detail, $O_{i,f_i}(y_i^t, z_i^t)$ returns a pair of the subgradient of $f_i'(y_i^t)$ and the function value $f_i(z_i^t)$, namely,

$$(f_i'(y_i^t), f_i(z_i^t)) \in O_{i,f_i}(y_i^t, z_i^t) := (O_{i,f_i}^{\mathrm{sg}}(y_i^t, z_i^t), O_{i,f_i}^{\mathrm{fv}}(y_i^t, z_i^t)),$$

where $f_i'(y_i^t) \in \partial f_i(y_i^t)$ is an arbitrary selection of subgradient element from subdifferential of $f_i$ at the point $y_i^t$. We assume similarly the oracle for each constraint function $g_i$, $O_{i,g_i}(y_i^t, z_i^t)$ which returns a pair $(g_i'(y_i^t), g_i(z_i^t))$, where $g_i'(y_i^t) \in \partial g_i(y_i^t)$. Additionally, $x_i^t$ represents the local model updated by worker $i$ after the $t$-th query. It is important to note that $y_i^t$ and $z_i^t$ are not necessarily equal to the previous local model $x_i^{t-1}$; instead, they may serve as auxiliary vectors.

Between the $(t-1)$-th and $t$-th gradient queries, each worker is allowed to communicate with the server by transmitting (compressed) vectors. For worker $i$, we let $\mathcal{V}_{w_i \to s}^t$ denote the set of vectors that worker $i$ aims to send to the server, i.e., the vectors before compression. Due to communication compression, the vectors received by the server from worker $i$, which we denote by $\mathcal{V}_{w_i \to s}^{t,\star}$, are the compressed version of $\mathcal{V}_{w_i \to s}^t = \mathcal{C}_i(\mathcal{V}_{w_i \to s}^{t,\star})$[6] with some underlying compressors $\mathcal{C}_i$. Note that $\mathcal{V}_{w_i \to s}^t$ is a set that may include multiple vectors, and its cardinality equals the rounds of communication. After receiving the compressed vectors from all workers, the server will broadcast some vectors back to all workers. We let $\mathcal{V}_{s \to w}^t$ denote the set of vectors that the server aims to send to workers. Since we consider the setting with unidirectional compression only, then $\mathcal{V}_{s \to w}^t \equiv \mathcal{V}_{s \to w}^{t,\star}$[7].

We now extend the zero-respecting property (Huang et al., 2022) to distributed learning with communication compression with functional constraints.

**Definition E.1.** We say a distributed algorithm $A$ is zero-respecting if for any $t \geq 0$ and $1 \leq k \leq d$, the following requirements are satisfied:

1. If worker $i$ queries at $y_i^t$ and $z_i^t$ with $[y_i^t]_k \neq 0$, then one of the following must be true:

$$\begin{cases} \text{there exists some } 0 \leq s < t \text{ such that } [x_i^s]_k \neq 0; \\ \text{there exists some } 1 \leq s < t \text{ such that } [O_{f_i,i}(y_i^s)]_k \neq 0 \text{ or } [O_{g_i,i}(y_i^s)]_k \neq 0; \\ \text{there exists some } 1 \leq s < t \text{ such that worker } i \text{ has received some } v \in \mathcal{V}_{s \to w}^t \text{ with } [v]_k \neq 0; \\ \text{there exists some } 1 \leq s < t \text{ such that worker } i \text{ has compressed some } v \in \mathcal{V}_{w_i \to s}^t \text{ with } [v]_k \neq 0; \end{cases}$$

---

[5]We consider deterministic oracles only.

[6]The compression is performed vector-wise.

[7]$\mathcal{V}_{s \to w}^{t,\star} = \mathcal{C}_0(\mathcal{V}_{s \to w}^t)$.

2. If the local model $x_i^t$ of worker $i$, after $t$-th query, has $[x_i^t]_k \neq 0$, then one of the following must be true:

$$\begin{cases} \text{there exists some } 0 \leq s < t \text{ such that } [x_i^s]_k \neq 0; \\ \text{there exists some } 1 \leq s < t \text{ such that } [O_{f_i,i}(y_i^s)]_k \neq 0 \text{ or } [O_{g_i,i}(y_i^s)]_k \neq 0; \\ \text{there exists some } 1 \leq s < t \text{ such that worker } i \text{ has received some } v \in \mathcal{V}_{s \to w}^t \text{ with } [v]_k \neq 0; \\ \text{there exists some } 1 \leq s < t \text{ such that worker } i \text{ has compressed some } v \in \mathcal{V}_{w_i \to s}^t \text{ with } [v]_k \neq 0; \end{cases}$$

3. If worker $i$ aims to send some $v \in \mathcal{V}_{w_i \to s}^t$ with $[v]_k \neq 0$, then one of the following must be true:

$$\begin{cases} \text{there exists some } 0 \leq s < t \text{ such that } [x_i^s]_k \neq 0; \\ \text{there exists some } 1 \leq s < t \text{ such that } [O_{i,f_i}(y_i^s)]_k \neq 0 \text{ or } [O_{i,g_i}(y_i^s)]_k \neq 0; \\ \text{there exists some } 1 \leq s < t \text{ such that worker } i \text{ has received some } v' \in \mathcal{V}_{s \to w}^t \text{ with } [v']_k \neq 0; \\ \text{there exists some } 1 \leq s < t \text{ such that worker } i \text{ has compressed some } v' \in \mathcal{V}_{w_i \to s}^t \text{ with } [v']_k \neq 0; \end{cases}$$

4. If the server aims to broadcast some $v \in \mathcal{V}_{s \to w}^t$ with $[v]_k \neq 0$, then one of the following must be true:

$$\begin{cases} \text{there exists some } 1 \leq s < t \text{ and } 1 \leq i \leq n \text{ such that the server has received some } v' \in \mathcal{V}_{w_i \to s}^s \text{ with } [v']_k \neq 0; \end{cases}$$

**Safe-EF is zero-respecting.** Fundamentally, the zero-respecting property ensures that any increase in the number of nonzero coordinates in $x_i^t$, $y_i^t$, or other related vectors at worker $i$ stems from its past local gradient updates, local compression operations, or synchronization with the server. Likewise, any expansion of nonzero coordinates in the server's vectors must result from receiving compressed messages from workers. Notably, this definition explicitly prohibits expanding the set of nonzero entries through function value queries of $f_i$ and/or $g_i$. Therefore, our algorithm class excludes zero-order methods. Nevertheless, function values can be used to set a stepsize or coefficients in linear combination to compute local model $x_i^t$. For instance, in Safe-EF function evaluation of $g_i$ are used to define an update direction:

$$h_i^t = f_i'(x^t) \mathbb{1}(g(x^t) \leq c) + g_i'(x^t) (g(x^t) > c).$$

In this case, function values are only used to choose which of the directions, $f_i'(x^t)$ or $g_i'(x^t)$, to follow, but they cannot be used to compute the update direction itself.

### E.2. Lower bound in unconstrained case

We first establish the lower bound in unconstrained setting when $g(x) \equiv 0$, which is the most challenging part of the proof. Without loss of generality, we assume that $x^0 = 0$. Given local loss functions $\{f_i\}_{i=1}^n \subseteq \mathcal{F}_{R,M}$, compressors $\{\mathcal{C}_i\}_{i=1}^n \subseteq \mathcal{C}(\delta)$, and an algorithm $A \in \mathcal{A}_{\{\mathcal{C}\}_{i=1}^n}^U$ to solve problem (1), we let $\hat{x}_{A,\{f_i\}_{i=1}^n,\{\mathcal{C}_i\}_{i=1}^n,T}$ denote the output of algorithm $A$ using no more than $T$ subgradient queries and rounds of communication by each worker node. Let us define the minimax measure in unconstrained case as

$$\inf_{A \in \mathcal{A}} \sup_{\{\mathcal{C}_i\}_{i=1}^n \subseteq \mathcal{C}(\delta)} \sup_{\{f_i\}_{i=1}^n \subseteq \mathcal{F}_{R,M}} \mathbb{E}\left[ f(\hat{x}_{A,\{f_i\}_{i=1}^n,\{\mathcal{C}_i\}_{i=1}^n,T}) - f^* \right]. \tag{31}$$

In (31), we do not require the compressors $\{\mathcal{C}_i\}_{i=1}^n$ to be distinct or independent. We allow the compression parameter $\delta$ to be accessible by algorithm $A$. Let $[x]_j$ denote the $j$-th coordinate of a vector $x \in \mathbb{R}^d$ for $j \in [d]$, and define $\text{prog}(x)$ as

$$\text{prog}(x) := \begin{cases} 0 & \text{if } x = 0; \\ \max_{1 \leq j \leq d}\{j : [x]_j \neq 0\} & \text{otherwise.} \end{cases}$$

In other words, $\text{prog}(x)$ outputs the largest coordinate of input $x$ that corresponds to a non-zero entry. Importantly, $\text{prog}(x)$ satisfies $\text{prog}(\mathcal{X} \cup \mathcal{Y}) = \max\{\text{prog}(\mathcal{X}), \text{prog}(\mathcal{Y})\}$ for any $\mathcal{X}, \mathcal{Y} \in \mathbb{R}^d$, and $\text{prog}(\mathcal{X}) \leq \text{prog}(\widetilde{\mathcal{X}})$ for any $\mathcal{X} \subseteq \widetilde{\mathcal{X}} \subseteq \mathbb{R}^d$ (see, e.g., (Huang et al., 2022)). Now we are ready to state and prove the lower bound stated in the unconstrained setting.

**Theorem E.2** (Unconstrained setting). *For any $R, M > 0, n \geq 2, \delta \leq 0.3, T \geq \delta^{-2}$ there exist functions $\{f_i\}_{i=1}^n \subseteq \mathcal{F}_{R,M}$, compressors $\{\mathcal{C}_i\}_{i=1}^n \subseteq \mathcal{C}(\delta)$, oracles $\{\mathcal{O}_{f_i,i}\}_{i=1}^n$, and the starting point $x^0 = 0$ such that for any first-order algorithm $A \in \mathcal{A}_{\{\mathcal{C}_i\}_{i=1}^n}^U$ run for $T \leq d$ iterations from $x^0$, satisfies*

$$\mathbb{E}\left[ f(\hat{x}_{A,\{f_i\}_{i=1}^n,\{\mathcal{C}\}_{i=1}^n,T}) - \min_{x \in \mathbb{R}^d} f(x) \right] \geq \Omega\left( \frac{MR}{\sqrt{\delta T}} \right).$$

*Proof.* **Step 1.** Let us fix some $R$ and define $\mathcal{S} := \{x \in \mathbb{R}^d \mid \|x\|_2 \leq \frac{R}{2}\}$. Let $h \colon \mathbb{R}^d \to \mathbb{R}$ be defined as

$$h(x) := \begin{cases} C \cdot \max\limits_{1 \leq j \leq T} x_j + \frac{\mu}{2}\|x\|_2^2 & \text{if } x \in \mathcal{S}, \\ C \cdot \max\limits_{1 \leq j \leq T} x_j + \frac{\mu R}{4}\|x\|_2 & \text{if } x \notin \mathcal{S}. \end{cases}$$

Here we assume that $T \leq d$. The constant $C = \frac{M\sqrt{T}}{1+\sqrt{\delta T}}$ and $\mu = \frac{2M}{R(1+\sqrt{\delta T})}$. This implies that $C = \frac{R\mu\sqrt{T}}{2}$. Note that it is never optimal to have $[x^*]_j \neq 0$ for $T < j \leq d$, and by symmetry, we know that

$$[x^\star]_1 = \cdots = [x^\star]_T.$$

Thus, as long as $C \leq \frac{R\mu\sqrt{T}}{2}$ the optimal solution $x^*$ and optimal value of the problem $f^* := \min_x f(x)$ are given by

$$[x^*]_j = \begin{cases} -\frac{C}{\mu T} & \text{for } 1 \leq j \leq T, \\ 0 & \text{for } T < j \leq d, \end{cases} \quad \text{and} \quad f^* = -\frac{C^2}{2\mu T}.$$

One can show that the function $h$ is convex. Indeed, this is because taking max and/or a sum of convex functions preserves convexity. We consider the following subgradient oracle $O_h$

$$h'(x) = \begin{cases} \mu x + Ce_k & \text{if } x \in \mathcal{S}, \\ \mu R \frac{x}{4\|x\|} + Ce_k & \text{otherwise}, \end{cases}$$

where $k$ is the smallest index such that $[x]_k = \max\limits_{1 \leq j \leq T}[x]_j$. We set $f_i \equiv h$ with $O_i \equiv O_h$ for all $i \in [n]$. Note that the first part of the subgradient (either $\mu x$ or $\mu R \frac{x}{2\|x\|}$) is proportional to $x$. Therefore, the algorithms are hampered by oracle $O_i$ to reach more non-zero coordinates due to the second part $Ce_k$ only. However, it might increase $\text{prog}(O_i(x))$ at most by one, namely,

$$\text{prog}(O_i(x)) \leq \text{prog}(x) + 1. \tag{32}$$

**Step 2.** Next, we assume that each worker $i$ uses Rand-$K$ compressor with $K = \lceil d\delta \rceil$. Moreover, we assume that the randomness of the compressors is shared among workers. Then this compressor belongs to $\mathbb{C}(\delta)$. This step ensures there is no speedup of the final rate in the number of workers $n$.

**Step 3.** We let $v_{s \to w}^t$ be the vector that workers receive from the central server in the $t$-th communication (similar definition is used for $v_{w_i \to s}^t$) and let $x_i^t$ be the local model that worker $i$ produces after the $t$-th communication round. Recall that algorithms satisfy the zero-respecting property. Therefore, we find that each worker can only achieve one more non-zero coordinate in the local model by local subgradient updates based on the received messages from the central server. Thus, we have that

$$\text{prog}(x_i^t) \leq \max\limits_{1 \leq s \leq t} \text{prog}(v_{s \to w}^s) + 1. \tag{33}$$

By further noting that vector $v_{s \to w}^t$ sent by the central server can be traced back to past vectors received from all workers, we have

$$\text{prog}(v_{s \to w}^t) \leq \max\limits_{1 \leq s \leq t} \max\limits_{1 \leq i \leq n} \text{prog}(v_{w_i \to s}^s). \tag{34}$$

Combining (33) and (34), we reach

$$\text{prog}(x_i^t) \leq \max\limits_{1 \leq s \leq t} \max\limits_{1 \leq i \leq n} \text{prog}(v_{w_i \to s}^s) + 1. \tag{35}$$

**Step 4.** Let

$$\hat{x} \in \text{span}\left(\left\{x_i^t \mid 0 \leq t \leq T, 1 \leq i \leq n\right\}\right).$$

be the final algorithm output after $T$ subgradient queries on each worker. By (35), we have

$$\text{prog}(\hat{x}) \leq \max\limits_{1 \leq t \leq T} \max\limits_{1 \leq i \leq n} \text{prog}(v_{w_i \to s}^t) + 1.$$

By Lemma E.3, we have

$$\mathbb{P}(\max\limits_{1 \leq t \leq T} \max\limits_{1 \leq i \leq n} \text{prog}(v_{w_i \to s}^t) \geq T - 1) \leq \exp\left((e-1)T\lceil d\delta \rceil / -T + 1\right).$$

Note that if $\text{prog}(\hat{x}) < T$ then we have

$$f(\hat{x}) \geq 0 \Leftrightarrow f(\hat{x}) - f^\star \geq -f^\star = \frac{C^2}{2\mu T}.$$

Therefore, we have

$$\mathbb{E}\left[f(\hat{x}) - f^\star\right] \geq \left(1 - \exp\left((e-1)T\lceil d\delta \rceil / d - T + 1\right)\right)\frac{C^2}{2\mu T}.$$

If we let $d = \lfloor 5T\delta \rfloor$ and $T$ to be no less than $\frac{1}{\delta^2}$, we have

$$d = \lfloor 5T\delta \rfloor \geq 5T\delta - 1 \geq 4T\delta + \frac{1}{\delta^2}\delta - 1 \geq 4T\delta \geq \frac{4}{\delta} \geq 4.$$

Then it is easy to verify

$$(e-1)T\lceil d\delta \rceil/d + 1 - T \leq (e-1)T(d\delta+1)/d + 1 - T$$

$$= (e-1)T\delta + \frac{(e-1)T}{d} + 1 - T$$

$$\leq (e-1)T\delta + (e-1)\frac{T\delta}{4} + 1 - T$$

$$= (e-1)\frac{5T\delta}{4} + 1 - T.$$

Note that since $\delta \leq 0.3$ and $T \geq \frac{1}{\delta^2}$ we have

$$(e-1)\frac{5T\delta}{4} + 1 - T \leq -1 \Leftrightarrow T\left(1 - \frac{(e-1)5\delta}{4}\right) \geq 2 \Leftarrow 6.25 \cdot \left(1 - \frac{(e-1)5 \cdot 0.4}{4}\right) \approx 3.95 > 2.$$

Since the last inequality holds, then we have $(e-1)\frac{5T\delta}{4} + 1 - T < -1$. Therefore, this leads to

$$\mathbb{E}\left[f(\hat{x}) - f^\star\right] \geq \Omega\left(\frac{C^2}{2\mu T}\right) = \Omega\left(\frac{M^2 T}{(1+\sqrt{\delta T})^2}\frac{1}{2T}\frac{R(1+\sqrt{\delta T})}{2M}\right) = \Omega\left(\frac{MR}{1+\sqrt{\delta T}}\right).$$

$$\square$$

**Lemma E.3** (Technical lemma). *In example used in the proof of Theorem E.2, it holds that*

$$\mathbb{P}\left(\max_{1 \leq t \leq T} \max_{1 \leq i \leq n} \mathrm{prog}(v_{w_i \to s}^t) \geq T - 1\right) \leq \exp\left((e-1)T\lceil d\delta \rceil/d - T + 1\right).$$

*Proof.* Note that at the $t$-th round of communication where $1 \leq t \leq T$, the non-zero coordinates of $v_{w_i \to s}^{(t,\star)}$, *the vector that is to be transmitted by worker $i$ to the server before compression*, are achieved by utilizing previously received vectors $\{v_{s \to w}^{(s)}: 1 \leq s \leq t - 1\}$ and local subgradient queries. Following the argument in Step 3 of Theorem E.2, we find that worker $i$ can only achieve one more non-zero coordinate in $v_{w_i \to s}^{(t,\star)}$ by local subgradient updates based on received vectors $\{v_{s \to w}^{(s)} \mid 1 \leq s \leq t - 1\}$. Therefore, it holds that

$$\mathrm{prog}(v_{w_i \to s}^{(t,\star)}) \leq \max_{1 \leq s \leq t-1} \mathrm{prog}(v_{s \to w}^{(s)}) + 1 \leq \max_{1 \leq s \leq t-1} \max_{1 \leq i \leq n} \mathrm{prog}(v_{w_i \to s}^s) + 1 =: B^{(t-1)}. \tag{36}$$

We additionally define $B^{(0)} = 1$. By the definition of $B^{(t)}$ and that

$$\mathrm{prog}(v_{w_i \to s}^t) \leq \mathrm{prog}(v_{w_i \to s}^{(t,\star)}), \tag{37}$$

it naturally holds that

$$B^{(t-1)} \leq B^{(t)} = \max_{1 \leq s \leq t} \max_{1 \leq i \leq n} \mathrm{prog}(v_{w_i \to s}^s) + 1$$

$$= \max\left\{B^{(t-1)}, \max_{1 \leq i \leq n} \mathrm{prog}(v_{w_i \to s}^t) + 1\right\}$$

$$\overset{(37)}{\leq} \max\left\{B^{(t-1)}, \max_{1 \leq i \leq n} \mathrm{prog}(v_{w_i \to s}^{(t,\star)}) + 1\right\}$$

$$\overset{(36)}{\leq} \max\left\{B^{(t-1)}, B^{(t-1)} + 1\right\} \leq B^{(t-1)} + 1. \tag{38}$$

Therefore, one round of communication can increase $B^{(t)}$ at most by 1. Moreover, (38) implies that $B^{(t)} = B^{(t-1)} + 1$ only if $\max_{1 \leq i \leq n} \mathrm{prog}(v_{w_i \to s}^{(t,\star)}) = \max_{1 \leq i \leq n} \mathrm{prog}(v_{w_i \to s}^t)$. Let $k = \max_{1 \leq i \leq n} \mathrm{prog}(v_{w_i \to s}^{(t,\star)})$. Recall that the compressors $\{\mathcal{C}\}_{i=1}^n$ share the randomness, we therefore conclude that having $\max_{1 \leq i \leq n} \mathrm{prog}(v_{w_i \to s}^t) = \max_{1 \leq i \leq n} \mathrm{prog}(v_{w_i \to s}^{(t,\star)}) = k$ is equivalent to that coordinate index $k$ is chosen to communicate in communication round $t$, which happens with probability $\frac{K}{d}$. Therefore, we

have

$$\mathbb{P}(B^{(t)} = B^{(t-1)} + 1) \leq \mathbb{P}(\max_{1 \leq i \leq n} \text{prog}(v_{w_i \to s}^t) = \max_{1 \leq i \leq n} \text{prog}(v_{w_i \to s}^{(t,\star)}))$$

$$= \mathbb{P}\left(\text{the coordinate index} \max_{1 \leq i \leq n} \text{prog}(v_{w_i \to s}^{(t,\star)}) \text{ is chosen at round } t\right) = \frac{K}{d}.$$

Let us define the event $E^t = \{\text{the coordinate index} \max_{1 \leq i \leq n} \text{prog}(v_{w_i \to s}^{(t,\star)}) \text{ is chosen at round } t\}$. Since the compression happens uniformly at random, we have $\mathbb{1}(E^{(1)}), \ldots, \mathbb{1}(E^t)$ are i.i.d. $\text{Be}(\frac{K}{d})$ random variables where $\mathbb{1}(\cdot)$ is the indicator function. By the above argument, we also have $B^{(t)} - B^{(t-1)} \leq \mathbb{1}(E^t)$ for any $1 \leq t \leq T$. As a result, we reach by Markov's inequality

$$\mathbb{P}(B^{(T)} \geq T) = \mathbb{P}(e^{B^{(T)}} \geq e^T)$$

$$\leq e^{-T}\mathbb{E}\left[\exp\left(B^{(T)}\right)\right]$$

$$= e^{-T}\mathbb{E}\left[\exp\left(B^{(0)} + \sum_{t=1}^{T}(B^{(t)} - B^{(t-1)})\right)\right]$$

$$\leq e^{-T}\mathbb{E}\left[e^{B^{(0)}}\right]\prod_{t=1}^{T}\mathbb{E}\left[\exp\left(\mathbb{1}(E^t)\right)\right]$$

$$= e^{-(T-1)}\prod_{t=1}^{T}\left((1 - \frac{K}{d}) \cdot 1 + \frac{K}{d} \cdot e\right)$$

$$= e^{-(T-1)}\prod_{t=1}^{T}\left(1 + \frac{K}{d}(e - 1)\right)$$

$$\leq e^{-(T-1)}\prod_{t=1}^{T}e^{(e-1)K/d}$$

$$= e^{(e-1)TK/d - T + 1}.$$

This concludes the proof of the lemma. $\qquad\square$

### E.3. Proof of Theorem 4.1 (constrained case)

Now we are ready to extend the proof of Theorem 4.1 to constrained setting based on the construction in Theorem E.2. Notice that the function classes $\mathcal{F}_{R,M}$ and $\mathcal{G}_{R,M}$ for objective and constraints have the same properties: convex with $M$-bounded subgradients. Moreover, in the construction of Theorem E.2, all functions $f_i$ are identical and equal to $f$. Thus, we can set $g_i(x) := f(x) - \min_{y \in \mathbb{R}^d} f(y)$ for all $i \in [n]$. Then such problem is in the class $\mathcal{H}_{R,M}$ by construction, and it has a unique feasible point, $x^*$, which also coincides with the solution to unconstrained problem $\min_{x \in \mathbb{R}^d} f(x)$. Since $\partial f_i(x) = \partial g_i(x)$ for any $x \in \mathbb{R}^d$ and all $i \in [n]$, the trajectory of zero-respecting algorithm on the unconstrained problem $\min_{x \in \mathbb{R}^d} f(x)$ and the constrained problem

$$\min_{x \in \mathbb{R}^d} f(x) \qquad \text{s.t.} \qquad g(x) \leq 0$$

are identical. Therefore, the statement of Theorem E.2 implies the lower bound in Theorem 4.1.

## F. Convergence Upper Bound for Safe-EF in Stochastic Setting

We first recall a standard concentration inequality result for sub-Gaussian random vector.

**Lemma F.1** (Lemma C.3 from Gorbunov et al. (2019)). *Let $\{\xi_k\}_{k=1}^N$ be the sequence of random vectors with values in $\mathbb{R}^n$ such that*

$$\mathbb{E}\left[\xi_k \mid \xi_{k-1}, \ldots, \xi_1\right] = 0 \text{ a.s. } \forall\, k \in \{1, \ldots, N\},$$

*and set $S_N := \sum_{k=1}^N \xi_k$. Assume that the sequence is $\{\xi_k\}_{k=1}^N$ are sub-Gaussian, i.e.,*
$$\mathbb{E}\left[\exp(\|\xi_k\|^2/\sigma_k^2) \mid \xi_{k-1}, \ldots, \xi_1\right] \leq \exp(1) \text{ a.s. } \forall\, k \in \{1, \ldots, N\},$$
*where $\sigma_1, \ldots, \sigma_N$ are some positive numbers. Then for all $b \geq 0$ we have*

$$\mathbb{P}\left(\|S_N\| \geq (\sqrt{2} + \sqrt{2}b)\sqrt{\sum_{k=1}^N \sigma_k^2}\right) \leq \exp(-b^2/3).$$

We first establish several lemmas.

**Lemma F.2.** *Assume that Assumption 5.1 holds. Assume that the compressors $\{\mathcal{C}_i\}_{i=1}^n$ are deterministic (e.g., Top-K). Then for all $t \geq 0$ and $i \in [n]$ we have $\|e_i^t\|^2 \leq \frac{4(1-\delta)}{\delta^2}M^2$.*

*Proof.* Using the properties of the compressors $\{\mathcal{C}_i\}_{i=1}^n$, we get by induction[8] that (with the choice $\eta = \frac{\delta}{2(1-\delta)}$)

$$\|e^{t+1}\|^2 = \left\|\frac{1}{n}\sum_{i=1}^n e_i^{t+1}\right\|^2 \leq \frac{1}{n}\sum_{i=1}^n \|e_i^{t+1}\|^2 = \frac{1}{n}\sum_{i=1}^n \|e_i^t + h_i^t - \mathcal{C}_i(e_i^t + h_i^t)\|^2$$

$$\leq \frac{1-\delta}{n}\sum_{i=1}^n \|e_i^t + h_i^t\|^2$$

$$\leq (1-\delta)(1+\eta)\frac{1}{n}\sum_{i=1}^n \|e_i^t\|^2 + (1-\delta)(1+\eta^{-1})M^2$$

$$\leq \sum_{l=0}^t [(1-\delta)(1+\eta)]^{t-l}(1-\delta)(1+\eta^{-1})M^2$$

$$\leq \frac{(1-\delta)(1+\eta^{-1})}{1-(1-\delta)(1+\eta)}M^2 = \frac{(1-\delta)(1+\eta^{-1})}{\delta - \eta(1-\delta)}M^2 = \frac{2(1-\delta)(1+\eta^{-1})}{\delta}M^2 \leq \frac{4(1-\delta)}{\delta^2}M^2,$$

which concludes the proof. $\qquad\square$

**Theorem F.3.** *Let Assumptions 5.1 and 5.2 hold. Let $\beta \in (0, 1)$ be the failure probability. Suppose $\gamma^2 w_t \leq \frac{n}{32M^2}$.[9] For every $0 \leq t \leq T-1$ we have*

$$\mathbb{E}\left[\exp(S_t) \mid \mathcal{F}_t\right] \leq \exp\left(48M^2\sum_{l=t}^{T-1}\gamma^2 w_l + \frac{8\sigma_{\mathrm{fv}}^2}{nN_{\mathrm{fv}}}\sum_{l=t}^{T-1} w_l^2\gamma^2\right),$$

*where $S_t$ is defined in (44).*

*Proof.* We use the same definition of $\widetilde{x}^t$ established in (22):
$$\widetilde{x}^t = x^t - \gamma e^t \quad \text{where} \quad \widetilde{x}^0 = x^0. \tag{39}$$
We start by extending the norm of $\|\widetilde{x}^{t+1} - x\|^2$:
$$\|\widetilde{x}^{t+1} - x\|^2 = \|\widetilde{x}^t - x^*\|^2 - 2\gamma\langle h^t, \widetilde{x}^t - x^*\rangle + \gamma^2\|h^t\|^2$$
$$= \|\widetilde{x}^t - x^*\|^2 - 2\gamma\langle h^t, x^t - x^*\rangle - 2\gamma\langle h^t, \widetilde{x}^t - x^t\rangle + \gamma^2\|h^t\|^2.$$

Rearranging terms gives us
$$2\gamma\langle h^t, x^t - x\rangle = \|\widetilde{x}^t - x\|^2 - \|\widetilde{x}^{t+1} - x\|^2 - 2\gamma\langle h^t, \widetilde{x}^t - x^t\rangle + \gamma^2\|h^t\|^2. \tag{40}$$

---

[8]The base of induction obviously holds since $\|e_i^0\| = 0$.

[9]This restriction is needed to apply Lemma 2.2 from Liu et al. (2023).

Note that for $t \in \mathcal{N}$ we have $h^t = \frac{1}{n}\sum_{i=1}^n g_i'(x^t, \xi_i^t)$, and for $t \in \mathcal{B}$ we have $h^t = \frac{1}{n}\sum_{i=1}^n f_i'(x^t, \xi_i^t)$. Therefore, we get from (40)

$$
\frac{2\gamma}{n}\sum_{i=1}^n \langle g_i'(x^t, \xi_i^t), x^t - x\rangle \mathbb{1}(t \in \mathcal{N}) + \frac{2\gamma}{n}\sum_{i=1}^n \langle f_i'(x^t, \xi_i^t), x^t - x\rangle \mathbb{1}(t \in \mathcal{B})
$$

$$
\leq \|\widetilde{x}^t - x\|^2 - \|\widetilde{x}^{t+1} - x\|^2 - 2\gamma\langle h^t, \widetilde{x}^t - x^t\rangle
$$

$$
+ \frac{\gamma^2}{n}\sum_{i=1}^n \|g_i'(x^t, \xi_i^t)\|^2 \mathbb{1}(t \in \mathcal{N}) + \frac{\gamma^2}{n}\sum_{i=1}^n \|f_i'(x^t, \xi_i^t)\|^2 \mathbb{1}(t \in \mathcal{B})
$$

$$
\leq \|\widetilde{x}^t - x\|^2 - \|\widetilde{x}^{t+1} - x\|^2 - 2\gamma\langle h^t, \widetilde{x}^t - x^t\rangle
$$

$$
+ \frac{2\gamma^2}{n}\sum_{i=1}^n \|g_i'(x^t)\|^2 \mathbb{1}(t \in \mathcal{N}) + \frac{2\gamma^2}{n}\sum_{i=1}^n \|g_i'(x^t) - g_i'(x^t, \xi_i^t)\|^2 \mathbb{1}(t \in \mathcal{N})
$$

$$
+ \frac{2\gamma^2}{n}\sum_{i=1}^n \|f_i'(x^t)\|^2 \mathbb{1}(t \in \mathcal{B}) + \frac{2\gamma^2}{n}\sum_{i=1}^n \|f_i'(x^t) - f_i'(x^t, \xi_i^t)\|^2 \mathbb{1}(t \in \mathcal{B}). \tag{41}
$$

Note that we have

$$
|\langle h^t, \widetilde{x}^t - x^t\rangle| \leq \|h^t\| \cdot \gamma\|e^t\|
$$

$$
\leq M \cdot \gamma \frac{2\sqrt{1-\delta}}{\delta}M = \frac{2\sqrt{1-\delta}}{\delta}\gamma M^2.
$$

Therefore, we continue from (41) as follows

$$
\frac{2\gamma}{n}\sum_{i=1}^n \langle g_i'(x^t, \xi_i^t), x^t - x\rangle \mathbb{1}(t \in \mathcal{N}) + \frac{2\gamma}{n}\sum_{i=1}^n \langle f_i'(x^t, \xi_i^t), x^t - x\rangle \mathbb{1}(t \in \mathcal{B})
$$

$$
\leq \|\widetilde{x}^t - x\|^2 - \|\widetilde{x}^{t+1} - x\|^2 + \frac{4\sqrt{1-\delta}}{\delta}\gamma^2 M^2 + 2\gamma^2 M^2
$$

$$
+ \frac{2\gamma^2}{n}\sum_{i=1}^n \|g_i'(x^t) - g_i'(x^t, \xi_i^t)\|^2 \mathbb{1}(t \in \mathcal{N}) + \frac{2\gamma^2}{n}\sum_{i=1}^n \|f_i'(x^t) - f_i'(x^t, \xi_i^t)\|^2 \mathbb{1}(t \in \mathcal{B}).
$$

We add and subtract full subgradients and derive

$$
\frac{2\gamma}{n}\sum_{i=1}^n \langle g_i'(x^t), x^t - x\rangle \mathbb{1}(t \in \mathcal{N}) + \frac{2\gamma}{n}\sum_{i=1}^n \langle f_i'(x^t), x^t - x\rangle \mathbb{1}(t \in \mathcal{B})
$$

$$
\leq \|\widetilde{x}^t - x\|^2 - \|\widetilde{x}^{t+1} - x\|^2 + \frac{4\sqrt{1-\delta}}{\delta}\gamma^2 M^2 + 2\gamma^2 M^2
$$

$$
+ \frac{2\gamma}{n}\sum_{i=1}^n \langle g_i(x^t) - g_i'(x^t, \xi_i^t), x^t - x\rangle \mathbb{1}(t \in \mathcal{N}) + \frac{2\gamma}{n}\sum_{i=1}^n \langle f_i'(x^t) - f_i'(x^t, \xi_i^t), x^t - x\rangle \mathbb{1}(t \in \mathcal{B})
$$

$$
+ \frac{2\gamma^2}{n}\sum_{i=1}^n \|g_i'(x^t) - g_i'(x^t, \xi_i^t)\|^2 \mathbb{1}(t \in \mathcal{N}) + \frac{2\gamma^2}{n}\sum_{i=1}^n \|f_i'(x^t) - f_i'(x^t, \xi_i^t)\|^2 \mathbb{1}(t \in \mathcal{B}).
$$

Now we use convexity of $g_i$ and $f_i$ to derive

$$
\frac{2\gamma}{n}\sum_{i=1}^n (g_i(x^t) - g_i(x))\mathbb{1}(t \in \mathcal{N}) + \frac{2\gamma}{n}\sum_{i=1}^n (f_i(x^t) - f_i(x))\mathbb{1}(t \in \mathcal{B})
$$

$$
\leq \|\widetilde{x}^t - x\|^2 - \|\widetilde{x}^{t+1} - x\|^2 + \frac{4\sqrt{1-\delta}}{\delta}\gamma^2 M^2 + 2\gamma^2 M^2
$$

$$
+ \frac{2\gamma}{n}\sum_{i=1}^n \langle g_i(x^t) - g_i'(x^t, \xi_i^t), x^t - x\rangle \mathbb{1}(t \in \mathcal{N}) + \frac{2\gamma}{n}\sum_{i=1}^n \langle f_i'(x^t) - f_i'(x^t, \xi_i^t), x^t - x\rangle \mathbb{1}(t \in \mathcal{B})
$$

$$
+ \frac{2\gamma^2}{n}\sum_{i=1}^n \|g_i'(x^t) - g_i'(x^t, \xi_i^t)\|^2 \mathbb{1}(t \in \mathcal{N}) + \frac{2\gamma^2}{n}\sum_{i=1}^n \|f_i'(x^t) - f_i'(x^t, \xi_i^t)\|^2 \mathbb{1}(t \in \mathcal{B}).
$$

We add and subtract $\frac{1}{n}\sum_{i=1}^{n} g_i(x^t, \xi_i^t)$ to obtain

$$\frac{2\gamma}{n}\sum_{i=1}^{n}(g_i(x^t, \xi_i^t) - g_i(x))\mathbb{1}(t \in \mathcal{N}) + \frac{2\gamma}{n}\sum_{i=1}^{n}(f_i(x^t) - f_i(x))\mathbb{1}(t \in \mathcal{B})$$

$$\leq \|\widetilde{x}^t - x\|^2 - \|\widetilde{x}^{t+1} - x\|^2 + \frac{4\sqrt{1-\delta}}{\delta}\gamma^2 M^2 + 2\gamma^2 M^2$$

$$+ \frac{2\gamma}{n}\sum_{i=1}^{n}\langle g_i(x^t) - g_i'(x^t, \xi_i^t), x^t - x\rangle\mathbb{1}(t \in \mathcal{N}) + \frac{2\gamma}{n}\sum_{i=1}^{n}\langle f_i'(x^t) - f_i'(x^t, \xi_i^t), x^t - x\rangle\mathbb{1}(t \in \mathcal{B})$$

$$+ \frac{2\gamma}{n}\sum_{i=1}^{n}(g_i(x^t, \xi_i^t) - g_i(x^t))\mathbb{1}(t \in \mathcal{N})$$

$$+ \frac{2\gamma^2}{n}\sum_{i=1}^{n}\|g_i'(x^t) - g_i'(x^t, \xi_i^t)\|^2\mathbb{1}(t \in \mathcal{N}) + \frac{2\gamma^2}{n}\sum_{i=1}^{n}\|f_i'(x^t) - f_i'(x^t, \xi_i^t)\|^2\mathbb{1}(t \in \mathcal{B}).$$

Now we set $x = x^*$. Since $\frac{1}{n}\sum_{i=1}^{n} g_i(x^t, \xi_i^t) \geq c$ for $t \in \mathcal{N}$ and $g(x^*) \leq 0$ we get

$$2\gamma c\mathbb{1}(t \in \mathcal{N}) + \frac{2\gamma}{n}\sum_{i=1}^{n}(f_i(x^t) - f_i(x))\mathbb{1}(t \in \mathcal{B}) - \|\widetilde{x}^t - x\|^2 + \|\widetilde{x}^{t+1} - x\|^2 - \frac{4\sqrt{1-\delta}}{\delta}\gamma^2 M^2 - 2\gamma^2 M^2$$

$$\leq \frac{2\gamma}{n}\sum_{i=1}^{n}\langle g_i(x^t) - g_i'(x^t, \xi_i^t), x^t - x\rangle\mathbb{1}(t \in \mathcal{N}) + \frac{2\gamma}{n}\sum_{i=1}^{n}\langle f_i'(x^t) - f_i'(x^t, \xi_i^t), x^t - x\rangle\mathbb{1}(t \in \mathcal{B})$$

$$+ \frac{2\gamma}{n}\sum_{i=1}^{n}(g_i(x^t, \xi_i^t) - g_i(x^t))\mathbb{1}(t \in \mathcal{N})$$

$$+ \frac{2\gamma^2}{n}\sum_{i=1}^{n}\|g_i'(x^t) - g_i'(x^t, \xi_i^t)\|^2\mathbb{1}(t \in \mathcal{N}) + \frac{2\gamma^2}{n}\sum_{i=1}^{n}\|f_i'(x^t) - f_i'(x^t, \xi_i^t)\|^2\mathbb{1}(t \in \mathcal{B}).$$

Let us denote $\omega_i^t := g_i'(x^t) - g_i'(x^t, \xi^t)$ and $\nu_i^t := f_i'(x^t) - f_i'(x^{\backprime}\xi_i^t)$. Then we have

$$2\gamma c\mathbb{1}(t \in \mathcal{N}) + \frac{2\gamma}{n}\sum_{i=1}^{n}(f_i(x^t) - f_i(x^*))\mathbb{1}(t \in \mathcal{B}) - \|\widetilde{x}^t - x^*\|^2 + \|\widetilde{x}^{t+1} - x^*\|^2 - \frac{4\sqrt{1-\delta}}{\delta}\gamma^2 M^2 - 2\gamma^2 M^2$$

$$\leq \frac{2\gamma}{n}\sum_{i=1}^{n}\langle \omega_i^t, x^t - x^*\rangle\mathbb{1}(t \in \mathcal{N}) + \frac{2\gamma}{n}\sum_{i=1}^{n}\langle \nu_i^t, x^t - x^*\rangle\mathbb{1}(t \in \mathcal{B})$$

$$+ \frac{2\gamma}{n}\sum_{i=1}^{n}(g_i(x^t, \xi_i^t) - g_i(x^t))\mathbb{1}(t \in \mathcal{N}) + \frac{2\gamma^2}{n}\sum_{i=1}^{n}\|\omega_i^t\|^2\mathbb{1}(t \in \mathcal{N}) + \frac{2\gamma^2}{n}\sum_{i=1}^{n}\|\nu_i^t\|^2\mathbb{1}(t \in \mathcal{B}).$$

We add and subtract $\widetilde{x}^t$ in some terms to obtain

$$2\gamma c\mathbb{1}(t \in \mathcal{N}) + \frac{2\gamma}{n}\sum_{i=1}^{n}(f_i(x^t) - f_i(x^*))\mathbb{1}(t \in \mathcal{B}) - \|\widetilde{x}^t - x^*\|^2 + \|\widetilde{x}^{t+1} - x^*\|^2 - \frac{4\sqrt{1-\delta}}{\delta}\gamma^2 M^2 - 2\gamma^2 M^2$$

$$\leq \frac{2\gamma}{n}\sum_{i=1}^{n}\langle \omega_i^t, \widetilde{x}^t - x^*\rangle\mathbb{1}(t \in \mathcal{N}) + \frac{2\gamma}{n}\sum_{i=1}^{n}\langle \omega_i^t, x^t - \widetilde{x}^t\rangle\mathbb{1}(t \in \mathcal{N})$$

$$+ \frac{2\gamma}{n}\sum_{i=1}^{n}\langle \nu_i^t, \widetilde{x}^t - x^*\rangle\mathbb{1}(t \in \mathcal{B}) + \frac{2\gamma}{n}\sum_{i=1}^{n}\langle \nu_i^t, x^t - \widetilde{x}^t\rangle\mathbb{1}(t \in \mathcal{B})$$

$$+ \frac{2\gamma}{n}\sum_{i=1}^{n}(g_i(x^t, \xi_i^t) - g_i(x^t))\mathbb{1}(t \in \mathcal{N}) + \frac{2\gamma^2}{n}\sum_{i=1}^{n}\|\omega_i^t\|^2\mathbb{1}(t \in \mathcal{N}) + \frac{2\gamma^2}{n}\sum_{i=1}^{n}\|\nu_i^t\|^2\mathbb{1}(t \in \mathcal{B}).$$

Using (39) we derive

$$2\gamma c \mathbb{1}(t \in \mathcal{N}) + \frac{2\gamma}{n}\sum_{i=1}^{n}(f_i(x^t) - f_i(x^*))\mathbb{1}(t \in \mathcal{B}) - \|\widetilde{x}^t - x^*\|^2 + \|\widetilde{x}^{t+1} - x^*\|^2 - \frac{4\sqrt{1-\delta}}{\delta}\gamma^2 M^2 - 2\gamma^2 M^2$$

$$\leq \frac{2\gamma}{n}\sum_{i=1}^{n}\langle\omega_i^t, \widetilde{x}^t - x^*\rangle\mathbb{1}(t \in \mathcal{N}) + \frac{2\gamma^2}{n}\sum_{i=1}^{n}\langle\omega_i^t, e^t\rangle\mathbb{1}(t \in \mathcal{N})$$

$$+ \frac{2\gamma}{n}\sum_{i=1}^{n}\langle\nu_i^t, \widetilde{x}^t - x^*\rangle\mathbb{1}(t \in \mathcal{B}) + \frac{2\gamma^2}{n}\sum_{i=1}^{n}\langle\nu_i^t, e^t\rangle\mathbb{1}(t \in \mathcal{B})$$

$$+ \frac{2\gamma}{n}\sum_{i=1}^{n}(g_i(x^t, \xi_i^t) - g_i(x^t))\mathbb{1}(t \in \mathcal{N}) + \frac{2\gamma^2}{n}\sum_{i=1}^{n}\|\omega_i^t\|^2\mathbb{1}(t \in \mathcal{N}) + \frac{2\gamma^2}{n}\sum_{i=1}^{n}\|\nu_i^t\|^2\mathbb{1}(t \in \mathcal{B}).$$

Since $\|\omega_i^t\|, \|\nu_i^t\| \leq 2M$ we get from Lemma F.2

$$2\gamma c \mathbb{1}(t \in \mathcal{N}) + \frac{2\gamma}{n}\sum_{i=1}^{n}(f_i(x^t) - f_i(x^*))\mathbb{1}(t \in \mathcal{B}) - \|\widetilde{x}^t - x^*\|^2 + \|\widetilde{x}^{t+1} - x^*\|^2 - \frac{4\sqrt{1-\delta}}{\delta}\gamma^2 M^2 - 2\gamma^2 M^2$$

$$\leq \frac{2\gamma}{n}\sum_{i=1}^{n}\langle\omega_i^t, \widetilde{x}^t - x^*\rangle\mathbb{1}(t \in \mathcal{N}) + \frac{2\gamma^2}{n}\sum_{i=1}^{n}2M \cdot \frac{2\sqrt{1-\delta}}{\delta}M\mathbb{1}(t \in \mathcal{N})$$

$$+ \frac{2\gamma}{n}\sum_{i=1}^{n}\langle\nu_i^t, \widetilde{x}^t - x^*\rangle\mathbb{1}(t \in \mathcal{B}) + \frac{2\gamma^2}{n}\sum_{i=1}^{n}2M \cdot \frac{2\sqrt{1-\delta}}{\delta}M\mathbb{1}(t \in \mathcal{B}) + \frac{2\gamma}{n}\sum_{i=1}^{n}(g_i(x^t, \xi_i^t) - g_i(x^t))\mathbb{1}(t \in \mathcal{N})$$

$$+ \frac{2\gamma^2}{n}\sum_{i=1}^{n}\|\omega_i^t\|^2\mathbb{1}(t \in \mathcal{N}) + \frac{2\gamma^2}{n}\sum_{i=1}^{n}\|\nu_i^t\|^2\mathbb{1}(t \in \mathcal{B}).$$

Rearranging terms, we obtain

$$2\gamma c \mathbb{1}(t \in \mathcal{N}) + \frac{2\gamma}{n}\sum_{i=1}^{n}(f_i(x^t) - f_i(x^*))\mathbb{1}(t \in \mathcal{B}) - \|\widetilde{x}^t - x^*\|^2 + \|\widetilde{x}^{t+1} - x^*\|^2 - \frac{12\sqrt{1-\delta}}{\delta}\gamma^2 M^2 - 2\gamma^2 M^2$$

$$\leq \frac{2\gamma}{n}\sum_{i=1}^{n}\langle\omega_i^t, \widetilde{x}^t - x^*\rangle\mathbb{1}(t \in \mathcal{N}) + \frac{2\gamma}{n}\sum_{i=1}^{n}\langle\nu_i^t, \widetilde{x}^t - x^*\rangle\mathbb{1}(t \in \mathcal{B}) + \frac{2\gamma}{n}\sum_{i=1}^{n}(g_i(x^t, \xi_i^t) - g_i(x^t))\mathbb{1}(t \in \mathcal{N})$$

$$+ \frac{2\gamma^2}{n}\sum_{i=1}^{n}\|\omega_i^t\|^2\mathbb{1}(t \in \mathcal{N}) + \frac{2\gamma^2}{n}\sum_{i=1}^{n}\|\nu_i^t\|^2\mathbb{1}(t \in \mathcal{B}).$$

Now we define

$$A_t := 2\gamma c \mathbb{1}(t \in \mathcal{N}) + \frac{2\gamma}{n}\sum_{i=1}^{n}(f_i(x^t) - f_i(x^*))\mathbb{1}(t \in \mathcal{B}) - \|\widetilde{x}^t - x^*\|^2 + \|\widetilde{x}^{t+1} - x^*\|^2 - \frac{12\sqrt{1-\delta}}{\delta}\gamma^2 M^2 - 2\gamma^2 M^2. \quad (42)$$

In the case $t \in \mathcal{N}$, we have

$$A_t = \frac{2\gamma c}{n} - \|\widetilde{x}^t - x^*\|^2 + \|\widetilde{x}^{t+1} - x^*\|^2 - \frac{12\sqrt{1-\delta}}{\delta}\gamma^2 M^2 - 2\gamma^2 M^2$$

$$\leq \frac{2\gamma}{n}\sum_{i=1}^{n}\langle\omega_i^t, \widetilde{x}^t - x^*\rangle + \frac{2\gamma}{n}\sum_{i=1}^{n}(g_i(x^t, \xi_i^t) - g_i(x^t)) + \frac{2\gamma^2}{n}\sum_{i=1}^{n}\|\omega_i^t\|^2.$$

In the case $t \in \mathcal{B}$, we have

$$A_t = \frac{2\gamma}{n}\sum_{i=1}^{n}(f_i(x^t) - f_i(x^*)) - \|\widetilde{x}^t - x^*\|^2 + \|\widetilde{x}^{t+1} - x^*\|^2 - \frac{12\sqrt{1-\delta}}{\delta}\gamma^2 M^2 - 2\gamma^2 M^2$$

$$\leq \frac{2\gamma}{n}\sum_{i=1}^{n}\langle\nu_i^t, \widetilde{x}^t - x^*\rangle + \frac{2\gamma^2}{n}\sum_{i=1}^{n}\|\nu_i^t\|^2.$$

Following Liu et al. (2023) we define $Z_t$ as follows

$$Z_t := w_t A_t - v_t \|\widetilde{x}^t - x^*\|^2, \quad (43)$$

where $w_t$ and $v_t$ will be defined later. Next, we define

$$S_t := \sum_{l=t}^{T-1} Z_t. \tag{44}$$

Let us define the natural filtration $\mathcal{F}_t := \sigma(\xi_0, \ldots, \xi_{t-1})$. We will show by induction that

$$\mathbb{E}\left[\exp(S_t) \mid \mathcal{F}_t\right] \leq \exp\left(48M^2 \sum_{l=t}^{T-1} w_l \gamma^2 + \frac{8\sigma_{\mathrm{fv}}^2}{nN_{\mathrm{fv}}} \sum_{l=t}^{T-1} w_l^2 \gamma^2\right).$$

The base of induction is trivial for $t = T$ since $S_T = 0$. Assume that the statement holds for $t \in \{0, \ldots, T-1\}$. We have
$$\mathbb{E}\left[\exp(S_t) \mid \mathcal{F}_t\right] = \mathbb{E}\left[\exp(S_{t+1} + Z_t) \mid \mathcal{F}_t\right]$$
$$= \mathbb{E}\left[\mathbb{E}\left[\exp(S_{t+1} + Z_t \mid \mathcal{F}_{t+1}\right] \mid \mathcal{F}_t\right].$$
We now analyze the inner expectation. Conditioned on $\mathcal{F}_{t+1}$ we have $Z_t$ fixed. Using the inductive hypothesis, we derive

$$\mathbb{E}\left[\exp(Z_t + S_{t+1}) \mid \mathcal{F}_{t+1}\right] \leq \exp(Z_t) \exp\left(48M^2 \sum_{l=t+1}^{T-1} w_l \gamma^2\right).$$

Therefore,

$$\mathbb{E}\left[\exp(Z_t + S_{t+1}) \mid \mathcal{F}_t\right] \leq \mathbb{E}\left[\exp(Z_t) \mid \mathcal{F}_t\right] \exp\left(48M^2 \sum_{l=t+1}^{T-1} w_l \gamma^2\right). \tag{45}$$

From (42), (43), and assuming that $t \in \mathcal{N}$ we have the following bound

$$\exp(Z_t) = \exp\left(w_t \frac{2\gamma c}{n} - w_t \|\widetilde{x}^t - x^*\|^2 + w_t \|\widetilde{x}^{t+1} - x^*\|^2 - w_t\left(2 + \frac{12\sqrt{1-\delta}}{\delta}\right)\gamma^2 M^2 - v_t \|\widetilde{x}^t - x^*\|^2\right)$$

$$\leq \exp\left(\frac{2\gamma w_t}{n} \sum_{i=1}^n \langle \omega_i^t, \widetilde{x}^t - x^* \rangle + \frac{2\gamma^2 w_t}{n} \sum_{i=1}^n \|\omega_i^t\|^2 + \frac{2\gamma w_t}{n} \sum_{i=1}^n (g_i(x^t, \xi_i^t) - g_i(x^t))\right) \exp(-v_t \|\widetilde{x}^t - x^*\|^2).$$

Next, we use Lemma 2.2 from Liu et al. (2023) (with $a = \frac{2\gamma w_t}{n}(\widetilde{x}^t - x^*)$ and $b^2 = \frac{2\gamma^2 w_t}{n}$ for the terms with $\omega_i^t$, and with $a = \frac{2\gamma w_t}{n} \cdot 1$ for the terms with $g_i(x^t, \xi_i^t) - g_i(x^t)$) and independence of function and subgradient evaluations

$$\mathbb{E}\left[\exp\left(\frac{2\gamma w_t}{n} \sum_{i=1}^n \langle \omega_i^t, \widetilde{x}^t - x^* \rangle + \frac{2\gamma^2 w_t}{n} \sum_{i=1}^n \|\omega_i^t\|^2 + \frac{2\gamma w_t}{n} \sum_{i=1}^n (g_i(x^t, \xi_i^t) - g_i(x^t))\right) \mid \mathcal{F}_t, t \in \mathcal{N}\right]$$

$$\leq \exp\left(n \cdot \left[3\left\{\frac{4\gamma^2 w_t^2}{n^2} \cdot 4M^2 \|\widetilde{x}^t - x^*\|^2 + \frac{2\gamma^2 w_t}{n} \cdot 4M^2\right\} + 2\frac{4\gamma^2 w_t^2}{n^2} \frac{\sigma_{\mathrm{fv}}^2}{N_{\mathrm{fv}}}\right]\right)$$

$$= \exp\left(\frac{48\gamma^2 w_t^2}{n} M^2 \|\widetilde{x}^t - x^*\|^2 + 24\gamma^2 w_t M^2 + \frac{8\gamma^2 w_t^2}{n} \frac{\sigma_{\mathrm{fv}}^2}{N_{\mathrm{fv}}}\right). \tag{46}$$

Therefore, from (45) we derive using the definition of $v_t := \frac{48\gamma^2 w_t^2}{n} M^2$

$$\mathbb{E}\left[\exp(S_t) \mid \mathcal{F}_t\right] \leq \exp\left(\left[\frac{48\gamma^2 w_t^2}{n} M^2 - v_t\right] \|\widetilde{x}^t - x^*\|^2 + 24M^2 \sum_{l=t}^{T-1} w_l \gamma^2 + \frac{8\sigma_{\mathrm{fv}}^2}{nN_{\mathrm{fv}}} \sum_{l=t}^{T-1} w_l^2 \gamma^2\right)$$

$$= \exp\left(48M^2 \sum_{l=t}^{T-1} w_l \gamma^2 + \frac{8\sigma_{\mathrm{fv}}^2}{nN_{\mathrm{fv}}} \sum_{l=t}^{T-1} w_l^2 \gamma^2\right).$$

This concludes the transition step in the case $t \in \mathcal{N}$.

Now we move on to the case $t \in \mathcal{B}$. The derivations are similar, but we do not have function values. Therefore, instead of

$$\exp\left(\frac{48\gamma^2 w_t^2}{n} M^2 \|\widetilde{x}^t - x^*\|^2 + 24M^2 w_t \gamma^2 + \frac{8\sigma_{\mathrm{fv}}^2}{nN_{\mathrm{fv}}} w_t^2 \gamma^2\right)$$

in (46) we get

$$\exp\left(\frac{48\gamma^2 w_t^2}{n} M^2 \|\widetilde{x}^t - x^*\|^2 + 24M^2 w_t \gamma^2\right).$$

Therefore, the transition step holds in both cases. $\qquad\square$

**Corollary F.4.** *Let $\beta \in (0, 1)$ be a failure probability. Suppose the sequence $\{w_t\}$ satisfy the restrictions of Theorem F.3*

*and* $w_t + \underbrace{\frac{48\gamma^2 w_t^2}{n} M^2}_{=v_t} \leq w_{t-1}$. *Let the stepsize* $\gamma = \frac{\tilde{\gamma}}{\sqrt{T}}$. *Then with probability at least* $1 - \beta$

$$\sum_{t \in \mathcal{N}} \gamma c + \sum_{t \in \mathcal{B}} \gamma(f(x^t) - f(x^*)) \leq C_1 \log \frac{1}{\beta} + \|\tilde{x}^0 - x^*\|^2 + \gamma^2 M^2 \left(50 + \frac{12\sqrt{1-\delta}}{\delta}\right) T$$

$$+ \frac{8\sigma_{\text{fv}}^2}{C_1 n N_{\text{fv}}} T\gamma^2,$$

*where* $C_1 := \frac{48\tilde{\gamma}^2 M^2}{n}$.

*Proof.* Let $T = 48M^2 \sum_{t=0}^{T-1} w_t \gamma^2 + \frac{8\sigma_{\text{fv}}^2}{nN_{\text{fv}}} \sum_{t=0}^{T-1} w_t^2 \gamma^2 + \log \frac{1}{\beta}$. By Theorem F.3 and Markov's inequality, we have

$$\mathbb{P}(S_0 \geq T) \leq \mathbb{P}(\exp(S_0) \geq \exp(T))$$

$$\leq \exp(-T)\mathbb{E}\left[\exp(S_0)\right]$$

$$\leq \exp(-T) \exp\left(48M^2 \sum_{t=0}^{T-1} \gamma^2 w_t + \frac{8\sigma_{\text{fv}}^2}{nN_{\text{fv}}} \sum_{t=0}^{T-1} \gamma^2 w_t^2\right)$$

$$= \beta.$$

Note that since $w_t + v_t \leq w_{t-1}$ by the assumption of the lemma

$$S_0 = \sum_{t=0}^{T-1} Z_t$$

$$= \sum_{t=0}^{T-1} \left[w_t \left(2\gamma c \mathbb{1}(t \in \mathcal{N}) + 2\gamma(f(x^t) - f(x^*))\mathbb{1}(t \in \mathcal{B})\right) - (v_t + w_t)\|\tilde{x}^t - x^*\|^2 + w_t\|\tilde{x}^{t+1} - x^*\|^2\right.$$

$$\left.- w_t \left(2 + \frac{12\sqrt{1-\delta}}{\delta}\right)\gamma^2 M^2\right]$$

$$\geq \sum_{t=0}^{T-1} \left[2\gamma w_t \left(c\mathbb{1}(t \in \mathcal{N}) + (f(x^t) - f(x^*))\mathbb{1}(t \in \mathcal{B})\right) - \sum_{t=0}^{T-1} \left(w_{t-1}\|\tilde{x}^t - x^*\|^2 - w_t\|\tilde{x}^{t+1} - x^*\|^2\right)\right.$$

$$\left.- \sum_{t=0}^{T-1} w_t \left(2 + \frac{12\sqrt{1-\delta}}{\delta}\right)\gamma^2 M^2\right]$$

$$\geq \sum_{t=0}^{T-1} 2\gamma w_t \left(c\mathbb{1}(t \in \mathcal{N}) + (f(x^t) - f(x^*))\mathbb{1}(t \in \mathcal{B})\right) - w_0\|x^0 - x^*\|^2 + w_{T-1}\|\tilde{x}^T - x^*\|^2$$

$$- \sum_{t=0}^{T-1} w_t \left(2 + \frac{12\sqrt{1-\delta}}{\delta}\right)\gamma^2 M^2$$

$$\geq \sum_{t=0}^{T-1} 2\gamma w_t \left(c\mathbb{1}(t \in \mathcal{N}) + (f(x^t) - f(x^*))\mathbb{1}(t \in \mathcal{B})\right) - w_0\|x^0 - x^*\|^2 + w_{T-1}\|\tilde{x}^T - x^*\|^2$$

$$- \sum_{t=0}^{T-1} w_t \left(2 + \frac{12\sqrt{1-\delta}}{\delta}\right)\gamma^2 M^2.$$

Therefore, with a probability of at least $1 - \beta$ we have

$$\sum_{t \in \mathcal{N}} 2\gamma w_t c + \sum_{t \in \mathcal{B}} 2\gamma w_t (f(x^t) - f(x^*)) + w_{T-1}\|\widetilde{x}^T - x^*\|^2$$

$$\leq S_0 + w_0\|x^0 - x^*\|^2 + \sum_{t=0}^{T-1} w_t \left(2 + \frac{12\sqrt{1-\delta}}{\delta}\right)\gamma^2 M^2$$

$$\leq \log\frac{1}{\beta} + w_0\|x^0 - x^*\|^2 + \gamma^2\left(48M^2 + 2M^2 + \frac{12\sqrt{1-\delta}}{\delta}M^2\right)\sum_{t=0}^{T-1} w_t + \frac{8\sigma_{\text{fv}}^2}{nN_{\text{fv}}}\sum_{t=0}^{T-1} w_t^2\gamma^2.$$

We need to satisfy the following restrictions on $w_t$:

$$w_t \leq \frac{n}{32\gamma^2 M^2}$$

$$w_t + \frac{48\gamma^2}{n}w_t^2 \leq w_{t-1}.$$

Let

$$C_1 := \frac{48\widetilde{\gamma}^2 M^2}{n}. \tag{47}$$

Then we set $w_{T-1} = \frac{1}{C_1 + \frac{48\widetilde{\gamma}^2 M^2}{n}} = \frac{1}{2C_1}$. Next, we set $w_{t-1}$ such that the second inequality holds with equality, namely,

$$w_{t-1} = w_t + \frac{48\gamma^2 M^2}{n}w_t^2 = w_t + \frac{C_1}{T}w_t^2.$$

We can show by induction that $w_t \leq \frac{1}{C_1 + \frac{C_1}{T}t}$. Indeed, the base of induction holds by the choice of $w_{T-1}$. Assume it holds at $t$, let us show that it holds at $t - 1$ as well:

$$w_{t-1} = w_t + \frac{C_1}{T}w_t^2$$

$$\leq \frac{1}{C_1 + \frac{C_1}{T}t} + \frac{C_1}{T(C_1 + \frac{C_1}{T}t)^2}$$

$$\leq \frac{1}{C_1 + \frac{C_1}{T}t} + \frac{(C_1 + \frac{C_1}{T}t) - (C_1 + \frac{C_1}{T}(t-1))}{(C_1 + \frac{C_1}{T}(t-1))(C_1 + \frac{C_1}{T}t)}$$

$$= \frac{1}{C_1 + \frac{C_1}{T}t}\left(\frac{C_1 + \frac{C_1}{T}(t-1)}{C_1 + \frac{C_1}{T}(t-1)} + \frac{C_1 + \frac{C_1}{T}t - (C_1 + \frac{C_1}{T}(t-1))}{C_1 + \frac{C_1}{T}(t-1)}\right) = \frac{1}{C_1 + \frac{C_1}{T}(t-1)}.$$

Now we show that the first condition is satisfied as well

$$w_t\gamma^2 = w_t\frac{\widetilde{\gamma}^2}{T} \leq \frac{1}{\frac{C_1}{T}t}\frac{\widetilde{\gamma}^2}{T} = \frac{1}{\frac{48\widetilde{\gamma}^2 M^2}{n}t}\widetilde{\gamma}^2 = \frac{n}{48M^2 t} \leq \frac{n}{32M^2}.$$

Therefore, with a probability at least $1 - \beta$, we have

$$\sum_{t \in \mathcal{N}} 2\gamma w_t c + \sum_{t \in \mathcal{B}} 2\gamma w_t (f(x^t) - f(x^*)) + w_{T-1}\|\widetilde{x}^T - x^*\|^2$$

$$\leq \log\frac{1}{\beta} + w_0\|x^0 - x^*\|^2 + \gamma^2\left(48M^2 + 2M^2 + \frac{12\sqrt{1-\delta}}{\delta}M^2\right)\sum_{t=0}^{T-1} w_t + \frac{8\sigma_{\text{fv}}^2}{nN_{\text{fv}}}\sum_{t=0}^{T-1} w_t^2\gamma^2. \tag{48}$$

Since $w_{T-1} = \frac{1}{2C_1}$ and $\frac{1}{2C_1} \leq w_t \leq \frac{1}{C_1}$ we have with probability at least $1 - \beta$

$$\frac{1}{C_1}\sum_{t \in \mathcal{N}}\gamma c + \frac{1}{C_1}\sum_{t \in \mathcal{B}}\gamma(f(x^t) - f(x^*))$$

$$\leq \log\frac{1}{\beta} + \frac{1}{C_1}\|x^0 - x^*\|^2 + \gamma^2\left(50M^2 + \frac{12\sqrt{1-\delta}}{\delta}M^2\right)\sum_{t=0}^{T-1} w_t + \frac{8\sigma_{\text{fv}}^2}{nN_{\text{fv}}}\sum_{t=0}^{T-1} w_t^2\gamma^2. \tag{49}$$

We estimate the sums $\sum_{t=0}^{T-1} w_t \leq \frac{T}{C_1}$ and $\sum_{t=0}^{T-1} w_t^2 \leq \frac{T}{C_1^2}$. Therefore, we derive

$$\frac{1}{C_1} \sum_{t \in \mathcal{N}} \gamma c + \frac{1}{C_1} \sum_{t \in \mathcal{B}} \gamma(f(x^t) - f(x^*))$$

$$\leq \log \frac{1}{\beta} + \frac{1}{C_1} \|x^0 - x^*\|^2 + \gamma^2 M^2 \left(50 + \frac{12\sqrt{1-\delta}}{\delta}\right) \frac{T}{C_1} + \frac{8\sigma_{\text{fv}}^2}{nN_{\text{fv}}} \frac{T}{C_1^2} \gamma^2. \tag{50}$$

Canceling $C_1$ in both sides, we finally obtain

$$\sum_{t \in \mathcal{N}} \gamma c + \sum_{t \in \mathcal{B}} \gamma(f(x^t) - f(x^*))$$

$$\leq C_1 \log \frac{1}{\beta} + \|x^0 - x^*\|^2 + \gamma^2 M^2 \left(50 + \frac{12\sqrt{1-\delta}}{\delta}\right) T + \frac{8\sigma_{\text{fv}}^2}{C_1 nN_{\text{fv}}} T\gamma^2. \tag{51}$$

$\square$

**Lemma F.5.** *Let $\beta \in (0, 1)$ be the failure probability and $C_1$ be defined as in(47). Suppose that the stepsize $\gamma = \frac{\tilde{\gamma}}{\sqrt{T}}$ and threshold $c$ satisfy*

$$\frac{T}{2}\gamma c > C_1 \log \frac{1}{\beta} + \|x^0 - x^*\|^2 + \gamma^2 M^2 \left(50 + \frac{12\sqrt{1-\delta}}{\delta}\right) T + \frac{8\sigma_{\text{fv}}^2}{C_1 nN_{\text{fv}}} T\gamma^2. \tag{52}$$

*Then we have with probability at least $1 - \beta$*

$$\sum_{t \in \mathcal{N}} \gamma c + \sum_{t \in \mathcal{B}} \gamma(f(x^t) - f(x^*))$$

$$\leq C_1 \log \frac{1}{\beta} + \|x^0 - x^*\|^2 + \gamma^2 M^2 \left(50 + \frac{12\sqrt{1-\delta}}{\delta}\right) T + \frac{8\sigma_{\text{fv}}^2}{C_1 nN_{\text{fv}}} T\gamma^2. \tag{53}$$

*Moreover, assume that (53) holds. Then $\mathcal{B}$ is non-empty, i.e. $\bar{x}^T = \frac{1}{|\mathcal{B}|} \sum_{t \in \mathcal{B}} x^t$ is well-defined, and one of the following conditions holds*

1. *$|\mathcal{B}| \geq \frac{T}{2}$, or*

2. *$\gamma \sum_{t \in \mathcal{B}} f(x^t) - f(x^*) \leq 0$.*

*Proof.* Assume that $\mathcal{B} = \emptyset$. Then from Corollary F.4 we have that with probability at least $1 - \beta$ we have

$$T\gamma c \leq C_1 \log \frac{1}{\beta} + \|x^0 - x^*\|^2 + \gamma^2 M^2 \left(50 + \frac{12\sqrt{1-\delta}}{\delta}\right) T + \frac{8\sigma_{\text{fv}}^2}{C_1 nN_{\text{fv}}} T\gamma^2,$$

This contradicts the assumption of the lemma. Hence, we must have $\mathcal{B} \neq \emptyset$. Now assume that (53) holds. If we have $\gamma \sum_{t \in \mathcal{B}} f(x^t) - f(x^*) \leq 0$, then the second condition holds. Assume that $\gamma \sum_{t \in \mathcal{B}} f(x^t) - f(x^*) > 0$, then from (53) we obtain

$$\sum_{t \in \mathcal{N}} \gamma c \leq C_1 \log \frac{1}{\beta} + \|x^0 - x^*\|^2 + \gamma^2 M^2 \left(50 + \frac{12\sqrt{1-\delta}}{\delta}\right) T + \frac{8\sigma_{\text{fv}}^2}{C_1 nN_{\text{fv}}} T\gamma^2.$$

Assume that $|\mathcal{B}| < \frac{T}{2}$, this means that $|\mathcal{N}| \geq \frac{T}{2}$. Therefore, we have

$$\frac{T}{2}\gamma c \leq \sum_{t \in \mathcal{N}} \gamma c \leq C_1 \log \frac{1}{\beta} + \|x^0 - x^*\|^2 + \gamma^2 M^2 \left(50 + \frac{12\sqrt{1-\delta}}{\delta}\right) T + \frac{8\sigma_{\text{fv}}^2}{C_1 nN_{\text{fv}}} T\gamma^2,$$

which contradicts (52). Hence, if $\gamma \sum_{t \in \mathcal{B}} (f(x^t) - f(x^*)) > 0$, then $|\mathcal{B}| \geq \frac{T}{2}$.

$\square$

Now we are ready to establish our main convergence result in the stochastic setting.

**Theorem F.6.** *Let $\beta \in (0, 1)$ be the failure probability and $C_1$ be defined as in (47). Suppose that the choice of $\gamma$ and $c$ are chosen such that (52) holds. Then we have with a probability of at least $1 - \beta$ that*

$$f(\bar{x}^T) - f(x^*) \leq \frac{2C_1 \log \frac{1}{\beta} + 2\|x^0 - x^*\|^2}{\gamma T} + 2\gamma M^2 \left(50 + \frac{12\sqrt{1-\delta}}{\delta}\right) + \frac{16\sigma_{\text{fv}}^2}{C_1 nN_{\text{fv}}}\gamma.$$

*Proof.* We start by using the results Lemma F.5. Using the convexity of $f$ and Jensen's inequality we get that if part 2. holds, then with a probability of at least $1 - \beta$ we have

$$f(\overline{x}^T) - f(x^*) \leq 0.$$

If part 2. does not hold, then $|\mathcal{B}| \geq \frac{T}{2}$. Therefore, from (52) we obtain

$$f(\overline{x}^T) - f(x^*) \leq \frac{2}{\gamma T}\left(C_1 \log\frac{1}{\beta} + \|x^0 - x^*\|^2 + \gamma^2 M^2\left(50 + \frac{12\sqrt{1-\delta}}{\delta}\right) + \frac{8\sigma_{\mathrm{fv}}^2}{C_1 n N_{\mathrm{fv}}}T\gamma^2\right)$$

$$= \frac{2C_1\log\frac{1}{\beta} + 2\|x^0 - x^*\|^2}{\gamma T} + 2\gamma M^2\left(50 + \frac{12\sqrt{1-\delta}}{\delta}\right)T + \frac{16\sigma_{\mathrm{fv}}^2}{C_1 n N_{\mathrm{fv}}}\gamma.$$

$\square$

**Corollary F.7.** *Let* $\beta \in (0, 1/2)$ *be the failure probability. Let*

$$R^2 \geq \|x^0 - x^*\|^2 + \frac{\sigma_{\mathrm{fv}}^2/N_{\mathrm{fv}}}{6M^2}.$$

*If* $\gamma = \frac{\widetilde{\gamma}}{\sqrt{T}} = \frac{R\sqrt{\delta}}{M\sqrt{T}}$, *i.e.,* $\widetilde{\gamma} = \frac{R\sqrt{\delta}}{M}$ *and* $c = \frac{128RM(1+\log 1/\beta)}{\sqrt{\delta T}}$, *then we have with a probability of at least* $1 - 2\beta$

$$f(\overline{x}^T) - f(x^*) \leq \frac{MR}{\sqrt{\delta T}}\left(48\log\frac{1}{\beta} + 128\right),$$

$$g(\overline{x}^T) \leq \frac{256RM(1 + \log 1/\beta)}{\sqrt{\delta T}}.$$

*Proof.* First, we check that the stepsize $\gamma$ and threshold $c$ satisfy (52). We have with $C_1 = \frac{48\widetilde{\gamma}^2 M^2}{n}$

$$\frac{48\frac{R^2\delta}{M^2}M^2}{n}\log\frac{1}{\beta} + \|x^0 - x^*\|^2 + \frac{R^2\delta}{M^2 T}M^2\left(50 + \frac{12}{\delta}\right)T + \frac{8\sigma_{\mathrm{fv}}^2}{nN_{\mathrm{fv}}}\frac{n}{48\frac{R^2\delta}{M^2}M^2}T\frac{R^2\delta}{M^2 T}$$

$$\leq \frac{48R^2\delta M^2}{nM^2}\log\frac{1}{\beta} + \|x^0 - x^*\|^2 + 50R^2\delta + 12R^2 + \frac{\sigma_{\mathrm{fv}}^2/N_{\mathrm{fv}}}{6M^2}$$

$$\leq \frac{48R^2\delta}{n}\log\frac{1}{\beta} + \|x^0 - x^*\|^2 + 62R^2 + R^2$$

$$\leq 64R^2\log\frac{1}{\beta} + 64R^2.$$

At the same time, we have

$$\frac{T}{2}\gamma c = \frac{T}{2}\frac{R\sqrt{\delta}}{M\sqrt{T}}\frac{128RM(1+\log 1/\beta)}{\sqrt{\delta T}} = 64R^2(1 + \log 1/\beta).$$

Therefore, with a probability of at least $1 - \beta$ we have

$$f(\overline{x}^T) - f(x^*) \leq \frac{48R^2\delta\log\frac{1}{\beta} + 2\|x^0 - x^*\|^2}{T}\frac{M\sqrt{T}}{R\sqrt{\delta}} + 2\frac{R\sqrt{\delta}}{M\sqrt{T}}M^2\left(50 + \frac{12}{\delta}\right) + \frac{16\sigma_{\mathrm{fv}}^2/N_{\mathrm{fv}}}{48\frac{R^2\delta}{M^2}n}\frac{R\sqrt{\delta}}{M\sqrt{T}}$$

$$= (48\delta\log\frac{1}{\beta} + 2)\frac{MR}{\sqrt{\delta T}} + 100\frac{RM\sqrt{\delta}}{M\sqrt{T}} + 24\frac{RM}{\sqrt{\delta T}} + 2\frac{MR\sqrt{\delta}}{\sqrt{\delta T}}$$

$$= \frac{MR}{\sqrt{\delta T}}\left(48\log\frac{1}{\beta} + 128\right). \tag{54}$$

For the constraint violation we have that

$$g(\overline{x}^T) \leq \frac{1}{|\mathcal{B}|}\sum_{t\in\mathcal{B}}g(x^t) \leq \max_{t\in\mathcal{B}}g(x^t).$$

Moreover, from (15) and Lemma F.1 we have

$$\mathbb{P}\left(\left|\sum_{i=1}^n g_i(x^t) - g_i(x^t, \xi_i^t)\right| > (\sqrt{2} + \sqrt{2}b)\sqrt{\sum_{i=1}^n \frac{\sigma_{\mathrm{fv}}^2}{N_{\mathrm{fv}}}}\right) \leq \exp(-b^2/3).$$

This implies that

$$\mathbb{P}\left(g(x^t) > \frac{1}{n}\sum_{i=1}^{n} g_i(x^t, \xi_i^t) + (\sqrt{2} + \sqrt{2}b)\frac{\sigma_{\text{fv}}}{\sqrt{nN_{\text{fv}}}}\right) \leq \exp(-b^2/3).$$

' Since for $t \in \mathcal{B}$ we have $\frac{1}{n}\sum_{i=1}^{n} g_i(x^t, \xi_i^t) \leq c$, then we get

$$\mathbb{P}\left(g(\overline{x}^T) \leq c + (\sqrt{2} + \sqrt{2}b)\frac{\sigma_{\text{fv}}}{\sqrt{nN_{\text{fv}}}}\right) \geq 1 - T\exp(-b^2/3).$$

Choosing $b^2 = 3\log\frac{T}{\beta}$ we obtain

$$\mathbb{P}\left(g(\overline{x}^T) \leq c + (\sqrt{2} + \sqrt{2}b)\frac{\sigma_{\text{fv}}}{\sqrt{nN_{\text{fv}}}}\right) \geq 1 - \beta.$$

Now we choose $N_{\text{fv}} \geq (\sqrt{2} + \sqrt{2}b)^2\frac{\sigma_{\text{fv}}^2}{nc^2}$ we obtain

$$\mathbb{P}\left(g(\overline{x}^T) \leq 2c\right) \geq 1 - \beta. \tag{55}$$

Thus with probability at least $1 - 2\beta$ we have both (54) and (55) hold. The batch-size $N_{\text{fv}}$ depends on the problem constants as follows

$$N_{\text{fv}} \geq (\sqrt{2} + \sqrt{2}b)\frac{\sigma_{\text{fv}}^2}{nc^2} = \widetilde{\mathcal{O}}\left(\frac{\sigma_{\text{fv}}^2}{n\frac{R^2M^2}{\delta T}}\right) = \widetilde{\mathcal{O}}\left(\frac{\sigma_{\text{fv}}^2\delta T}{nR^2M^2}\right).$$

The number of iterations of Safe-EF to converge to $\varepsilon$-accuracy is

$$T = \widetilde{\mathcal{O}}\left(\frac{R^2M^2}{\delta\varepsilon^2}\right).$$

Therefore, the batch-size required in the stochastic setting is of order

$$N_{\text{fv}} \geq \widetilde{\mathcal{O}}\left(\frac{\sigma_{\text{fv}}^2\delta\frac{R^2M^2}{\delta\varepsilon^2}}{nR^2M^2}\right) = \widetilde{\mathcal{O}}\left(\frac{\sigma_{\text{fv}}^2}{n\varepsilon^2}\right).$$

This concludes the proof.

$\square$

## G. Primal-dual Methods

**A short primer on primal-dual methods.** In Section 1, we briefly mentioned the primal-dual approach to solving the constrained problem (1), (4), here we elaborate more on this direction. Consider the Lagrangian with non-negative multiplier $\lambda$:

$$\mathcal{L}(x, \lambda) := f(x) + \lambda \, g(x) = \frac{1}{n} \sum_{i=1}^{n} f_i(x) + \frac{\lambda}{n} \sum_{i=1}^{n} g_i(x).$$

Primal-dual schemes aim to find the saddle-point of this Lagrangian. If Slater's conditions hold, i.e., $f(x)$ is convex and there exists a strictly feasible solution $g(x) < 0$, then the strong duality holds, that is

$$\min_x \max_{\lambda \geq 0} \mathcal{L}(x, \lambda) = \max_{\lambda \geq 0} \min_x \mathcal{L}(x, \lambda),$$

and general purpose methods for minimizing the primal-dual gap, $\mathrm{Gap}(x^t, \lambda^t) := \max_{\lambda \geq 0} \mathcal{L}(\lambda, x^t) - \min_x \mathcal{L}(\lambda^t, x)$, can be used. The basic variant of such a scheme is Gradient Descent Ascent:

$$\text{Primal-dual} \qquad \begin{aligned} x^{t+1} &= x^t - \gamma_t \left( f'(x^t) + \lambda^t g'(x^t) \right), \\ \lambda^{t+1} &= \Pi_{\lambda \geq 0}(\lambda^t + \eta_t \, g(x^{t+1})), \end{aligned} \qquad (56)$$

where $\{\gamma_t\}$, $\{\eta_t\}$ are primal and dual stepsizes respectively, and $\Pi_{\lambda \geq 0}$ denotes the projection onto the non-negative ray. Similarly to the design of Safe-EF, we can write down an error feedback variant of this method for distributed optimization Algorithm 2. The intuitive justification of Algorithm 2 is similar to that of Safe-EF in Appendix D. However, a rigorous convergence analysis of $\mathrm{Gap}(x^t, \lambda^t)$ for Algorithm 2 remains open since even the analysis of (56) (special case of Algorithm 2 in case of no compression) typically requires the projection step in $x^t$ variable. This is problematic for EF analysis because the virtual iterates $\hat{x}^t$ defined in (22) do not have such simple form anymore.

---

**Algorithm 2** Primal-dual Error Feedback for Constrained Optimization with Bidirectional Compression

1: **Input:** initial point $x^0, \lambda^0 \in \mathbb{R}^d$, stepsizes $\{\gamma_t\}$, $\{\eta_t\}$, compressors $\mathcal{C}$ and $\mathcal{C}_s$ at the workers and the server
2: **for** $t = 0, \ldots, T-1$ **do**
3:     **for** $i = 1, \ldots, n$ **do**
4:         Compute $h_i^t = f_i'(x^t) + \lambda_t g_i'(x^t)$
5:         Compute $v_i^t = \mathcal{C}(e_i^t + h_i^t)$ and send to server
6:         Compute $e_i^{t+1} = e_i^t + h_i^t - v_i^t$
7:     **end for**
8:     Compute $v^t = \frac{1}{n} \sum_{i=1}^{n} v_i^t$
9:     Compute $w^{t+1} = w^t - \gamma_t v^t$
10:    Compute $x^{t+1} = x^t + \mathcal{C}_s(w^{t+1} - x^t)$ and send $\mathcal{C}_s(w^{t+1} - x^t)$ to workers
11:    **for** $i = 1, \ldots, n$ **do**
12:        Compute $x^{t+1} = x^t + \mathcal{C}_s(w^{t+1} - x^t)$
13:        Compute $g_i(x^{t+1})$ and send to server           ➤ Cheap communication of one float
14:    **end for**
15:    Compute $u^{t+1} = \frac{1}{n} \sum_{i=1}^{n} g_i(x^{t+1})$
16:    Compute $\lambda^{t+1} = \Pi_{\lambda \geq 0}(\lambda^t + \eta_t u^{t+1})$
17: **end for**

---

**Experiments.** Although a rigorous convergence analysis for Primal-dual remains open, we investigate its practical performance through empirical evaluation. We follow the same experimental setup as before and compare Safe-EF with Algorithm 2, analyzing its sensitivity to different dual initializations $\lambda^0$. We present our results in Figure 7, where we compare the objective and constraint after 500M samples, the number of samples required for Safe-EF to converge. As shown, different values of $\lambda^0$ have significant impact on the performance of Primal-dual. In contrast, Safe-EF that does not require additional tuning of hyperparameters and only slightly underperforms Primal-dual when $\lambda^0 = 2$.

## H. Additional Experiments

**Cartpole.** We repeat our safety experiment using the Cartpole environment from Brax (Freeman et al., 2021), with the exception of using $K/d = 0.01$ instead of $K/d = 0.1$. As before, we compare Safe-EF with EF14 (Seide et al., 2014),

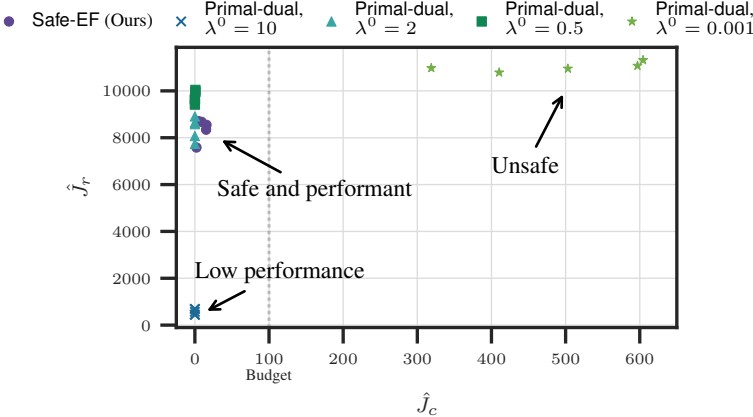

Figure 7: Objective and constraint values of Safe-EF compared to Primal-dual with different initialization values of $\lambda^0$. Each point represents a distinct experiment trial with a different random seed. Safe-EF ensures safety and achieves solid performance without requiring additional hyperparameter tuning.

EF21 (Richtárik et al., 2021) and Parallel-CRPO. The results are presented in Figure 8. Similarly to the experiments with the Humanoid, Safe-EF rapidly satisfies the constraints with only a slight performance reduction in the objective. EF14 outperforms Safe-EF, however violates the constraints. Further, EF21 diverges during the last part of training. Finally, as Parallel-CRPO does not employ compression at all, it requires significantly more gigabytes per worker to converge.

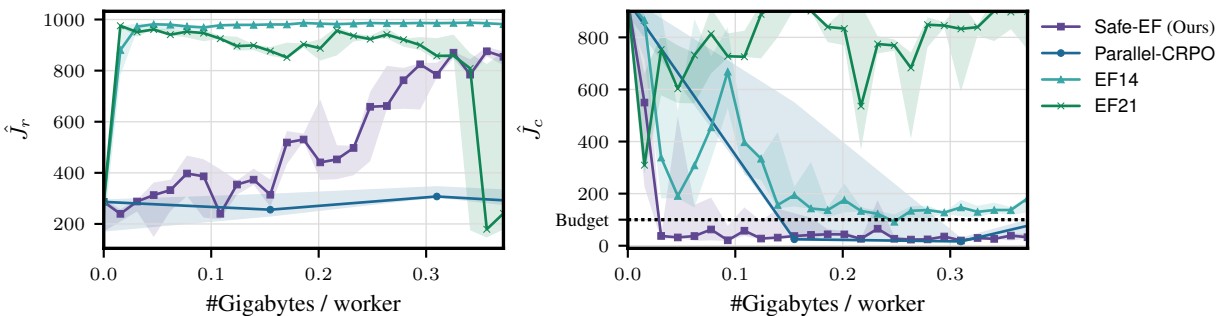

Figure 8: Objective and constraint in the Cartpole environment. Safe-EF satisfies the constraints while maintaining competitive performance.

**Price of compression.** We follow the same evaluation protocol used in Figure 3 however now, instead of measuring how many gigabytes are required to reach a certain benchmark performance, we use a fixed sample "budget", and evaluate the performance achieved by each algorithm under this budget. Accordingly, we record $\hat{J}_r$ after 100M and 500M samples, corresponding to 4883 and 24415 iterations respectively, for different values of $K/d$. We present the results in Figure 9. As shown, both Top-$K$ and Rand-$K$ perform well under diminishing values of $K/d$ after 500M samples. For a training budget of 100M samples, Top-$K$ significantly surpasses CGD and Rand-$K$.

**Non-distributed baseline.** We show that Safe-EF is able to find a non-trivial policy, by comparing it against Parallel-CRPO and its non-distributed variant, CRPO, where the latter is trained and evaluated only on the nominal model $p$. We present our results in Figure 10.

**Learning curves.** In Figure 11 we provide the full learning curves of the experiment trials used for Figures 3 and 9.

**Neyman-Pearson classification.** We test Safe-EF on Neyman-Pearson (NP) classification problem following the work of He et al. (2024). This statistical formulation aims to minimize type II error while enforcing an upper bound on type I error,

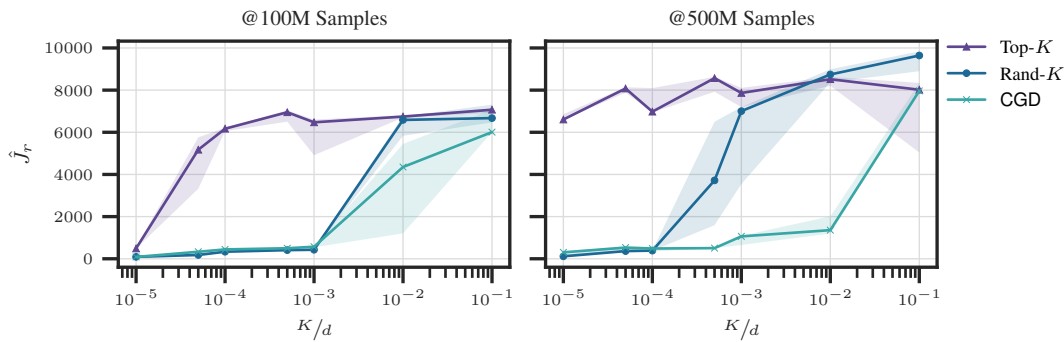

Figure 9: Performance for different compression ratios. Safe-EF with Top-$K$ and Rand-$K$ strategies outperform the CGD baseline. For a training budget of 500M samples, Top-$K$ reaches adequate performance, even under severe compression.

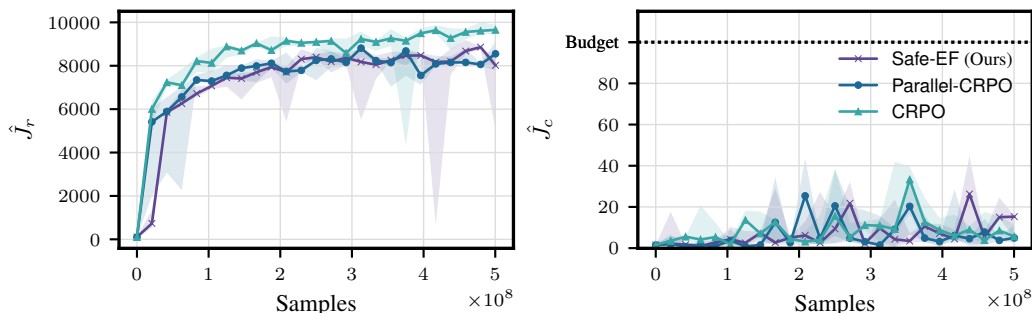

Figure 10: Safe-EF performance is only slightly degraded compared to a non-distributed baseline in terms of sample efficiency. However, in the distributed setup, as we observed in Figure 4, Safe-EF significantly outperforms Parallel-CRPO in communication efficiency.

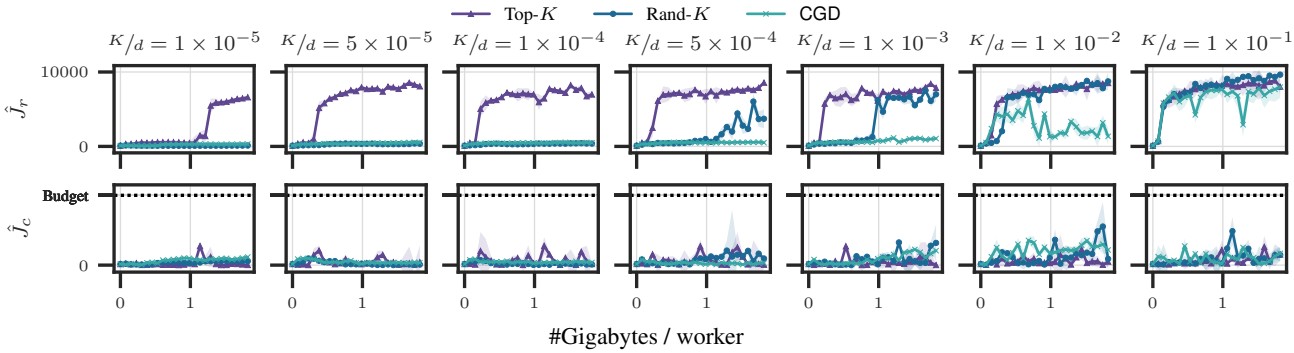

Figure 11: Objective and constraint learning curves for different compression ratio. Safe-EF with Top-$K$ outperforms Rand-$K$ and CGD, even under small compression values.

making it particularly relevant for applications with asymmetric misclassification costs, such as medical diagnosis. The NP classification is

$$\min_x f(x) = \frac{1}{n_0} \sum_{i=1}^{n_0} \phi(h_x, z_{i,0}), \quad \text{s.t.} \quad g(x) = \frac{1}{n_1} \sum_{i=1}^{n_1} \phi(h_x, z_{i,1}) \leq c,$$

where $f_x$ is a classifier parameterized by $x$ (3 layers MLP with 64 units in each layer and ReLu activation); $\phi$ is a cross-entropy loss; $\{z_{i,0}\}_{i=1}^{n_0}$ and $\{z_{i,1}\}_{i=1}^{n_1}$ are training samples from class 0 and class 1, respectively. The constraint ensures that the classification error for class 1 does not exceed a predefined threshold $c$. Our results are presented in Figure 12.

This experiment further supports the argument that Safe-EF is useful for federated learning by showing its effectiveness in a

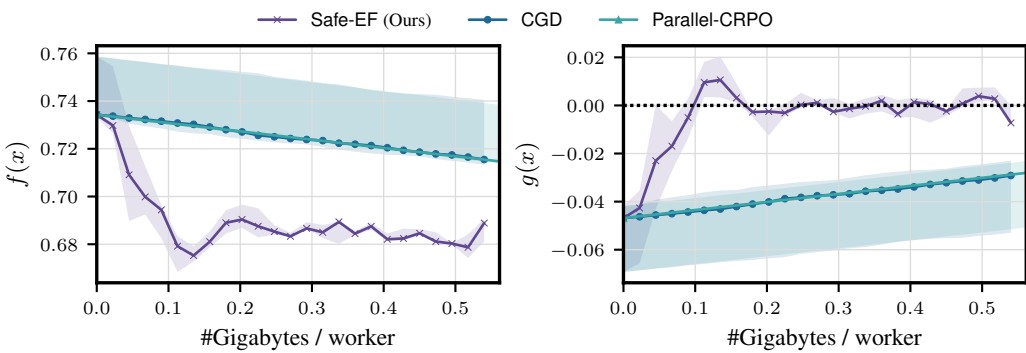

Figure 12: Objective and constraint for Neyman-Pearson classification. Compared to the CGD and Parallel-CRPO baselines, Safe-EF both satisfies the constraint and minimizes the loss while requiring significantly less communication overhead.

Table 1: The algorithms' hyperparameters used in the training from Section 6.1. Here $\gamma$ denotes the stepsize for all algorithms, $\beta$ is the momentum parameter for EF21M, and $\eta$ is the control stepsize for EControl.

|  | Safe-EF | CGD | EF21 | EF21M | EControl |
|---|---|---|---|---|---|
| $s = 0.1$ | $\gamma = 0.01$ | $\gamma = 0.01$ | $\gamma = 0.003$ | $\gamma = 0.01, \beta = 0.001$ | $\gamma = 0.003, \eta = 0.01$ |
| $s = 1.0$ | $\gamma = 0.01$ | $\gamma = 0.01$ | $\gamma = 0.003$ | $\gamma = 0.01, \beta = 0.001$ | $\gamma = 0.003, \eta = 0.01$ |
| $s = 1.0$ | $\gamma = 0.003$ | $\gamma = 0.01$ | $\gamma = 0.001$ | $\gamma = 0.001, \beta = 0.1$ | $\gamma = 0.001, \eta = 0.1$ |

---

**Algorithm 3** Synthetic data generation mechanism

---

1: **Parameters:** number of nodes $n$, dimension $d$, noise scalers $\zeta$ and $s$
2: Generate $\mathbf{A} \sim \mathcal{N}(0, \mathbf{I}) \in \mathbb{R}^{d \times d}$ and $x_0 \sim \mathcal{N}(0, \mathbf{I}) \in \mathbb{R}^d$
3: Normalize $\mathbf{A} \leftarrow \mathbf{A}/\|\mathbf{A}\|_F$
4: **for** $i = 1, \ldots, n$ **do**
5:     Generate $\mathbf{A}_i \sim \mathcal{N}(0, \mathbf{I}) \in \mathbb{R}^{d \times d}$
6:     Normalize $\mathbf{A}_i \leftarrow \mathbf{A}_i/\|\mathbf{A}_i\|_F$
7:     Shift $\mathbf{A}_i \leftarrow \mathbf{A} + s\mathbf{A}_i$
8:     Sample independently $\xi \sim \mathcal{N}(0, 1) \in \mathbb{R}^d$
9:     Compute $b_i = \mathbf{A}_i x_0 + \zeta \xi$
10: **end for**
11: **Return** $\{\mathbf{A}_i, b_i\}_{i=1}^n$

---

well-established classification framework.

# I. Additional Details on the Experimental Setup

**Data generation.** We generate matrices $\{\mathbf{A}_i\}_{i=1}^n$ and shifts $\{b_i\}_{i=1}^n$ according to Algorithm 3. Here parameter $s$ controls how different the matrices $\mathbf{A}_i$ are from each other. In our experiments, we vary $s \in \{0.1, 1.0, 10.0\}$ and set $\zeta = 10^{-3}$.

**Hyper-parameter tuning for Section 6.1.** For all algorithms mentioned in Section 6.1, we tune the stepsize $\gamma \in \{0.01, 0.003, 0.001, 0.0003, 0.0001, 0.00003\}$. For EF21M we tune the momentum parameter $\beta \in \{0.0001, 0.001, 0.01, 0.1, 0.5, 0.9\}$, and for EControl, we tune $\eta \in \{0.0001, 0.001, 0.01, 0.1, 0.5, 0.9\}$. The best hyperparameters are reported in Table 1.

**Humanoid.** We use the Humanoid environment implementation from Brax (Freeman et al., 2021) and extend it with an indicator cost function for whenever any one of the joint angles goes outside of a predefined limits. We perturb the dynamics

$p_i$ of each worker by sampling the ground's friction coefficient and the gear parameter of the joints' motors. Sampling is done with a uniform distribution, with a symmetric interval centered around the nominal value given in Brax.

**Cartpole.**    As with the Humanoid, we use the environment implementation provided by Brax. The cost function is an indicator for whenever the 'cart' exceeds a predefined distance from the center position. The dynamics are perturbed in the same fashion as the Humanoid, using a uniform distribution centered around nominal values. However in this experiment, we perturb the mass of the 'pole' and the gear parameter of the cart's motor.

**Hyper-parameters tuning for Section 6.2.**    As mentioned before, our implementation of Safe-EF builds on PPO (Schulman et al., 2017). We follow the standard follow the standard implementation provided in Brax, including their default hyper-parameters used for the Humanoid environment. Notably, in all of our experiments, we keep the default value $\gamma = 0.0003$, with Adam as optimizer (Kingma & Ba, 2014). In practice, we found the default set of parameters to work well with Safe-EF. The only deviation from these parameters is the entropy regularization coefficient, which we set to $0.01$ from $0.001$.

For more specific details, please use our open-source implementation `https://github.com/yardenas/safe-ef`.

