# OpenReview forum: "Safe-EF: Error Feedback for Non-smooth Constrained Optimization"
_ICML.cc/2025/Conference — ICML 2025 poster_

### Official Review · Reviewer_gSmf · 2025-03-11

**Overall Recommendation:** 3

**Summary:**

he paper establishes a convergence lower bound for the non-smooth convex distributed setting, where EF-21 and similar methods operate. Next, it proposes Safe-EF (Algorithm 1), an extension of EF14 (Seide et al., 2014) that incorporates safety constraints and bidirectional compression. Safe-EF is provably effective in non-smooth distributed settings and efficiently minimizes the objective function. The paper proves that the convergence rate of Safe-EF matches the aforementioned lower bound up to a numerical constant, assuming a constant accuracy of the server compression C₀.

Furthermore, the paper studies Safe-EF in practically relevant stochastic scenarios, where exact subgradients and function evaluations are unavailable. It establishes high-probability bounds in these settings.

Extensive experiments and ablation studies validate Safe-EF, demonstrating its effectiveness on the challenging task of distributed humanoid robot control.

Paper contains source code for repoducability.

**Claims And Evidence:**

No claims. The paper is strong and presents an interesting perspective for EF type methods in case of non-smooth convex distributed settings with constraints.

**Essential References Not Discussed:**

No. All is fine. With one exception.

Thanks authors for "B. Failure of CGD and EF21 in Non-smooth Convex Setting".

However, Convergence Upper Bound for EF21 in Smooth Convex Setting is not necessarily optimal selection of stepsize.
The work "Error Feedback Reloaded: From Quadratic to Arithmetic Mean of Smoothness Constants" https://openreview.net/pdf?id=Ch7WqGcGmb, https://arxiv.org/abs/2402.10774
Improves quadratic mean to arithmetic mean.

Please, revisit Chapter C for Smooth Convex Setting under this consideration.

**Ethical Review Concerns:**

No concerns.

**Experimental Designs Or Analyses:**

Yes. No issues.

**Methods And Evaluation Criteria:**

Yes

**Other Comments Or Suggestions:**

(1) Convergence Upper Bound for EF21 in Smooth Convex Setting should be revisited under a better setting of selecting meta-parameters

(2) I have observed \gamma = 1/\sqrt{T} as step-size in Figure 1, but this information is missed in other experiments.
Please be specific about stepsize selection (theoretical, tuned for specific instance, estimated in another form) in experiments from Figures 2-11 in Experiment sections. Please elaborate about your selection.

(3) Please provide information about the number of trials in experiments with error bars.

(4) Theorem 4.2 contains a reference to Appendix D. This suggests select \gamma and c as a function of T. Please elaborate more and provide implicit or explicit formulas in terms of the target discrepancy in function gap. Either provide the notion that T is the conceptual budget for iterations. But I strongly ask to elaborate on what constants should be estimated before Safe-EF can be launched.

**Other Strengths And Weaknesses:**

No.

**Questions For Authors:**

Please address suggestions (1) - (4). Under this condition, I am glad to recommend the paper for acceptance.

**Relation To Broader Scientific Literature:**

Error Feedback is a popular and immensely effective mechanism for fixing convergence issues that arise in distributed training. While EF was proposed almost a decade ago, and despite concentrated effort by the community to advance the theoretical understanding of this mechanism, there is still a lot to explore.

**Theoretical Claims:**

Yes.

---

> ### Author Rebuttal · Authors · 2025-03-31
>
> 1. We thank the reviewer for this valuable comment and pointing out this reference! Indeed, the workers cloning idea described in that paper can be extended to our result in convex (smooth) setting. We have revisited the convergence analysis of EF21 in our supplementary Theorem C.1. following the analysis from the proposed reference. The updated rate is $\mathbb{E}[f(x^T) - f(x^*)] \le \frac{R_{\mathcal{X}}^2}{\gamma T}\left(1+ \log\left(\frac{\gamma \Lambda_0 T}{R_{\mathcal{X}}^2}\right)\right),$
> where $$\gamma \le \mathcal{O}\left(\frac{\delta}{\frac{1}{n}\sum_{i=1}^nL_i}\right).$$ The potential improvement compared to our Theorem C.1. is in the difference between the average $\frac{1}{n}\sum_{i=1}^nL_i$ and the quadratic mean $\sqrt{\frac{1}{n}\sum_{i=1}^n L_i^2}$. This derivation follows directly from our analysis by replacing equation (15) in the proof of Theorem C.1. with more refined Lemma 7 in Appendix C.1. of the suggested paper. We will include the improved derivation in the next revision and cite the missing reference.
> 2. We appreciate this useful suggestion. The hyperparameters used in Section 6.1 are provided in Appendix I. We perform a grid search over several choices of $\gamma$ and select the one that gives the smallest train loss at the end of training. Table 1 in Appendix I gives concrete numbers for the selected step size for each heterogeneity level $s = 0.1, 1.0, 10.0$. In Safe RL experiment, we don’t tune the hyper-parameters and use the default step-size $\gamma = 10^{-4}$ from the (non-distributed) PPO implementation that is known to work well on standard RL benchmarks. The slackness parameter $c$ is set to $0$, which is sufficient to get decent performance for both tasks. We did not try to further tune these parameters and agree that, in general, these parameters are problem-dependent. We will clarify these details in the revision, thank you for this advice!
> 3. Thank you for bringing this to our attention. We highlight that we use $5$ seeds for all of our RL experiments, as also mentioned in the “Setup” paragraph of section 6.2.
> 4. We emphasize that the choice of $c$ highly depends on the application. The choice of $c$ is dictated by the fact of how much constraint intolerance we can allow. From theory perspective, the choice of hyperparameters $\gamma = \mathcal{O}(\frac{R\sqrt{\delta\delta_s }}{M\sqrt{T}})$ and $c=\mathcal{O}(\frac{MR}{\sqrt{\delta\delta_s T}})$ depends on the compression levels $\delta$ and $\delta_{s}$, the gradient bound $M$, number of iterations $T,$ and the upper bound on the initial distance to the optimal set $R=\\|x^0-x^*\\|^2.$ The compression levels and the number of iterations are typically known beforehand as it is a user who sets them. Therefore, there is no need to estimate them. To set $\gamma$ and $c$ we should provide an estimate on $M$ and $R.$  For example, we can estimate $M = \max_i\\|f_i^\prime(x^0)\\|$ and set $R \geq \\|x^0\\|,$ if $x^*$ is close to zero. In general, we believe that choosing $\gamma, c = \mathcal{O}(1/\sqrt{T})$ or diminishing $\gamma, c = \mathcal{O}(1/\sqrt{t})$, $1 \leq t \leq T$, can be a good starting point.
>
> We appreciate the reviewer for their thoughtful feedback! We believe we have addressed the main concerns (1–4) and would appreciate if you could confirm that everything is resolved and consider adjusting the score.

---

### Official Review · Reviewer_TUHp · 2025-03-12

**Overall Recommendation:** 4

**Summary:**

In this paper, the authors investigate a non-smooth optimization setting with bounded gradients in the context of Top-K compression. Communication compression using contractive compressors (e.g., Top-K) is commonly preferred in practice due to its efficiency; however, it can significantly degrade performance if not properly managed.

The authors demonstrate a scenario in which Error Feedback (EF) can lead to divergence but also propose a method to mitigate this issue. They contribute to the theoretical understanding of EF in the canonical non-smooth convex setting by establishing new lower complexity bounds for first-order algorithms under contractive compression.

To address the limitations of existing approaches, the authors introduce Safe-EF, a novel algorithm that achieves the established lower bound (up to a constant factor) while incorporating safety constraints crucial for practical applications. Furthermore, they extend their approach to the stochastic setting, enhancing its applicability.

The proposed method is thoroughly evaluated through extensive experiments in a reinforcement learning setup, specifically simulating distributed humanoid robot training. The results demonstrate that Safe-EF effectively maintains safety constraints while reducing communication complexity, highlighting its potential for real-world deployment in distributed optimization scenarios.

**Claims And Evidence:**

Most of the claims in the paper are well-formulated and supported by either rigorous proofs or relevant references. The arguments are clearly articulated, contributing to the overall clarity and coherence of the work.

However, I identified one claim that might be considered questionable:

**Lines 102–104 (right side, page 2):**
*"Despite their importance, constrained optimization with communication compression remains under-explored."*

While I believe this statement is accurate, it would benefit from further elaboration. Providing additional context, references, or a more detailed discussion would help clarify the extent to which this area has been explored and why it remains an open challenge. This would make the claim more accessible to a broader audience and strengthen its impact.

**Essential References Not Discussed:**

There are two papers that are discussing biased SGD and consider contractive compression in details, which is highly related to the current paper.

Ajalloeian, Ahmad, and Sebastian U. Stich. "On the convergence of SGD with biased gradients." arXiv preprint arXiv:2008.00051 (2020).

Demidovich, Yury, et al. "A guide through the zoo of biased SGD." Advances in Neural Information Processing Systems 36 (2023): 23158-23171.

**Experimental Designs Or Analyses:**

In the experimental section, the authors conduct experiments on synthetic data to illustrate their theoretical findings.

Subsequently, they explore the application of Safe-EF in reinforcement learning. In this setup, each worker represents a humanoid robot that collects noisy measurements of certain utility and constraint functions to solve a constrained Markov decision process (CMDP).

The experimental evaluation consists of several key components:

Comparison with CGD: The authors first evaluate Safe-EF using Top-K and Rand-K sparsifiers and compare its performance with a constrained version of CGD employing a Top-K sparsifier.

Constraint Satisfaction Analysis: They then analyze Safe-EF’s ability to satisfy constraints, comparing it against unsafe error feedback algorithms (EF14 and EF21). Additionally, they benchmark Safe-EF against a parallel variant of CRPO, a CMDP solver that enforces constraints through the subgradient switching method.

Effect of the Number of Workers: The authors examine how the performance of Safe-EF varies with the number of available workers, providing insights into its scalability.

Effect of Batch Size: Finally, they study how different batch sizes impact the method’s performance.

The presented experiments are valuable and well-structured, providing strong empirical support for the proposed approach. However, exploring additional settings could further strengthen the study by demonstrating the robustness of Safe-EF across a broader range of scenarios.

**Methods And Evaluation Criteria:**

In this paper, the authors present theoretical convergence guarantees alongside experimental results for reinforcement learning tasks. These criteria are both relevant and well-justified, as they ensure that the proposed method is supported by rigorous mathematical analysis while also being validated through practical implementation. The combination of theoretical and empirical evaluation strengthens the credibility of the approach, demonstrating its effectiveness in real-world reinforcement learning scenarios. This dual perspective enhances the overall contribution of the paper by providing both fundamental insights and practical applicability.

**Other Comments Or Suggestions:**

Please review the issues identified in the previous sections and ensure they are addressed appropriately.

**Other Strengths And Weaknesses:**

The paper is well-written and presents its ideas in a clear and structured manner. The theoretical contributions are strong and well-founded, making a significant impact on the understanding of the problem. Additionally, the figures and plots are carefully formatted, with appropriate use of grids and markers, which enhances readability and interpretability. The attention to detail in visualization and presentation reflects the authors’ commitment to clarity and precision.

The stochastic case extension section could be further expanded, as its current description lacks sufficient detail. Providing a more comprehensive discussion would enhance the clarity and depth of this section.

**Questions For Authors:**

Do you have any insights on how to eliminate the additional logarithmic factors in order to achieve the optimal rate?

**Relation To Broader Scientific Literature:**

The key contributions of the paper are closely related to the broader scientific literature on optimization. In particular, the derivation of lower bounds is especially valuable, as such results are typically more challenging to establish and provide fundamental insights into the theoretical limits of the considered optimization setting.

**Theoretical Claims:**

In this paper, the authors establish tight convergence guarantees by providing both lower and upper bounds for the considered setting of non-smooth convex optimization with contractive compression operators.

The lower bounds presented in this work are particularly valuable, as deriving them is generally more challenging than establishing upper bounds. The proposed algorithm achieves a convergence rate that matches the lower bound (Equation 9), up to numerical and logarithmic factors.

While the presence of additional logarithmic factors could be perceived as a drawback, this is a relatively minor issue compared to the overall theoretical contribution of the paper. The extension to the stochastic setting further enhances the significance of the work, broadening its applicability.

I have briefly reviewed the analysis, and it appears to be correct. However, there is always a possibility that some errors may have been overlooked, so I recommend a thorough double-check for accuracy.

---

> ### Author Rebuttal · Authors · 2025-03-31
>
> 1. We appreciate the reviewer’s comment and agree that further elaboration on the lack of prior work in this area would strengthen the manuscript. To the best of our knowledge, constrained optimization with communication compression remains largely unexplored. In fact, besides the two works we cited—(Fatkhullin et al, 2021), which considers projection-based methods, and (Nazykov et al, 2024), which employs linear minimization oracles—we are unaware of any other studies that explicitly address even simple constrained problems in the presence of contractive compression. In the more limited setting with Rand-$K$ operator (unbiased compressor), the recent work (Laurent Condat and Peter Richt{\'a}rik. Randprox: Primal-dual optimization algorithms with randomized proximal updates, 2023) has established some results in the proximal setting. However, this is still limited to constraints/proximal functions with simple structure (e.g., Euclidean ball, simplex, affine constraints, etc).
>
>     The situation becomes even more challenging when considering the general constraints of the form (4) in the manuscript, as no existing work tackles such cases. Fundamentally, addressing this problem requires overcoming new technical difficulties arising from the interaction between constrained optimization and communication compression. Specifically, in addition to the usual error accumulation from compression in unconstrained settings, constrained problems introduce further complexities: (i) compression errors affect the feasibility of iterates, making it harder to ensure constraint satisfaction, and (ii) primal-dual methods introduce an additional projection step, which requires the control of errors from projection, the objective and constraint updates simultaneously. These challenges necessitate novel algorithmic techniques, which we discuss in the subsequent paragraph of our introduction by distinguishing between primal-only and primal-dual approaches.
>
>     We will revise the manuscript to clarify these points and emphasize the fundamental difficulties that remain open in this research direction.
> 2. We thank the reviewer for providing the references. We agree that these studies are relevant to our work, and we will incorporate them in the introduction. These papers examine a general case of SGD with biased stochastic gradients. The special case of their analyses is CGD, described in the introduction. However, their analysis does not fully align with our setting because they consider (i) a single-node training regime $n=1$, (ii) $L$-smooth functions while we focus on a non-smooth setting, and (iii) a plain SGD without a memory buffer to mitigate the bias (e.g., the compression error). Nonetheless, we agree that extending our results from unbiased gradients to biased ones is an interesting research direction for future work.
> 3. Currently, a detailed description of the stochastic setting is provided in section F due to the page limit. We will try to reorganize the main body to fit the description in the main paper. We appreciate this suggestion.
> 4. The logarithmic dependency on the failure probability $\beta$ is a typical outcome of the high probability analysis that comes from the use of concentration inequalities [1,2]. To the best of our knowledge, the logarithmic dependency on the failure probability cannot be avoided as there exists a lower bound for a sample complexity of any learning algorithm Theorem 5.2 in [3]. We will add a note on this aspect in the revised version of the paper.
>
>     [1] Nemirovski et al., Robust stochastic approximation approach to stochastic programming, SIAM 2009
>
>     [2] Ghadimi \& Lan, Optimal stochastic approximation algorithms for strongly convex stochastic composite optimization i: A generic algorithmic framework, SIAM 2012
>
>     [3] Anthony \& Bartlett, Neural Network Learning: Theoretical Foundations, Cambridge University Press, 2009

---

> > ### Comment · Reviewer_TUHp · 2025-04-02
> >
> > Thank you to the authors for their responses!
> >
> > I appreciate the clarifications and look forward to the promised adjustments being implemented. I believe the work is of high quality and will maintain my score of acceptance.
> >
> > Best regards,
> > Reviewer

---

### Official Review · Reviewer_K1fK · 2025-03-13

**Overall Recommendation:** 4

**Summary:**

The paper presents Safe-EF, an error feedback (EF) algorithm designed for non-smooth constrained optimization in distributed settings. It establishes lower complexity bounds for first-order algorithms with contractive compression and introduces Safe-EF, which matches these bounds while ensuring safety constraints. The algorithm uses bidirectional compression and a constraint-aware switching mechanism. The effectiveness of Safe-EF is demonstrated in a reinforcement learning (RL) setting, specifically in distributed humanoid robot training, where it reduces communication overhead while maintaining constraint satisfaction.

**Claims And Evidence:**

The claims about Safe-EF’s theoretical guarantees (matching lower bounds) are well-supported with clear derivations. The empirical results effectively show that Safe-EF outperforms EF21 and CGD in terms of communication efficiency and constraint satisfaction. However, the claim that Safe-EF applies to federated learning (FL) is less substantiated, as the experiments do not fully capture the typical FL challenges (e.g., heterogeneous data, decentralized aggregation).

**Essential References Not Discussed:**

The paper cites relevant work in error feedback, compressed optimization, and distributed learning. I do not see any critical omissions.

**Experimental Designs Or Analyses:**

The experiments convincingly show Safe-EF’s superiority over EF21 and CGD. However, some key experimental details are missing:

-No clear explanation for why EF14 works while EF21 fails in non-smooth settings.

-No guidance on practical choices for γ (step size) and c (constraint threshold), making Safe-EF harder to implement.

-The humanoid RL task is well-chosen, but its setup is not clearly described, making reproducibility difficult.

**Methods And Evaluation Criteria:**

The problem setting and evaluation criteria are reasonable for distributed optimization. The experiments use a multi-agent RL setting, which is a valid but non-standard way to test federated learning algorithms. The humanoid training task is complex and relevant, but an additional FL benchmark would strengthen the argument that Safe-EF is useful for FL.

**Other Comments Or Suggestions:**

Move key proofs to the main text instead of the appendix.

**Other Strengths And Weaknesses:**

Strengths:

Novel bidirectional compression approach.

Strong theoretical guarantees.

Clear improvements over EF21 and CGD in non-smooth settings.

Weaknesses:

Federated learning framing is weak; an FL benchmark would make the claims stronger.

The introduction is too long and should be split into introduction, practical challenges, and related work.

Lack of practical guidance on hyperparameters γ and c.

**Questions For Authors:**

Why does EF14 work while EF21 fails? More insight into this difference would strengthen the theoretical claims.

What values of γ and c should be used in practice? Can practical guidance be provided for choosing these parameters?

Could Safe-EF be tested in a standard FL benchmark? This would help justify its relevance to federated learning.

Would Safe-EF generalize to non-convex settings? Since many real-world RL problems are non-convex, would it still maintain convergence?

**Relation To Broader Scientific Literature:**

Safe-EF builds on prior error feedback methods (EF21, EF14, CGD) and contributes to communication-efficient distributed optimization. However, its positioning in federated learning needs stronger justification, as it lacks evaluation on heterogeneous FL datasets.

**Theoretical Claims:**

I reviewed the lower bound proof and the convergence analysis for Safe-EF. The derivations appear correct, and the methodology follows standard approaches in optimization theory. The results for error feedback in non-smooth settings are a useful contribution. However, the key proofs should be moved to the main text rather than being relegated to the appendix.

---

> ### Author Rebuttal · Authors · 2025-03-31
>
> 1. We appreciate the reviewer's high evaluation of our theoretical guarantees and empirical results. For empirical evaluation, we (i) test our method on a well-controlled synthetic environment with different levels of heterogeneity and (ii) provide an extensive ablation study in more challenging continuous control environments (Cartpole and Humanoid). We believe the latter Federated Safe RL setting is one of the most relevant and challenging applications of our theory that tests the real potential of Safe-EF. Different transition dynamics $p_i$ of each worker make the loss $f_i$ different, resulting in a highly heterogeneous setting. We mentioned this aspect in Section 6.2 and Appendix I, but we will put more emphasis on the heterogeneity aspect in the revision. In our work, we focus on non-smoothness, safety constraints, biased and bidirectional compression. Decentralized aggregation is an important topic, and we leave it to future work.
> 2. We test Safe-EF on Neyman-Pearson (NP) classification problem following the work (He et al., Federated Learning with Convex Global and Local Constraints, TMLR 2024). This statistical formulation aims to minimize type II error while enforcing an upper bound on type I error, making it particularly relevant for applications with asymmetric misclassification costs, such as medical diagnosis. The NP classification is $$\min_xf(x)=\frac{1}{n_0}\sum_{i=1}^{n_0}\phi(h_{x}, z_{i,0}),\text{ s.t. }\quad g(x)=\frac{1}{n_1}\sum_{i=1}^{n_1}\phi(h_{x}, z_{i,1})\le c,$$ where $f_x$ is a classifier parameterized by $x$ (3 layers MLP with 64 units in each layer and Relu activation); $\phi$ is a CE loss; $\\{z_{i,0}\\}\_{i=1}^{n_0}$ and $\\{z_{i,1}\\}_{i=1}^{n_1}$ are training samples from class 0 and class 1, respectively. The constraint ensures that the classification error for class 1 does not exceed a predefined threshold $c$. We include the results in https://ibb.co/SXYqWQtB. This benchmark further supports the argument that Safe-EF is useful for federated learning by showing its effectiveness in a well-established classification framework. We believe the Federated Safe RL remains the most relevant application naturally formulated as a stochastic optimization of the form (72), (73) in Appendix F (unlike the standard supervised learning task with finite dataset), but we welcome the reviewer for a discussion if this new set of experiments strengthens our argument that Safe-EF is useful for FL.
> 3. While we show our hard instance (functions, constraints, compressors) in the main paper, we will detail key steps of the proof in the revised version of the paper.
> 4. The design of EF21 algorithm can be seen as a variance reduction (VR) technique. In a nonsmooth setting, gradients can change abruptly. Thus, VR mechanisms such as EF21 may fail. In contrast, EF14 algorithm's design is based on tracking the compression error, and the control of gradient change is not needed.
> 5. We acknowledge reviewer's concern. From theory, in the choice $\gamma=O(\frac{R\sqrt{\delta\delta_s }}{M\sqrt{T}})$ and $c=O(\frac{MR}{\sqrt{\delta\delta_s T}})$ parameters $\delta,\delta_s,T$ are typically set by a user. Thus, there is no need to estimate them. We can estimate $M=\max_i\\|f_i^\prime(x^0)\\|$ and $R>\\|x^0\\|,$ if $x^*$ is close to zero. In general, we believe that choosing $\gamma,c=O(1/\sqrt{T})$ or $O(1/\sqrt{t})$ can be a good starting point. In our experiments, $\gamma=$1e-4 is chosen based on existing default value that is known to work well for PPO on standard RL benchmarks. The slackness parameter $c$ is set to 0, which is sufficient to get decent performance for both tasks. We did not try to further tune these parameters and agree that, in general, these parameters are problem-dependent.
> 6. Thank you for pointing this out. Based on your suggestion, we will add additional details on our experimental setup to Appendix I. We kindly ask the reviewer to point out more specifically which parts in the exposition should be further clarified.
> Additionally, we emphasize that the code for our experiments can be found in https://anonymous.4open.science/r/safe-ef-3ABC, as mentioned in Appendix I. We believe that this makes our experiments fairly easy to reproduce.
> 8. We reference all relevant work within each subsection or paragraph of the introduction while discussing practical challenges and existing solutions. While we can divide the first section into an introduction and practical challenges, separating the related work into a standalone section would not be appropriate. Any additional related work that is not directly relevant to our main narrative is included in the appendix.
> 9. Investigating convergence in the non-convex setting is indeed an interesting and important question motivated by real-world RL problems. We believe our analysis can be extended to weakly convex functions, i.e., $f(x)+r\\|x\\|^2$ and $g(x)+r\\|x\\|^2$ are convex following, e.g., (Jia and Grimmer, 2022); (Boob et al., 2023)

---

### Decision · Program_Chairs · 2025-05-01

**Decision:**

Accept (poster)

**Comment:**

This paper proposes Safe-EF algorithm for non-smooth constrained optimization in distributed environments under contractive communication compression. The main contributions include (i) tight lower complexity bounds for first-order methods under such compression and (ii) the Safe-EF algorithm, which matches these bounds (up to constants) while satisfying practical safety constraints through bidirectional compression. The theoretical contributions are strong and well-motivated. All reviewers agree that the lower-bound derivation is a valuable and nontrivial addition to the literature. The experimental evaluation focuses on a distributed reinforcement learning whose results are both compelling and challenging.